METHODS AND RESOURCES

# Control of stereocilia length during development of hair bundles

**Jocelyn F. Krey**[1], **Paroma Chatterjee**[1], **Julia Halford**[1], **Christopher L. Cunningham**[2], **Benjamin J. Perrin**[3], **Peter G. Barr-Gillespie**[1]*

**1** Oregon Hearing Research Center and Vollum Institute, Oregon Health & Science University, Portland, Oregon, United States of America, **2** Pittsburgh Hearing Research Center, University of Pittsburgh School of Medicine, Pittsburgh, Pennsylvania, United States of America, **3** Department of Biology, Indiana University-Purdue University Indianapolis, Indianapolis, Indiana, United States of America

* gillespp@ohsu.edu

**Data Availability Statement:** Underlying data (e.g., measurements of individual stereocilia in hair bundles) and aggregated data used to construct figures have been deposited in figshare: Krey, JF,

## Abstract

Assembly of the hair bundle, the sensory organelle of the inner ear, depends on differential growth of actin-based stereocilia. Separate rows of stereocilia, labeled 1 through 3 from tallest to shortest, lengthen or shorten during discrete time intervals during development. We used lattice structured illumination microscopy and surface rendering to measure dimensions of stereocilia from mouse apical inner hair cells during early postnatal development; these measurements revealed a sharp transition at postnatal day 8 between stage III (row 1 and 2 widening; row 2 shortening) and stage IV (final row 1 lengthening and widening). Tip proteins that determine row 1 lengthening did not accumulate simultaneously during stages III and IV; while the actin-bundling protein EPS8 peaked at the end of stage III, GNAI3 peaked several days later—in early stage IV—and GPSM2 peaked near the end of stage IV. To establish the contributions of key macromolecular assemblies to bundle structure, we examined mouse mutants that eliminated tip links (*Cdh23*$^{v2J}$ or *Pcdh15*$^{av3J}$), transduction channels (*Tmie*$^{KO}$), or the row 1 tip complex (*Myo15a*$^{sh2}$). *Cdh23*$^{v2J/v2J}$ and *Pcdh15*$^{av3J/av3J}$ bundles had adjacent stereocilia in the same row that were not matched in length, revealing that a major role of these cadherins is to synchronize lengths of side-by-side stereocilia. Use of the tip-link mutants also allowed us to distinguish the role of transduction from effects of transduction proteins themselves. While levels of GNAI3 and GPSM2, which stimulate stereocilia elongation, were greatly attenuated at the tips of *Tmie*$^{KO/KO}$ row 1 stereocilia, they accumulated normally in *Cdh23*$^{v2J/v2J}$ and *Pcdh15*$^{av3J/av3J}$ stereocilia. These results reinforced the suggestion that the transduction proteins themselves facilitate localization of proteins in the row 1 complex. By contrast, EPS8 concentrates at tips of all *Tmie*$^{KO/KO}$, *Cdh23*$^{v2J/v2J}$, and *Pcdh15*$^{av3J/av3J}$ stereocilia, correlating with the less polarized distribution of stereocilia lengths in these bundles. These latter results indicated that in wild-type hair cells, the transduction complex prevents accumulation of EPS8 at the tips of shorter stereocilia, causing them to shrink (rows 2 and 3) or disappear (row 4 and microvilli). Reduced rhodamine-actin labeling at row 2 stereocilia tips of tip-link and transduction mutants suggests that transduction's role is to destabilize actin filaments there. These results suggest that regulation of stereocilia length occurs through EPS8 and that CDH23

and Barr-Gillespie, PG. Control of stereocilia length during development of hair bundles. 2022. figshare. https://doi.org/10.6084/m9.figshare.21632636.v2.

**Funding:** This work was supported by the National Institute on Deafness and Other Communication Disorders (R01DC002368 to PGBG, R21DC019195 to CLC and R01DC015495 to BJP). The funders had no role in study design, data collection and analysis, decision to publish, or preparation of the manuscript.

**Competing interests:** The authors have declared that no competing interests exist.

**Abbreviations:** CV, coefficient of variation; F-actin, filamentous actin; Het, heterozygote; IHC, inner hair cell; IVC, individually ventilated cage; KO, knockout; OHC, outer hair cell; P, postnatal day; PSF, point-spread function; ROI, regions of interest; SIM, structured illumination microscopy.

and PCDH15 regulate stereocilia lengthening beyond their role in gating mechanotransduction channels.

## Introduction

Sensory hair cells of the inner ear are distinguished by the staircase arrangement of the approximately 100 stereocilia in their hair bundles, the sensory organelle that converts mechanical stimuli into electrical signals [1,2]. The staircase architecture enables directional sensitivity of mechanotransduction [3] and increases transduction sensitivity [4]. In the mature mammalian cochlea, stereocilia of inner hair cells (IHCs) and outer hair cells (OHCs) are arranged in 3 rows, albeit with distinct stereocilia lengths and widths [5].

During development, stereocilia in each row differentially widen and lengthen in specific stages [1,6]. MYO15A and EPS8 are found together at all stereocilia tips during early postnatal development and likely control the initial lengthening of stereocilia [7]. A complex of GPSM2 and GNAI3, coupled to MYO15A and EPS8 by WHRN, catalyzes lengthening of stereocilia beyond about a micrometer in postnatal stereocilia development [7–9]. This complex concentrates exclusively in row 1 after P7 and hence is referred to here as the row 1 tip complex. The 5 proteins together undergo phase separation, and the presence of GPSM2 both enhances phase separation and promotes F-actin bundling by EPS8 [10,11], activity that presumably underlies stereocilia lengthening. Mutations in any of the genes encoding these 5 proteins leads to hair bundles that have short stereocilia and a minimal staircase, i.e., only very small changes of stereocilia length in successive rows [12].

Lengths of the shorter rows of stereocilia are regulated differently. Several other proteins, including TWF2, EPS8L2, and CAPZB, are found predominantly at row 2 tips [13–15]; these proteins have actin-filament capping activity, so their presence may slow lengthening. Stereocilia lacking *Capzb* are narrow, then shorten and disappear [14], suggesting that CAPZB (and its binding partner TWF2) contribute both to stereocilia widening but also length stability.

Tension in tip links, extracellular filaments that interconnect stereocilia rows, elicits transduction currents by opening cation-conducting mechanotransduction channels [2,16]. Transduction regulates the final dimensions of stereocilia. Blockade of transduction channels elicits stereocilia shortening, suggesting a dynamic balance between actin polymerization and depolymerization [17]. Establishment of transduction currents requires the small membrane protein TMIE [18] and either TMC1 or TMC2, 2 channel-like proteins [19,20]. An ordered stereocilia staircase still forms in mouse mutants lacking either *Tmie* or both *Tmc1* and *Tmc2* [18,19], but row-specific distinctions of stereocilia width and length become more muted [6,21]. The presence of transduction-channel proteins also regulates the localization of proteins specific for row 1 and row 2 [6]. Although row 1 lacks transduction [22], in transduction mutants, GNAI3 and GPSM2 do not accumulate substantially at row 1 tips over postnatal development [6]. Significantly, transduction-channel proteins do localize to row 1 during early postnatal development [23].

Tip links are made up of dimers of CDH23, which project from the side of a taller stereocilium, and dimers of PCDH15, projecting from tips of short stereocilia [24–26]. The molecular motor MYO7A positions both molecules and tensions tip links [27–29]. All 3 of these proteins are members of the Usher I protein complex [30], and all are necessary for formation of tip links and activation of transduction. CDH23 and PCDH15 contribute to transient lateral links [24,26,31,32], which interconnect stereocilia along their shafts during development, as well as

kinocilial links [33,34], which connect several central row 1 stereocilium to the axonemal kino-cilium. Mutant mouse lines lacking functional genes for *Cdh23* or *Pcdh15* develop ragged stair-cases, with stereocilia of irregular lengths but relatively uniform diameters [35–37].

When tip links are under tension, the stereocilia membrane lifts off of the underlying F-actin core, a phenomenon called membrane tenting [38,39]; once the membrane is pulled away, actin polymerization is enhanced, especially on the side of stereocilium where the tip link anchors [17,40]. This phenomenon gives row 2 stereocilia a pronounced beveled shape [17,40]. These results suggest the possibility that tension in tip links promotes actin polymeri-zation in the shorter stereocilia rows, independent of channel gating [41]. Because tip links gate transduction channels [38], *Cdh23* and *Pcdh15* mutants lose both transduction and mem-brane tenting [37], and shorter stereocilia in homozygous *Cdh23* and *Pcdh15* mutants have rounded tips instead of beveled ones [37].

F-actin can be depolymerized at stereocilia tips due to the action of the ADF/CFL family members DSTN and CFL1 [42,43]; these proteins sever actin filaments, which can lead to dis-solution of actin structures [44]. Proteins in the ADF/CFL family typically associate with WDR1 (AIP1), which is located in stereocilia [45] and participates in length regulation [42]. DSTN and CFL1 are specifically located at row 2 stereocilia tips during early postnatal devel-opment, and this localization is disrupted in mutants lacking mechanotransduction [43]. Actin dynamics controlled by DSTN and CFL1 may underlie pruning of the shortest rows of stereocilia that occurs during hair-bundle maturation [46].

Thus, at least 5 key multimolecular assemblies control stereocilia dimensions: (1) tip links; (2) transduction channels; (3) the row 1 complex; (4) row 2 cappers; and (5) actin-severing proteins. Each of these assemblies is activated at different times during development and is responsible for 1 or more steps of hair-bundle development. Here, we defined with higher pre-cision the changes in stereocilia dimensions during postnatal development of apical IHCs from postnatal day 0.5 (P0.5) to P21.5. Moreover, we used mutant mice to define more specifi-cally the role of tip links on bundle development and the localization of tip-protein complexes. We examined stereocilia dimensions and row protein localization in *Pcdh15*^av3J^ and *Cdh23*^v2J^ null hair cells, as well as in *Tmie*^KO^ and *Myo15a*^sh2^ null hair cells. Besides gating transduction channels, we found that the tip-link proteins play an essential role in regulating stereocilia lengths; they also have a significant impact on accumulation of row-specific proteins at stereo-cilia tips, which in turn modulate stereocilia length. These observations complement previous results and provide us with a more comprehensive molecular understanding of bundle devel-opment in this hair-cell type.

## Results

### Quantitation of stereocilia actin-core dimensions using lattice structured illumination microscopy

We determined dimensions of stereocilia F-actin cores, labeled with fluorescent phalloidin, which were imaged in IHCs of C57BL/6 mice at precise times during early postnatal develop-ment. We used IHCs from the basal half of the apical turn of the cochlea, corresponding to 17% to 33% of cochlear length. Our measurements should reflect native lengths and widths; dimensions of mildly fixed, phalloidin-labeled stereocilia are not significantly different from dimensions of live stereocilia labeled with membrane dyes [47]. We improved resolution over conventional confocal microscopy by using lattice structured illumination microscopy (lattice SIM) [48–50], which has a point-spread function (PSF) of approximately 150 nm under our conditions [50]; typical confocal microscopy PSFs are approximately 230 nm [51]. We ren-dered image surfaces from each phalloidin-stained hair bundle (Fig 1A–1F); rendered surfaces

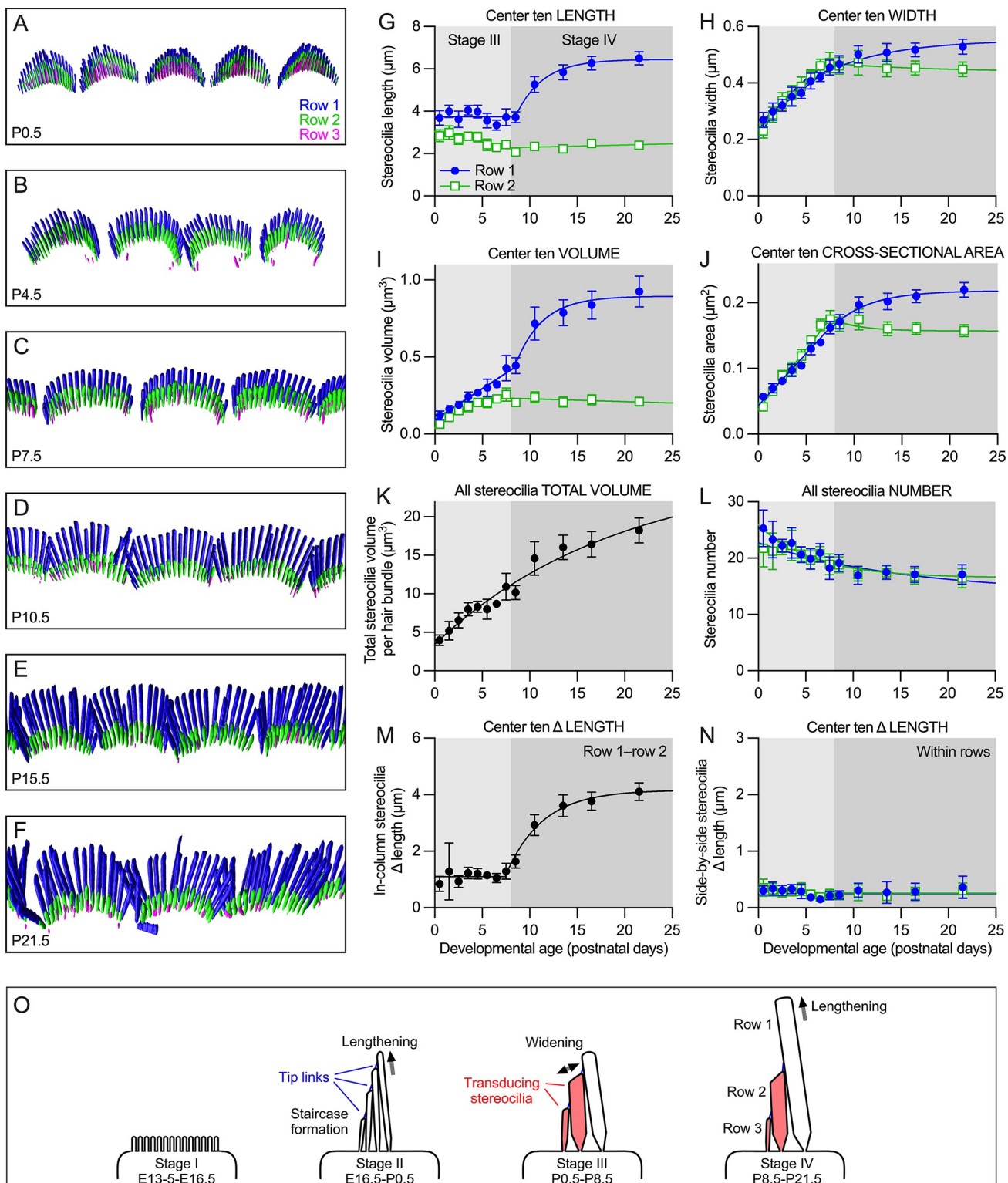

**Fig 1. Stereocilia dimensions during development of C57BL/6J apical IHCs.** (**A–F**) Imaris reconstruction of phalloidin-labeled IHC stereocilia from indicated ages. Stereocilia surfaces are color-coded according to row; each box is 14 × 35 µm. Imaris reconstructions show overlapping hair bundles of adjacent cells, presumably an artefact of sample preparation. (**G–N**) IHC stereocilia dimension measurements using reconstructed stereocilia surfaces. (**G**) Rows 1 and 2 stereocilia length during development; center 10 stereocilia in each row. Row 1 data were fit with a line with slope of zero from P0.5–P7.5 followed by an exponential climb from P7.5–P21.5; row 2 data were fit with an exponential decline. (**H**) Rows 1 and 2 stereocilia width (center 10). In

the panel, data for row 1 and for row 2 from P0.5–P7.5 were fit with an exponential climb; data for 2 from P7.5–P21.5 were fit with an exponential climb. The data could also be fit linearly; between P0.5 and P7.5, the slope for row 1 was 0.026 μm • day$^{-1}$ (95% confidence interval of 0.025–0.026), while the row 2 slope was 0.034 μm • day$^{-1}$ (95% confidence interval of 0.032–0.036). (**I**) Volume per individual row 1 or row 2 stereocilium (center 10). Data were fit with a linear increase from P0.5–P7.5, following by an exponential climb (row 1) or decline (row 2) from P7.5–P21.5. (**J**) Rows 1 and 2 stereocilia cross-sectional area (center 10). Data were fit with a linear increase from P0.5–P7.5, followed by an exponential climb (row 1) or decline (row 2) from P7.5–P21.5. (**K**) Total stereocilia volume per cell using rows 1, 2, and 3 stereocilia. Data were fit with an exponential climb. (**L**) Number of stereocilia per cell. Data were fit with an exponential decline. (**M**) Length difference between row 1 and row 2 stereocilia in the same column. Data were fit with a line with slope of zero from P0.5–P7.5, followed by an exponential climb from P7.5–P21.5. (**N**) Length difference between adjacent (side-by-side) stereocilia in the same row. Data were fit by exponential declines. For each plot, dimension measurements of stereocilia from each bundle were averaged to give an individual cell mean; then all means for individual cells (5–6 cells from each of 3–4 cochleas) were averaged and plotted ± SEM. (**O**) Diagrams of IHC bundle structure at each Tilney stage in mouse cochlea. Tip links are indicated as blue lines connecting adjacent stereocilia; they appear during stage II. Transducing stereocilia are shaded red and are the stereocilia that contain active transduction channels. The principal stereocilia-building function in each stage is indicated. The data underlying all the graphs shown in the figure can be found in figshare (https://doi.org/10.6084/m9.figshare.21632636.v2). IHC, inner hair cell.

are 3D models of specific structures computed from stacks of images by preprocessing, segmentation, and connected-component labeling steps. We saved separate surfaces for each row 1, 2, or 3 stereocilium, as well as for some row 4 stereocilia and apical-surface microvilli (referred together as row 4+ stereocilia). Because of their relatively large sizes, rows 1 and 2 stereocilia could be measured reliably; dimensions of rows 3 and 4+ stereocilia and microvilli could also be measured, but required careful choices of rendering parameters and editing of surfaces after rendering.

For row 1 and row 2, the variability in stereocilia length within each IHC hair bundle (represented by the coefficient of variation, CV) was high at P0.5 and declined by P7.5 (S1A Fig). The majority of length variability at intermediate ages during development was because the peripheral stereocilia in a row were systematically shorter than the central stereocilia of that row [6], a phenomenon that disappears by P11 [47] (depicted schematically in S1E–S1G Fig). We therefore restricted most of our measurements to the center 10 stereocilia of each row (panel F of S1 Fig), which could be reliably carried out only with rows 1 and 2. By focusing on these center stereocilia, length variability was reduced significantly in row 1 and row 2 at all ages except P0.5 and was constant across development (S1B Fig). The width CV for each bundle was also significantly lower for row 1 and especially row 2 for the center 10 stereocilia at most ages (S1C and S1D Fig).

## Stereocilia dimensions during stages III and IV of early postnatal development

Tilney [1] suggested that hair-bundle development in chickens could be divided into 4 stages, where stage I hair cells have apical surfaces with equal-length microvilli, stereocilia in stage II cells begin to lengthen and form the staircase, stage III stereocilia widen without lengthening, and stage IV stereocilia undergo final lengthening (Fig 1O). While these stages are present in apical IHCs of C57BL/6 mice [6], the temporal and spatial resolution of previous measurements was insufficient to define transition points between the stages. We improved on those measurements by (1) using the objective measurement approach described above; (2) including a larger number of samples for each time point; and (3) expanding the number of time points examined (Fig 1G–1N). For length and width measurements, as well as individual stereocilium volumes, we calculated the mean and CV for the center 10 stereocilia of each row of a single bundle; each symbol (each time point) in Fig 1G–1N corresponds to data from 22 to 24 single hair cells from 4 different cochleas. Total stereocilia volume and number measurements were made for all stereocilia in a bundle, also with a total of 22 to 24 single-cell measurements from 4 cochleas.

As previously reported [52], the number of IHC stereocilia decreased between P0.5 and P21.5, eventually reaching approximately 17 each of row 1 and row 2 stereocilia (Fig 1L). Row 1 stereocilia length remained constant between P0.5 and P8.5, and then increased exponentially through P21.5, nearly doubling in length over that time period (Fig 1G). The transition between the flat and exponential regimes was sharp. By contrast, row 2 lengths were greatest at P0.5; the length then decreased linearly until P8.5, when the length stabilized (Fig 1G). Row 2 stereocilia widened about 30% faster than row 1 stereocilia (Fig 1H). The difference in width between the rows at each time point was somewhat less than we previously estimated using a different quantitation method [6]. The widening of row 1 over the entire developmental period was fit by an exponential function, while row 2 started with exponential growth and switched to narrowing with an exponential time course. The more sustained widening in row 1 stereocilia meant that they eventually became about 20% wider than those in row 2 (Fig 1H).

If new actin filaments were added to a stereocilium at a constant rate, stereocilium cross-sectional area (but not width) would grow linearly. Indeed, stereocilia cross-sectional area for IHCs increased linearly between P0.5 and P8.5 (row 1) or P7.5 (row 2); after that time, row 1 area increased exponentially in row 1 and decreased exponentially in row 2 (Fig 1J). Again, the transition between the linear and exponential regimes occurred sharply around P8 (Fig 1J).

The volume of each center 10 row 1 or 2 IHC stereocilium increased linearly until P8.5 (row 1) or P7.5 (row 2); row 1 volume increased exponentially afterwards, while row 2 volume decreased exponentially (Fig 1I). To measure total stereocilia volume in a hair bundle, we included all stereocilia from rows 1 to 3; volume increased exponentially between P0.5 and P21.5 (Fig 1K), with no apparent transitions.

We also measured the difference in length between adjacent IHC stereocilia. The difference in length between adjacent rows 1 and 2 stereocilia pairs—those that are in a single column along the tip-link axis—was constant at approximately 1 μm between P0.5 and P7.5, then increased exponentially to over 4 μm by P21.5 (Fig 1M). Unsurprisingly, this growth profile resembled that of row 1 lengthening and was enhanced by the minor row 2 shortening over the same period. Immediately adjacent stereocilia (side-by-side) in a row were very similar in length (Fig 1N). Using the central 10 stereocilia of each row, we measured the absolute value of the difference in length between each pair of adjacent stereocilia, then plotted the mean and CV for each hair bundle. Row 1 and row 2 stereocilia pairs showed a narrow distribution of length differences within a bundle, averaging approximately 0.3 μm (Fig 1N).

## Using mouse mutants to dissect hair-bundle development

To understand the contribution of CDH23 and PCDH15 to hair-bundle development in IHCs, we investigated morphological features of hair bundles in postnatal apical IHCs from $Cdh23^{v2J}$ and $Pcdh15^{av3J}$ mice [36,53–55]. Although these mouse lines are referred to here as "tip-link mutants," they also lack mechanotransduction because of the loss of tip links [37]. We confirmed that PCDH15 immunoreactivity was lost in $Pcdh15^{av3J/av3J}$ mice and CDH23 immunoreactivity was absent in $Cdh23^{v2J/v2J}$ mice (S2 Fig). To disentangle loss of tip links from loss of transduction, we also evaluated $Tmie^{KO}$ mice, which represent the "transduction mutant" class; they phenocopy both the $Tmc1;Tmc2$ double knockout and $Cib2$ knockouts [18,23]. Phenotypes shared by tip-link and transduction mutants most likely result from the loss of transduction, while phenotypes specific for tip-link mutants arise from the absence of the links themselves or of signaling from their intracellular domains. We also analyzed $Myo15a^{sh2}$ mice, which represent the "row 1 tip complex mutant" class; they largely phenocopy $Gpsm2$, $Gnai3$, $Eps8$, and $Whrn$ mutants [7].

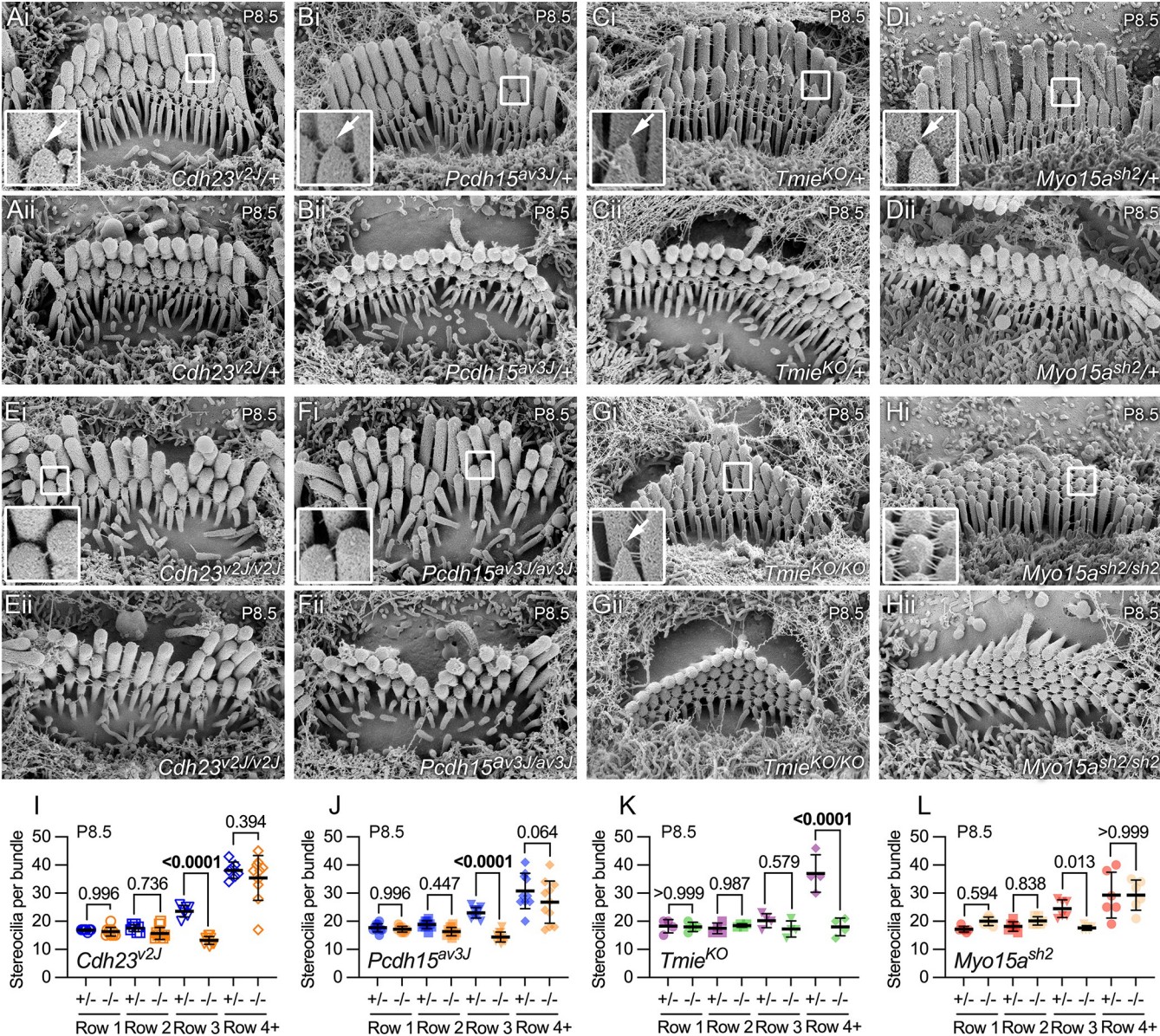

**Fig 2. Scanning electron microscopy of mutants showing links and tip profiles.** (**A–H**) Scanning electron micrographs of single IHC hair bundles from P8.5 cochleas of indicated genotypes. For each genotype, top panel (labeled i) is a profile example and bottom panel (ii) is a top-down example. The insets in Ai, Bi, Ci, Di, and Gi all provide examples of beveled or pointed stereocilia tips; the insets in Ei, Fi, and Hi provide example of rounded tips. Some links were still present in both *Cdh23^(v2J/v2J)* and *Pcdh15^(av3j/av3j)* bundles, but were not associated with beveled stereocilia tips. Panel widths, 6 µm; inserts are 0.5 µm wide and are magnified 3-fold on left. Arrows indicate tip links. (**I–L**) Quantitation of number of stereocilia per hair bundle from scanning electron micrographs. Rows 1, 2, and 3 separately plotted; additional rows of stereocilia and microvilli on each cell's apical surface were also counted (row 4+). *P* values for unpaired *t* tests are indicated. The data underlying all the graphs shown in the figure can be found in figshare (https://doi.org/10.6084/m9.figshare.21632636.v2). IHC, inner hair cell.

We examined hair-bundle structure, stereocilia shape, and presence of interstereocilia links using scanning electron microscopy of apical IHCs at P8.5, at the end of stage III (Fig 2). In heterozygote controls from each of the 4 mouse lines, bundle structure was similar with thick row 1 and 2 stereocilia and much thinner row 3 stereocilia (Fig 2A–2D); moreover, stereocilia length in each row was relatively uniform. By contrast, in tip-link mutants, while distinct rows 1, 2, and 3 usually could be identified, rows were less orderly and stereocilia length in each row

was irregular (Fig 2E and 2F). Rows 1 and 2 stereocilia were still relatively thick, and row 3 stereocilia were generally considerably thicker than those of controls (Fig 2E and 2F). As previously reported, $Tmie^{KO/KO}$ bundles have stereocilia that are more uniform in length and width (Fig 2G).

While tip links were impossible to convincingly distinguish from other links at early developmental ages using scanning electron microscopy [56], total links in hair bundles were abundant in heterozygote controls, as well as $Tmie^{KO/KO}$ and $Myo15a^{sh2/sh2}$ mutants; by contrast, there were qualitatively fewer total links in $Cdh23^{v2J/v2J}$ and $Pcdh15^{av3J/av3J}$ bundles, as reported previously [37]. Beveled row 2 tips were common in heterozygote control bundles and in $Tmie^{KO/KO}$ bundles (Fig 2A–2D and 2G), consistent with tip links causing membrane tenting and altering actin polymerization [6,18]. By contrast, row 2 tips of tip-link mutant stereocilia were rounded (Fig 2E and 2F), presumably because of the loss of tip links.

We examined scanning electron micrographs to determine the number of stereocilia in each row (Fig 2I–2IL). Pooled heterozygotes stereocilia from all 4 lines had 17.4 ± 0.3 row 1 stereocilia, 18.1 ± 0.3 row 2 stereocilia, 23.1 ± 0.5 row 3 stereocilia, and 51 ± 4 total stereocilia (mean ± SEM; $n$ = 27 cells). The values of 17 to 18 rows 1 and 2 stereocilia were very close to those estimated by lattice SIM at P7.5 (Fig 1L).

In IHCs homozygous for any of the 4 mutant alleles, the number of row 1 and row 2 stereocilia did not change significantly relative to heterozygotes (Fig 2I–2L). In tip-link mutants, the number of row 3 stereocilia decreased significantly (Fig 2I and 2J); by contrast, the number of stereocilia in row 3 did not change in $Tmie^{KO/KO}$ hair bundles, although the number of row 4 + stereocilia (stereocilia associated with the bundle beyond row 3 and microvilli on the apical surface) decreased significantly (Fig 2K).

## Increased variability in stereocilia dimensions of tip-link mutants

Although we previously measured stereocilia dimensions in transduction mutants [6], corresponding dimensions in tip-link mutants have yet to be systematically investigated. We therefore determined length, width, and volume of mutant IHC stereocilia from rendered lattice SIM images from P7.5, at the end of stage III, and P21.5, at the end of stage IV (Fig 3A–3H). Stereocilia of tip-link mutants were less well regimented than those in controls, making assignment to rows 1, 2, and 3 more difficult. Proximity to the bare zone along the lateral edge of a hair bundle was used to assign row 1, with neighboring row assignments made medially from row 1. For length and width measurements, we calculated the mean and CV for the central 10 stereocilia of each row; these were our single-cell data points (gray symbols). A cochlea measurement (colored symbols) was derived from mean or CV measurements from 5 to 6 single IHCs.

At P7.5, average lengths of the central 10 stereocilia in rows 1 and 2 changed only modestly between heterozygote control and $Cdh23^{v2J/v2J}$ (Fig 3Ii) or $Pcdh15^{av3J/av3J}$ stereocilia (Fig 3Mi). By contrast, average CV values for length (within each hair bundle) were significantly higher for mutant stereocilia, especially those of row 2 (Fig 3Iii and 3Mii). By P21.5, average lengths were still similar between control and mutant cells (Fig 3Ji and 3Ni); again, CV values were considerably higher for $Cdh23^{v2J/v2J}$ and $Pcdh15^{av3J/av3J}$ mutants, again most notably for row 2 (Fig 3Jii and 3Nii). Thus, $Cdh23^{v2J/v2J}$ and $Pcdh15^{av3J/av3J}$ mutants displayed higher variability of lengths of stereocilia in each IHC for rows 1 and 2 than controls, despite having similar average lengths.

Stereocilia lengths in $Tmie^{KO/KO}$ mutants at P7.5 were also not significantly different from those of controls (Fig 3Ki); by P21.5, however, row 1 stereocilia had lengthened considerably less than those of controls (Fig 3Li). Variability in rows 1 and 2 lengths was significantly

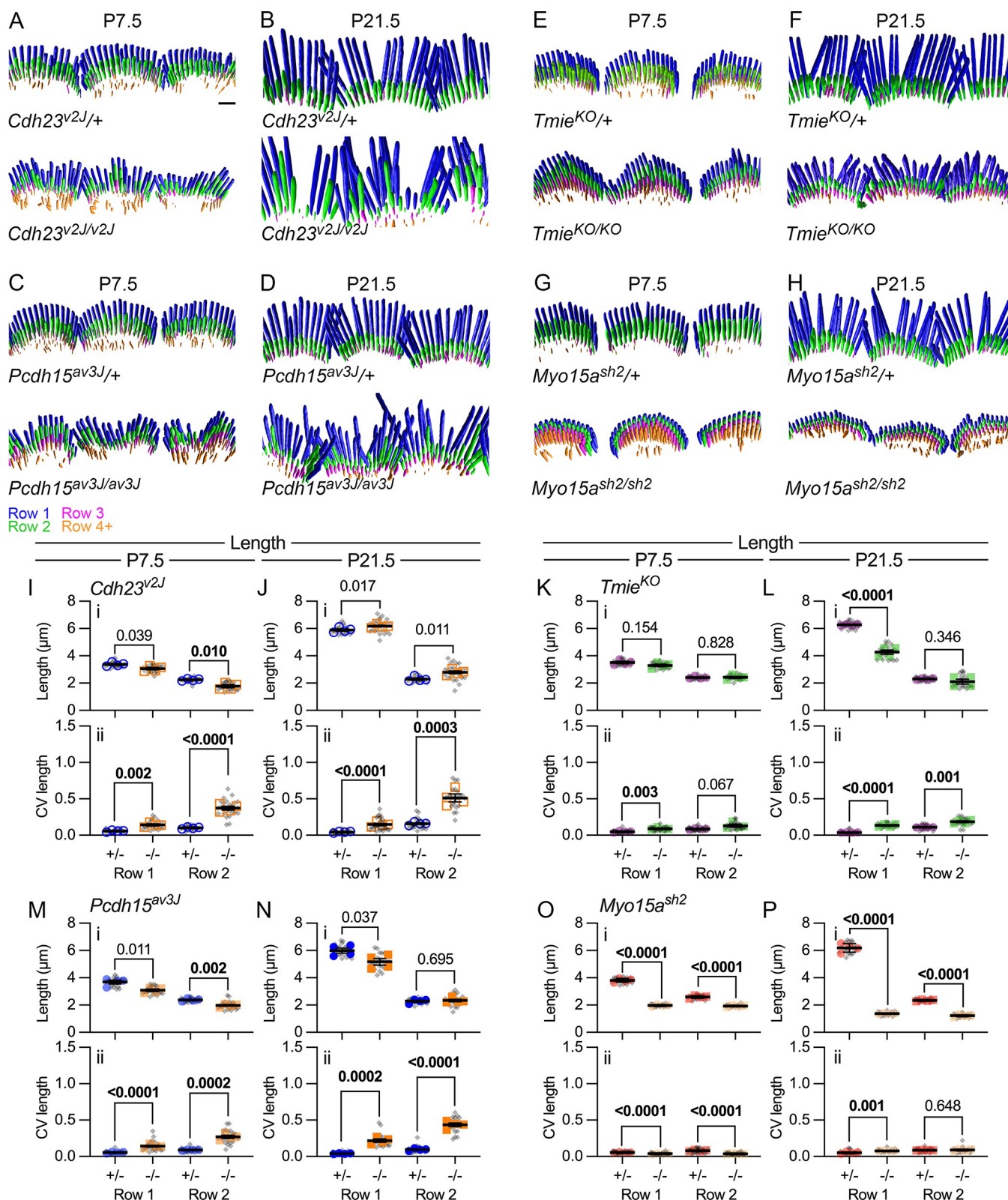

**Fig 3. Stereocilia length in IHCs of mutants.** (**A–H**) Imaris reconstruction of phalloidin-labeled stereocilia from *Cdh23^{v2J}* (A, B), *Pcdh15^{av3J}* (C, D), *Tmie^{KO}* (E, F), and *Myo15a^{sh2}* (G, H) IHCs at P7.5 and P21.5. Stereocilia surfaces are color-coded according to row; scale bar in A (2 μm) applies to all panels. (**I–P**) Average stereocilia length (panels labeled i) and CV (ii). Length measurements used reconstructed stereocilia surfaces. Lengths were determined for the center

10 stereocilia of each hair cell and means for all 10 stereocilia were plotted separately for rows 1 and 2 (gray symbols). The cell means were averaged for each cochlea (5–6 cells per cochlea) and plotted with colored symbols (4 cochleas per condition) [57]. Nested *t* tests were used to compare the cochlea values for each genotype [58]. *P* values are indicated and were bolded if <0.010. (**I, J**) Length and length CV at P7.5 and P21.5 for *Cdh23*$^{v2J}$. (**M, N**) Length and length CV at P7.5 and P21.5 for *Pcdh15*$^{av3J}$. (**K, L**) Length and length CV at P7.5 and P21.5 for *Tmie*$^{KO}$. (**O, P**) Length and length CV at P7.5 and P21.5 for *Myo15a*$^{sh2}$. The data underlying all the graphs shown in the figure can be found in figshare (https://doi.org/10.6084/m9.figshare.21632636.v2). CV, coefficient of variation; IHC, inner hair cell.

greater in *Tmie*$^{KO/KO}$ stereocilia as compared to controls (Fig 3Kii and 3Lii), although CV values were much less than those for tip-link mutants.

Stereocilia widths of the central 10 stereocilia changed in *Pcdh15*$^{av3J/av3J}$, *Cdh23*$^{v2J/v2J}$, and *Tmie*$^{KO/KO}$ mutants; in each case, rows 1 and 2 widths were similar in controls but row 2 was reduced in width in mutants (Fig 4Ai–4Fi). Width CV increased significantly in tip-link mutants, especially in row 2 (Fig 4Aii–4Dii); by contrast, width CV decreased in *Tmie*$^{KO/KO}$ mutants (Fig 4Eii–4Fii). Row 2 width decreased in all 3 mutants, suggesting that transduction promotes widening. However, widths were more variable in the tip-link mutants, suggesting that CDH23-PCDH15 links between adjacent stereocilia might also coordinate widening.

## Adjacent stereocilia length coordination is disrupted in tip-link mutant IHCs

An elevated CV for IHC stereocilia length for tip-link mutants (Fig 3) could arise from systematic variation in length within a row or from stochastic variation within the row. To probe the source of high CV, we used Imaris surfaces to measure the differences in length between adjacent (side-by-side) stereocilia of row 1 or row 2 of *Cdh23*$^{v2J/v2J}$, *Pcdh15*$^{av3J/av3J}$, *Tmie*$^{KO/KO}$, and *Myo15a*$^{sh2/sh2}$ hair bundles. The average difference in length was larger in tip-link mutants than in controls at P7.5 (row 1 in Fig 5Ci and row 2 in Fig 5Ei; compare to Fig 1N) and was particularly large at P21.5 (Fig 5Di and 5Fi). CV values for the length differences were high but similar for control and mutant stereocilia (Fig 5Cii–5Fii). Because the length differences were so small, the high CV values may simply reflect these small values relative to a presumed absolute measurement error of similar magnitude. Altogether, the length-difference results suggest that length variability is stochastic, and indicated that CDH23 and PCDH15 participate in coordination of the lengths of adjacent stereocilia, both for row 1 and for row 2.

*Tmie*$^{KO/KO}$ mutant IHCs showed a different pattern for adjacent stereocilia length differences (Fig 5C–5F). At P7.5, adjacent stereocilia length differences were similar in control and mutant hair bundles, both for adjacent stereocilia in row 1 (Fig 5Ci) or in row 2 (Fig 5Ei). At P21.5, *Tmie*$^{KO/KO}$ mutants showed much greater length variability for row 1 stereocilia (Fig 5Di), but not for row 2 (Fig 5Fi). These results suggest that the lengths of adjacent row 2 stereocilia remain coordinated in *Tmie*$^{KO/KO}$ mutants, which still have tip links, although row 1 coordination is reduced.

The difference in length between row 1 and row 2 stereocilia in a column was approximately the same for tip-link mutants and heterozygotes at P7.5 (Fig 5G) and P21.5 (Fig 5H). By contrast, CV values were much higher for tip-link mutants at both ages (Fig 5Gii and 5Hii). These results suggest that tip links also control the relative difference in length between row 1 and row 2, perhaps indirectly through controlling adjacent stereocilia lengths in the same row.

While *Tmie*$^{KO/KO}$ mutant IHCs had a similar in-column difference in length compared to heterozygotes at P7.5 (Fig 5G), that difference was much less in mutants than heterozygotes at P21.5 (Fig 5Hi), which corresponds to the reduced row 1 lengthening seen in these mutants (Fig 3). This reduced difference in length was accompanied by an increased CV for the difference, perhaps reflecting the reduced coordination among row 1 stereocilia (Fig 5Hii).

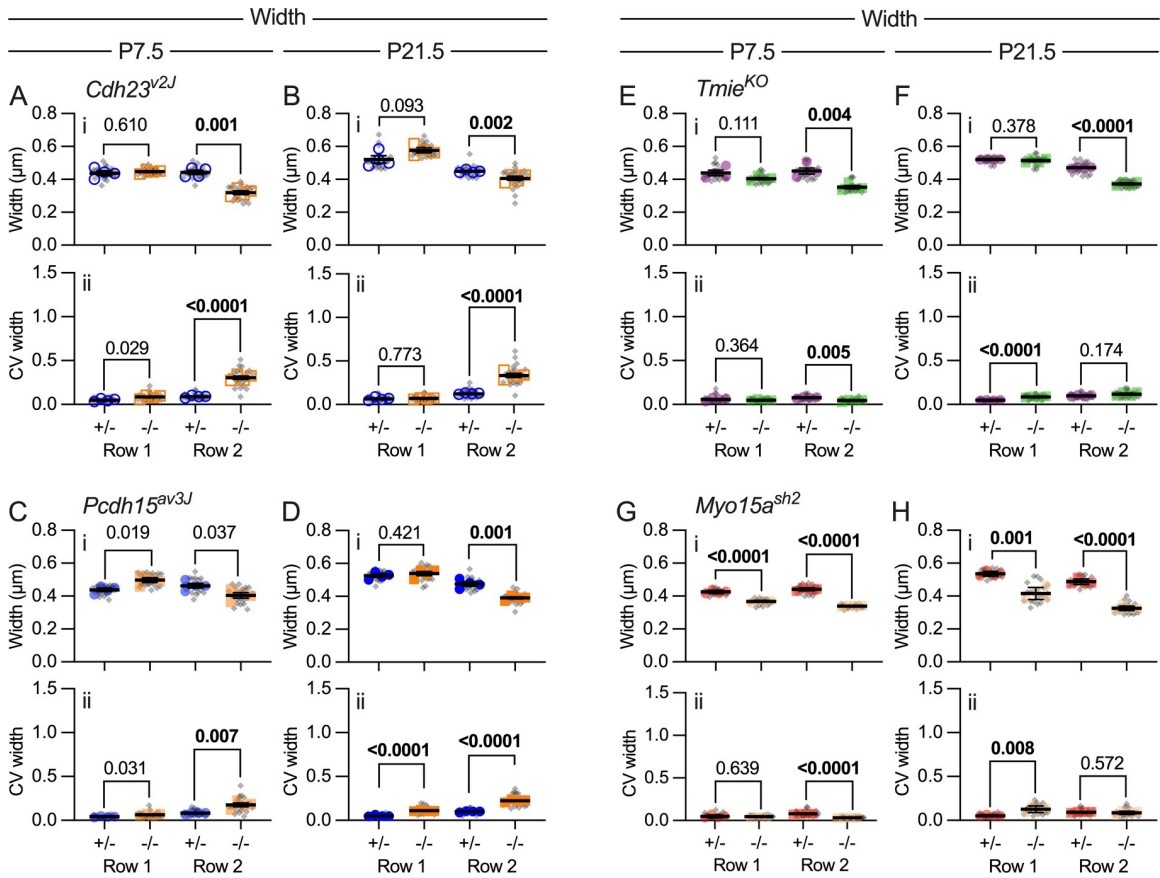

**Fig 4. Stereocilia width in mutants.** Width measurements used reconstructed stereocilia surfaces. (**A–H**) Average stereocilia width (panels labeled i) and CV (ii) were determined for the center 10 stereocilia of each hair cell (gray symbols), separately for rows 1 and 2. Plotting and statistical testing were as in Fig 3. (**A, B**) Width and width CV at P7.5 and P21.5 for *Cdh23*^v2J^. (**C, D**) Width and width CV at P7.5 and P21.5 for *Pcdh15*^av3J^. (**E, F**) Width and width CV at P7.5 and P21.5 for *Tmie*^KO^. (**G, H**) Width and width CV at P7.5 and P21.5 for *Myo15a*^sh2^. The data underlying all the graphs shown in the figure can be found in figshare (https://doi.org/10.6084/m9.figshare.21632636.v2). CV, coefficient of variation.

### *Myo15a* interacts genetically with *Pcdh15*

As previously reported, stereocilia of *Myo15a*^sh2/sh2^ IHC hair bundles were very short and were more uniform in length and width than *Myo15a*^sh2^/+ stereocilia (Fig 2G and 2H). Although they still support mechanotransduction [12], tips of *Myo15a*^sh2/sh2^ IHCs were rounded (Fig 2H), perhaps because the minimal staircase prevented tip links from sufficiently tenting membranes off the tips of the shorter rows. The number of stereocilia in each row of *Myo15a*^sh2/sh2^ bundles did not change appreciably (Fig 2L). As has been well documented [59], rows 1 and 2 stereocilia lengths and widths were significantly reduced for *Myo15a*^sh2/sh2^ mutants (Figs 3Oi, 3Pi, 4Gi and 4Hi), associated with lower CV values (Figs 3Oii, 3Pii and 4Gii–Hii). The adjacent stereocilia length differences for *Myo15a*^sh2/sh2^ at P7.5 (Fig 5C and 5E) and P21.5 (Fig 5D and 5F) were considerably less than those of heterozygote controls. Likewise, the in-column difference in length for *Myo15a*^sh2/sh2^ mutants was substantially lower than controls at both ages (Fig 5Gi and 5Hi); the CV for difference in length was increased substantially (Fig 5Gii and 5Hii), which may reflect errors in measuring these very short stereocilia.

We created double-knockout *Pcdh15*^av3J/av3J^;*Myo15a*^sh2/sh2^ (*Pcdh15;Myo15a* DKO) mice to test for a genetic interaction between *Pcdh15* and *Myo15a* (Fig 6). In *Pcdh15;Myo15a* DKO

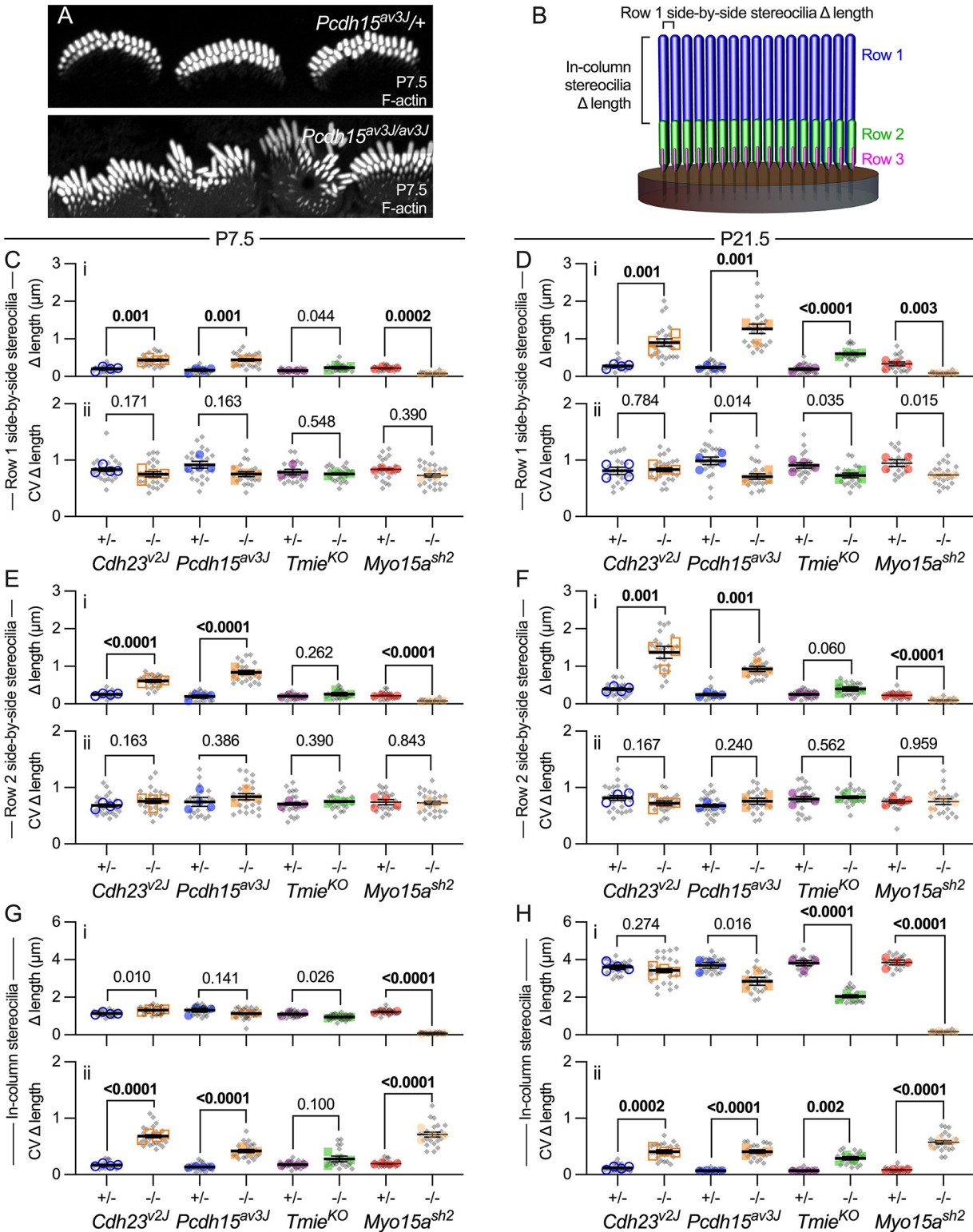

**Fig 5. Adjacent stereocilia length coordination in mutants.** (**A**) Phalloidin-labeled stereocilia from P7.5 *Pcdh15^{av3J}* IHCs. Adjacent stereocilia in homozygous mutant hair bundles are irregular in length. Panel widths: 30 μm. (**B**) Schematic showing measurements made. For each pair of adjacent stereocilia in row 1 and, separately, in row 2, we measured the difference in length between them (side-by-side stereocilia Δ length). For example, Δ length for the pair indicated in panel B would be |length$_1$ −length$_2$|, or the absolute value of the difference in lengths. In addition, for each row 1 −row 2 pair in a column, we measured the difference in length between the 2 stereocilia (in-column stereocilia Δ length). The

brackets in panel B mark this length difference for the first row 1-row 2 stereocilia pair. (C–H) Difference in adjacent stereocilia length for *Cdh23^{v2J}*, *Pcdh15^{av3J}*, *Tmie^{KO}*, and *Myo15a^{sh2}* mice using reconstructed stereocilia surfaces. Plotting and statistical testing were as in Fig 3. (C, D) Difference in length (Δ length) of side-by-side row 1 stereocilia at P7.5 and P21.5. (E, F) Δ length of side-by-side row 2 stereocilia at P7.5 and P21.5. (G, H) Δ length of in-column (adjacent rows 1 and 2) stereocilia at P7.5 and P21.5. The data underlying all the graphs shown in the figure can be found in figshare (https://doi.org/10.6084/m9.figshare.21632636.v2). IHC, inner hair cell.

IHC hair bundles, stereocilia were spread across the apical surface and were packed roughly hexagonally (Fig 6C and 6F), albeit not in a precise array as in *Pcdh15^{av3J}*/+;*Myo15a^{sh2/sh2}* (*shaker 2* KO; Fig 6B and 6E). Links near stereocilia tips were present not only in *shaker 2* KO bundles (Fig 6B), as expected, but also in *Pcdh15;Myo15a* DKO bundles (Fig 6C). As in *Pcdh15^{av3J/av3J}* bundles (Fig 2F), remaining ankle links and CDH23-CDH23 links may maintain stereocilia interactions in *Pcdh15;Myo15a* DKO bundles.

Lengths and widths of rows 1 and 2 *Pcdh15;Myo15a* DKO stereocilia were similar to those of *shaker 2* KO stereocilia (Fig 6G and 6I), although row 2 stereocilia were significantly shorter in the DKO bundles as compared to *sh2* KO bundles. *Pcdh15;Myo15a* DKO stereocilia showed a significant lateral-to-medial gradient in length, which was less apparent in *shaker 2* KO bundles. Row 2 stereocilia showed more length and width variation in DKO as compared to *sh2* KO bundles (Fig 6H and 6J), but the differences were not large.

The difference in length between adjacent stereocilia was similar in DKO hair bundles as compared to *sh2* KO bundles (Fig 6K), with no significant differences in CV (Fig 6L). Likewise, the difference in length between rows 1 and 2 stereocilia in a single column was no different in DKO bundles than in *sh2* KO bundles (Fig 6M). These results suggest that row 1 complex proteins are required to elongate stereocilia before PCDH15-dependent links can help refine bundle architecture.

## Disrupted pruning of stereocilia and microvilli in mutants

Mutations in tip-link genes also impacted the pruning of short rows of IHC stereocilia. In control hair cells, row 3 stereocilia were obvious at P7.5 (Fig 7A–7D) but were much harder to detect by P21.5 (Fig 7E–7H). Stereocilia and microvilli beyond row 3 (row 4+) were present at P7.5 (Fig 7A–7D and 7K) but had nearly completely disappeared by P21.5 (Fig 7E–7H and 7L). By contrast, not only did the number of stereocilia in rows 3 and 4+ increase in mutants (Fig 2I–7L), but their length and diameter—and thus volume—also increased in mutants (Fig 7). Rows 3 and 4+ stereocilia lengths and widths, measured using Imaris surface reconstructions, were quite variable since the phalloidin signal in these stereocilia in controls was very low and therefore challenging to render. Instead, we measured surface volumes, which incorporated both length and width, as they better reflected the overall F-actin amount in each stereocilium and were more consistently measured from control IHCs (Fig 7I–7L).

In tip-link mutants, the volume of each row 3 stereocilium increased significantly at P7.5 compared to controls and that translated into a significant increase in total volume in all stereocilia of row 3 (Fig 7A, 7B and 7I). This increase in individual and total volume for row 3 stereocilia was maintained at P21.5 for *Pcdh15^{av3J/av3J}* mutant hair bundles, but not for *Cdh23^{v2J/v2J}* mutant bundles (Fig 7E, 7F and 7J). This measurement was one of very few that distinguished the *Cdh23^{v2J/v2J}* and *Pcdh15^{av3J/av3J}* mutant hair cells. Row 4+ individual and total stereocilia volume also increased significantly in tip-link mutant IHCs at P7.5 (Fig 7K), but was less striking at P21.5 (Fig 7L).

Reduced pruning and increased volume of rows 3 and 4+ stereocilia were much more apparent in *Tmie^{KO/KO}* and *Myo15a^{sh2/sh2}* hair bundles (Figs 2K, 2L and 7). As previously reported, *Tmie^{KO/KO}* bundles had a prominent row 3 and often additional rows (Fig 7C and 7G); *Myo15a^{sh2/sh2}* bundles had 4 to 6 rows of stereocilia (Fig 7D and 7H). At P7.5, individual

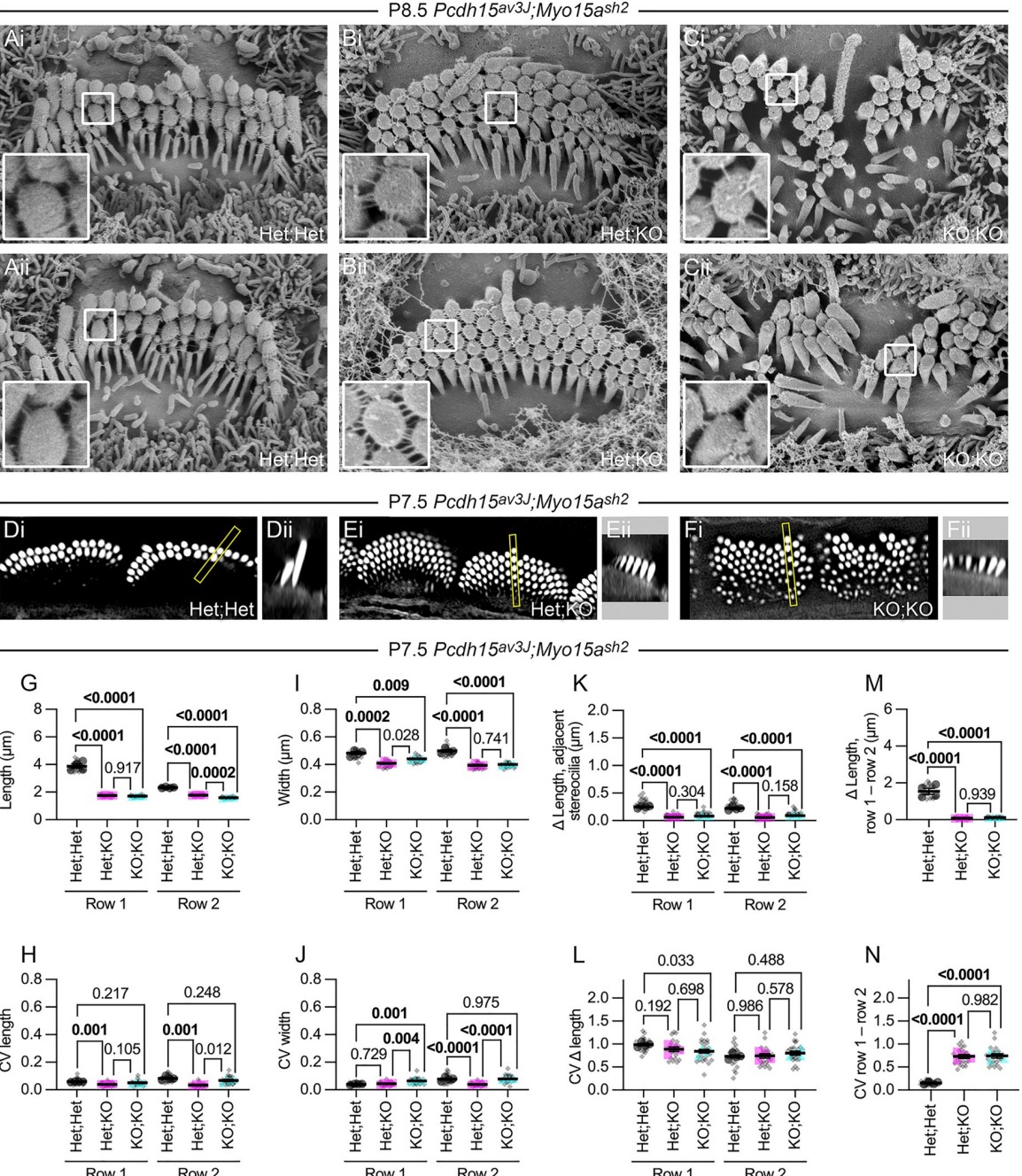

**Fig 6. *Pcdh15^{av3J};Myo15a^{sh2}* double-knockout mice.** (**A–C**) Scanning electron micrographs of single IHC hair bundles from P8.5 cochleas of indicated *Pcdh15^{av3J}/+;Myo15a^{sh2}/+* (Het;Het), *Pcdh15^{av3J}/+;Myo15a^{sh2/sh2}* (Het;KO), and *Pcdh15^{av3J/av3J}; Myo15a^{sh2/sh2}* (KO; KO) cochleas (2 examples per genotype). Panel widths, 6 μm. Inserts are 0.5 μm wide and are magnified 3-fold on left; arrows indicate tip links. (**D–F**) Phalloidin-labeled IHC hair bundles from Het;Het, Het;KO, and KO;KO cochleas. Panels labeled i (20 μm wide) are single x-y planes from image stacks; panels labeled ii (same scale as i panels) are x-z reslices from those stacks, with the region of the stack indicated with a yellow box in i. (**G–N**) Dimension measurements using reconstructed stereocilia surfaces. Plotting was as in Fig 3; statistical tests used one-way ANOVA analysis with the Šidák correction. (**G, H**) Length and CV length measurements. (**I, J**) Width and width CV measurements. (**K, L**) Difference in length (Δ length) and CV for Δ length of side-by-side row 1 stereocilia. (**M, N**) Difference in length (Δ length) and CV for Δ length of in-column (adjacent rows 1 and 2) stereocilia. The data underlying all the graphs shown in the figure can be found in figshare (https://doi.org/10.6084/m9.figshare.21632636.v2). CV, coefficient of variation; IHC, inner hair cell.

and summed volumes of row 3 (Fig 7Ii–7ii) and row 4+ (Fig 7Ki–7ii) $Tmie^{KO/KO}$ and $Myo15a^{sh2/sh2}$ stereocilia were substantially and significantly increased over controls; this effect remained at P21.5, although more dramatically for $Myo15a^{sh2/sh2}$ than $Tmie^{KO/KO}$ bundles (Fig 7Ji–7ii and 7Li–7ii). These results showed that pruning of short rows of stereocilia was altered in all mutant bundles.

## Total hair-bundle volume does not change in mutants

Total stereocilia volume per hair cell was not significantly different between $Cdh23^{v2J}$/+, $Pcdh15^{av3J}$/+, $Tmie^{KO}$/+ heterozygotes and C57BL/6 wild-type mice at P7.5 or at P21.5 (Fig 8). Despite the variability in stereocilia number, length, and width, total stereocilia volumes in $Cdh23^{v2J/v2J}$, $Pcdh15^{av3J/av3J}$, and $Tmie^{KO/KO}$ IHC hair bundles were statistically indistinguishable at P7.5 (Fig 8A). At P21.5, differences between $Cdh23^{v2J}$ and $Tmie^{KO}$ controls and mutants were modest, albeit statistically significant (Fig 8B). The similarity in stereocilia volume is broadly consistent with Tilney's proposal that each hair cell has a constant amount of F-actin in its bundle, but cells deploy it differently depending on other factors [60]. Although $Myo15a$ was an outlier at P21.5 (Fig 8B), actin and actin crosslinkers intended for stereocilia may have been directed instead to cytocauds in the cell body [61].

## Localization of row 1 tip complex proteins in control and mutant IHCs

Because proteins that localize to row 1 tips control stereocilia length, we examined in greater detail the appearance of EPS8, GNAI3, and GPSM2 at the tips of C57BL/6 stereocilia (Fig 9) and at mutant stereocilia tips (Figs 10–12). EPS8 increased at rows 1 and 2 tips during stage III, then showed an initial decrease and subsequent increase during stage IV (Fig 9I). GNAI3 increased throughout stage III; it peaked at P10.5, soon after stage IV begins (Fig 9J). By contrast, GPSM2 increased modestly at the end of stage III but shot up substantially during late stage IV, peaking at P15.5 (Fig 9K). GNAI3 and GPSM2 levels each dropped precipitously by P21.5 (Fig 9J and 9K). Our results showing sequential accumulation of EPS8, then GNAI3, and finally GPSM2 at stereocilia tips are not consistent with the earlier suggestion that an extended MYO15A-EPS8-WHRN-GPSM2-GNAI3 complex assembles near the stereocilia base and then moves to row 1 tips [7].

As in C57BL/6 stereocilia, in heterozygotes from mutant mouse lines, GNAI3, GPSM2, and EPS8 were concentrated at tips of row 1 stereocilia from IHCs (Fig 10A–10D and 10I–10L). WHRN was also concentrated at row 1 tips in heterozygotes (Fig 10E–10H). In $Pcdh15^{av3J/av3J}$ IHCs, GNAI3 and GPSM2 remained concentrated at row 1 but EPS8 was seen at all stereocilia tips in similar amounts (Fig 10B, 10F and 10J). These results contrasted with those from $Tmie^{KO/KO}$ IHCs (Fig 10D and 10L), where GPSM2 and GNAI3 never reached levels seen in $Tmie^{KO}$/+ controls (Fig 10C and 10K); WHRN intensity was reduced, but remained most abundant in row 1 (Fig 10H). EPS8 levels in shorter rows of $Tmie^{KO/KO}$ IHCs increased substantially [6], but EPS8 was still most abundant in row 1 (Fig 10D, 10H and 10L).

Quantitation of row 1 tip complex proteins allowed us to determine the distribution of these proteins between rows 1 and 2 stereocilia more thoroughly (Figs 11 and 12 and S4). Total EPS8 at stereocilia tips was not altered in tip-link mutants at P7.5 or P21.5 (Fig 11A and 11B); EPS8 decreased, albeit not significantly, in $Tmie^{KO/KO}$ hair bundles. Here, we used fluorescence intensity normalized to heterozygote levels for each age, which reduced signal variability but prevented us from comparing total fluorescence at different ages.

Frequency distribution plots, using normalized EPS8 intensities (signal at each tip divided by the total signal for all rows 1 and 2 stereocilia measured in the hair bundle), revealed how EPS8 was uniformly targeted to stereocilia tips in tip-link mutants. Fitting each frequency

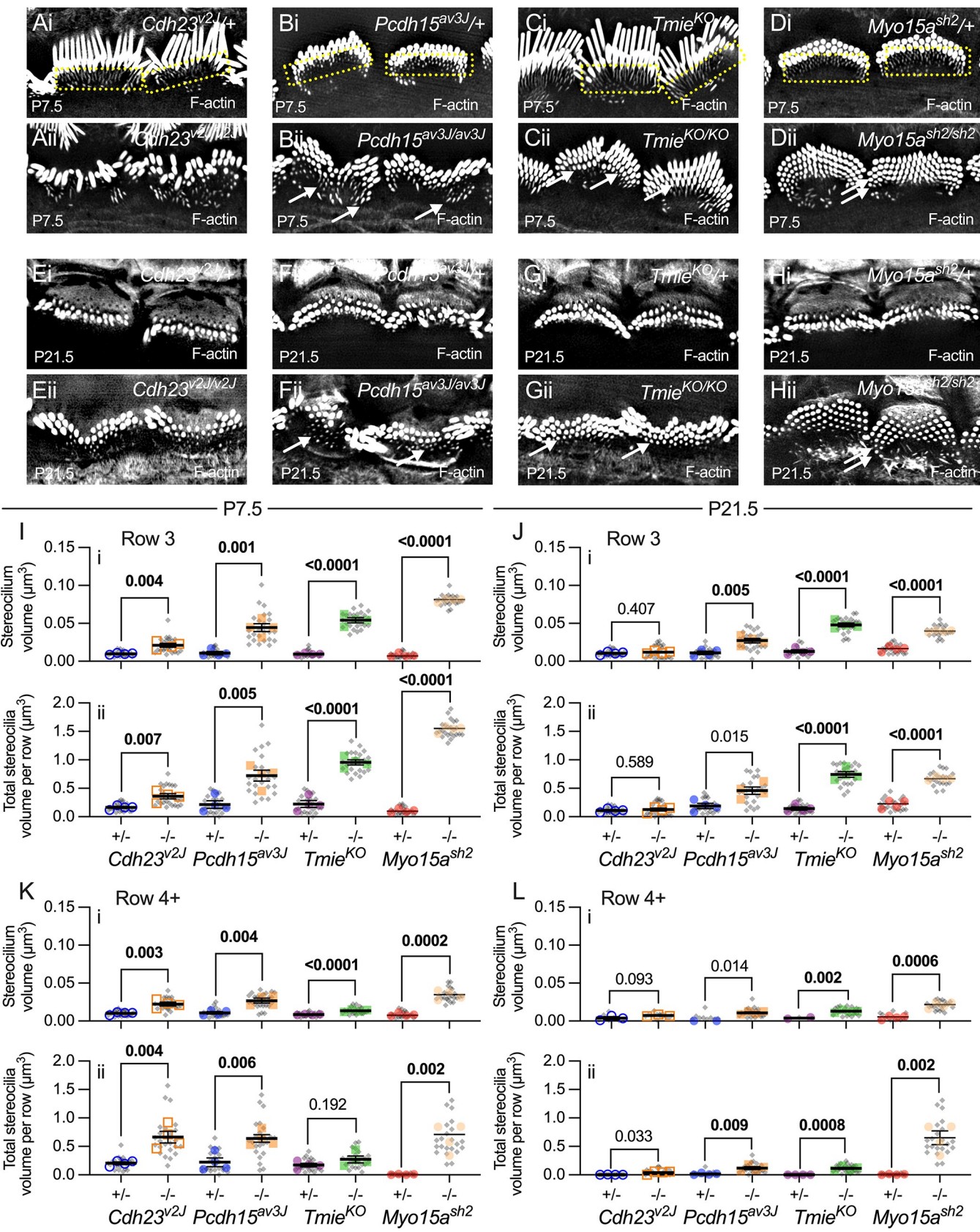

**Fig 7. Pruning of short stereocilia and microvilli in mutant IHCs.** (**A–H**) Phalloidin labeling of P7.5 (A–D) and P21.5 (E–H) IHC hair bundles (i, heterozygotes; ii, homozygotes). Yellow boxes in heterozygotes highlight rows 3–4 stereocilia (low signal intensity because they are short and narrow). Arrows indicate extra rows of thicker stereocilia. Panel widths: 20 μm. (**I–L**) Individual stereocilium volumes (panels labeled i) and total volumes per cell for all stereocilia in a row (ii). Row 3 (I, J) and row 4+ (K, L) are separately plotted. Volume measurements used reconstructed stereocilia surfaces. Plotting and statistical testing were as in Fig 3. The data underlying all the graphs shown in the figure can be found in figshare (https://doi.org/10.6084/m9.figshare.21632636.v2). IHC, inner hair cell.

distribution with a double Gaussian function, 2 prominent peaks were detected in $Cdh23^{v2J}/$ + or $Pcdh15^{av3J}/+$ controls at P7.5, consistent with high levels of EPS8 located at row 1 tips and lower levels present at row 2 (blue data in Fig 11D and 11F). Two peaks were detected at P21.5 (Fig 11F and 11J); one was large in magnitude (EPS8 at row 1), but the other was small (little EPS8 at row 2 tips). By contrast, P7.5 and P21.5 $Cdh23^{v2J/v2J}$ and $Pcdh15^{av3J/av3J}$ mutants each displayed a single distribution, centered at a tip intensity/average = 1 (orange data in Fig 11D, 11F, 11H and 11J). These results confirmed the relative uniformity of EPS8 at all stereocilia tips in tip-link mutants.

We also plotted the distribution of rows 1 and 2 stereocilia lengths for each dataset; in homozygous mutants, the length distributions were biased towards shorter stereocilia lengths and were broadened, especially at P21.5 (Fig 11C, 11E, 11G and 11I). These results also show that a consequence of the increased variability in rows 1 and 2 lengths in the tip-link mutants is an increase in stereocilia of intermediate lengths (Fig 11C, 11E, 11G and 11I), in between average lengths for rows 1 and 2 in the heterozygote controls. Thus, despite their lack of length coordination within rows, hair bundles of tip-link mutants as a whole contain stereocilia of more uniform lengths than those of controls.

In $Tmie^{KO/KO}$ hair bundles at P7.5, the EPS8 frequency distribution showed a broad double peak, unlike the 2 well separated peaks in the $Tmie^{KO}/+$ control (Fig 11L; see also S5J Fig); the distribution was weighted more towards high EPS8 tip/average values than in the tip-link mutant plots. At P7.5, $Tmie^{KO/KO}$ showed a broad distribution of stereocilia length that was still fit with 2 Gaussians (Fig 11K). EPS8 distribution and stereocilia length for $Tmie^{KO/KO}$ at P21.5 were broadly similar to those of $Cdh23^{v2J/v2J}$ and $Pcdh15^{av3J/av3J}$. These results highlight the effect of loss of CDH23 or PCDH15 on EPS8; while tip-link mutants have lost

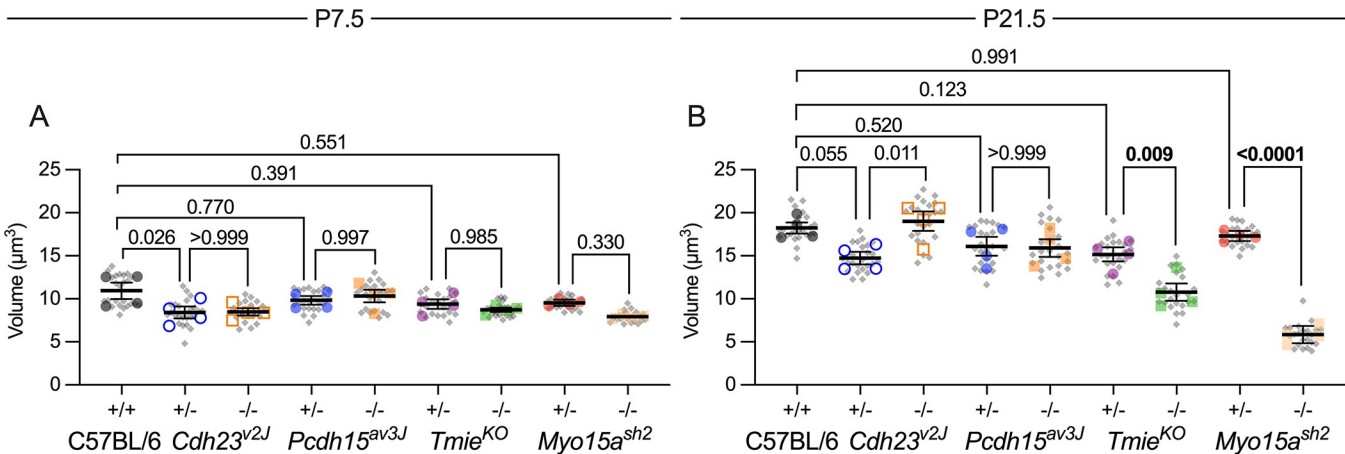

**Fig 8. Total stereocilia volume in mutant IHCs.** (**A, B**) Total stereocilia volume per hair bundle and volume CV for $Cdh23^{v2J}$, $Pcdh15^{av3J}$, $Tmie^{KO}$, and $Myo15a^{sh2}$ IHCs. Volume measurements used reconstructed stereocilia surfaces. Plotting was as in Fig 3; statistical tests used one-way ANOVA analysis with the Šidák correction. (**A**) P7.5. No significant differences in volume between any of the genotypes. (**B**) P21.5. Modest differences in total stereocilia volume were seen for $Cdh23^{v2J}/+$ vs. $Cdh23^{v2J/v2J}$ and $Tmie^{KO}/+$ vs. $Tmie^{KO/KO}$; a large difference in volume was seen for $Myo15a^{sh2}$ IHCs. The data underlying all the graphs shown in the figure can be found in figshare (https://doi.org/10.6084/m9.figshare.21632636.v2). CV, coefficient of variation; IHC, inner hair cell.

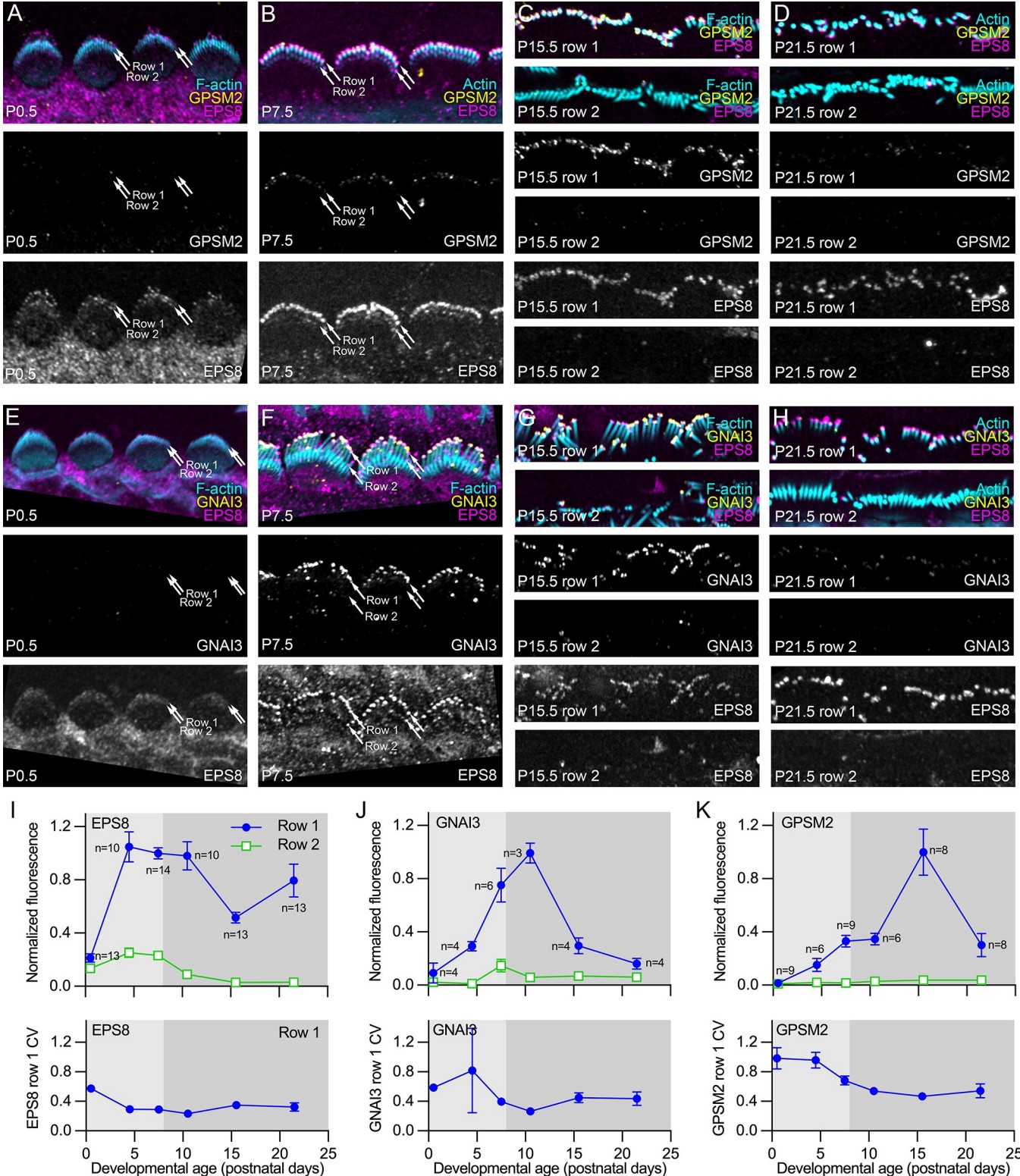

**Fig 9. Developmental expression of row 1 complex proteins in IHC stereocilia. (A–D)** Localization of GPSM2 and EPS8 in P0.5, P7.5, P15.5, and P21.5 IHCs. **(E, F)** Localization of GNAI3 and EPS8 in P0.5, P7.5, P15.5, and P21.5 IHCs. In (A, B) and (E, F), images are projections of horizontal slices with partially flattened hair bundles. In (C, D) and (G, H), images are projections of horizontal slices obtained at the level of row 1 tips only (top) or row 2 tips and row shafts (bottom). Images in A–D were acquired in the same imaging session and used the same acquisition, processing, and display parameters; the same holds for images in E–H. Arrows in A–F indicate level of rows 1 and 2. Panel widths: A–H, 30 μm. **(I-K)** Quantitation of EPS8, GNAI3, and GPSM2

immunofluorescence at rows 1 and 2 tips. Fluorescence was normalized to fluorescence at P7.5 (EPS8 and GNAI3) or P15.5 (GPSM2) to allow comparison between multiple experiments. The data underlying all the graphs shown in the figure can be found in figshare (https://doi.org/10.6084/m9.figshare.21632636.v2). IHC, inner hair cell.

transduction, as have $Tmie^{KO/KO}$ bundles, the impact on EPS8 distribution in the tip-link mutants is somewhat greater.

We also compared the EPS8 intensity at each tip with the length of that stereocilium at P15.5 (Fig 11O–11R). We found that these data were fit well with an exponential, showing that while the length of shorter stereocilia was correlated with EPS8 intensity, beyond a certain EPS8 intensity, stereocilia length no longer depended on EPS8 intensity. Loss of PCDH15 caused a broadening of this curve, although the trend was still the same. Loss of TMIE did not change the shape of the curve but instead lowered the plateau, reflecting the reduced stereocilia

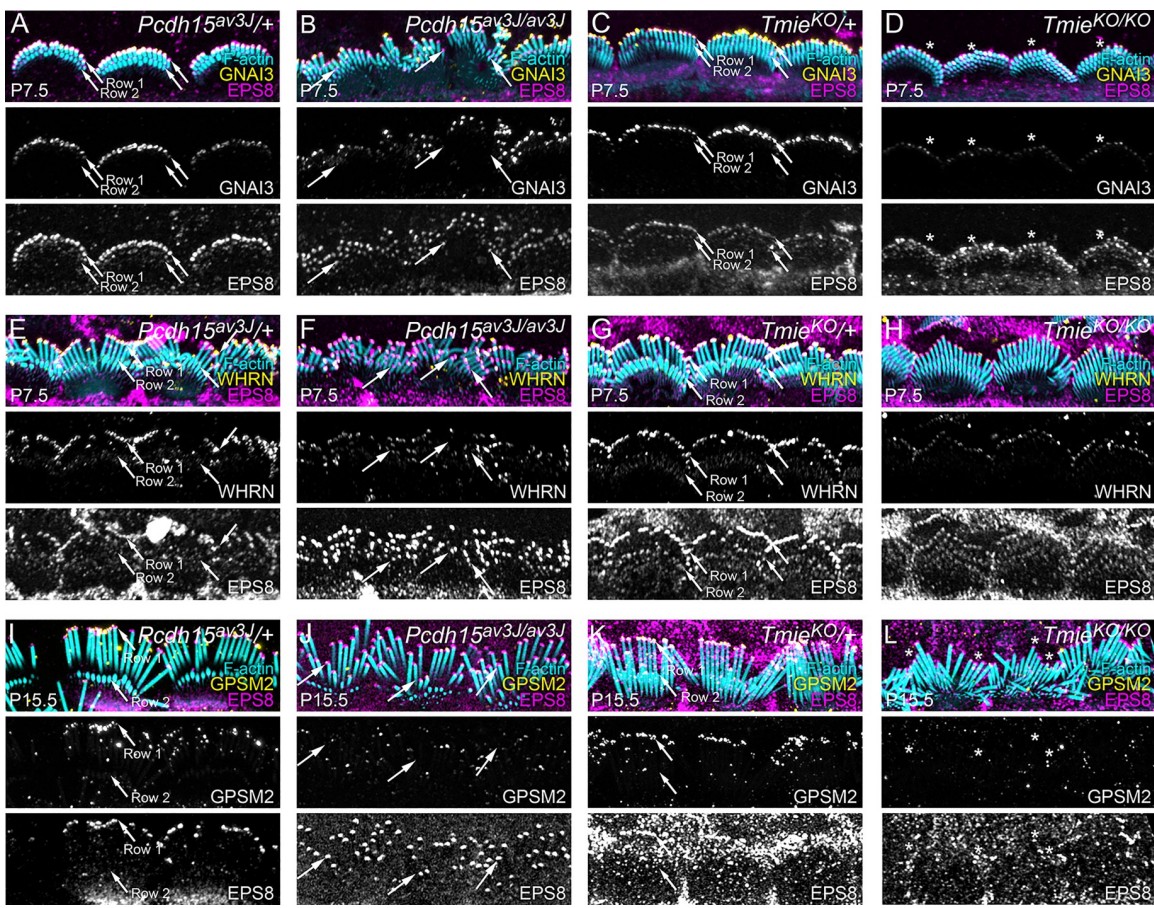

**Fig 10. Row 1 protein localization in mutant IHCs.** (**A–D**) Localization of GNAI3 and EPS8 in $Pcdh15^{av3J}$ and $Tmie^{KO}$ heterozygotes and homozygotes at P7.5. (**E–H**) Localization of WHRN and EPS8 in $Pcdh15^{av3J}$ and $Tmie^{KO}$ heterozygotes and homozygotes at P7.5. (**I–L**) Localization of GPSM2 and EPS8 in $Pcdh15^{av3J}$ and $Tmie^{KO}$ heterozygotes and homozygotes at P15.5. Images from all pairs of heterozygote and homozygote cochleas (A and B, C and D, E and F, G and H, I and J, K and L) were acquired in the same session and used the same acquisition, processing, and display parameters. All images are projections of horizontal slices; projections that allow visualization of the entire hair bundle typically had stereocilia splayed against the apical surface, especially at P15.5, which resulted in immunofluorescence signal from stereocilia being overlaid on top of immunofluorescence signal from apical surfaces. Small arrows in A, C, G, I, and K indicate positions of rows 1 and 2. Large arrows in B, F, and J show high levels of EPS8 in row 2 stereocilia of $Pcdh15^{av3J/av3J}$ IHCs. Asterisks in D and L show reduced levels of GNAI3 and GPSM2 in $Tmie^{KO/KO}$ IHCs. Panel widths: 30 μm. IHC, inner hair cell.

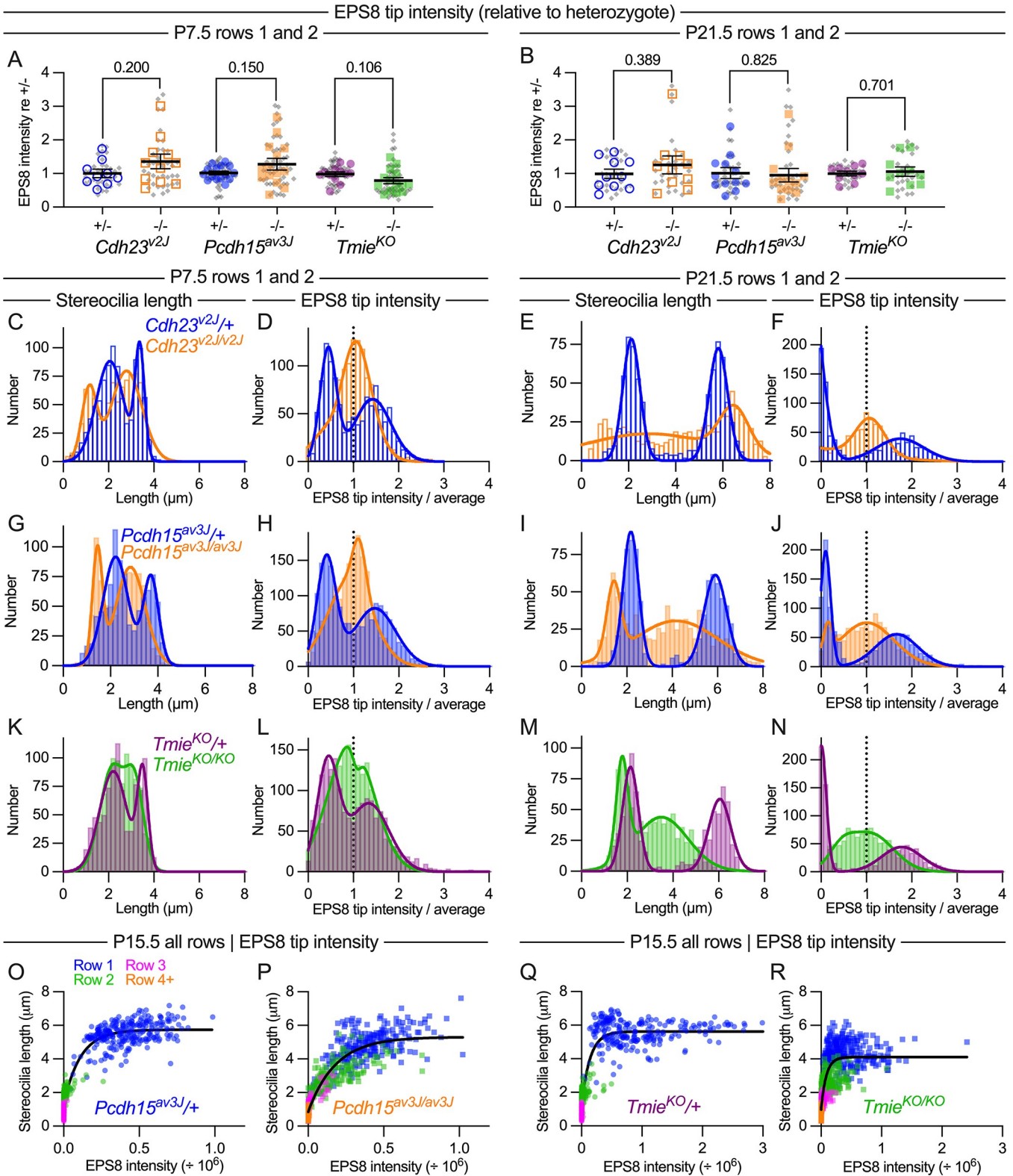

**Fig 11. Distribution of EPS8 at tips of stereocilia in tip-link and transduction mutant IHCs.** (**A, B**) EPS8 fluorescence average intensity per hair bundle for all measured stereocilia (rows 1 and 2) for *Cdh23^{v2J}*, *Pcdh15^{av3J}*, and *Tmie^{KO}* hair cells at P7.5 (A) and P21.5 (B). Plotting and statistical testing were as in Fig 3. (**C–F**) Frequency distribution of stereocilia length and EPS8 tip intensity in *Cdh23^{v2J}* hair cells. (**G–J**) Frequency distribution of stereocilia length and EPS8 tip intensity in *Pcdh15^{av3J}* hair cells. (**K–N**) Frequency distribution of stereocilia length and EPS8 tip intensity in *Tmie^{KO}* hair cells. Stereocilia length

measurements (C, E, G, I, K, M) and EPS8 tip intensities (D, F, H, J, L, N) were from all rows 1 and 2 stereocilia; distributions were fit with double Gaussian functions. (**O–R**) Correlation of normalized EPS8 intensity with stereocilia length. Data from individual rows were displayed with separate symbol colors but all data were used for fits; data were fit with $y = a + b \cdot \exp(-k \cdot x)$, where $y$ is stereocilia length, $x$ is EPS intensity, and $a$, $b$, and $k$ are constants. The data underlying all the graphs shown in the figure can be found in figshare (https://doi.org/10.6084/m9.figshare.21632636.v2). IHC, inner hair cell.

lengthening in these mutants. While more stereocilia had intermediate EPS8 intensity in both mutants, stereocilia belonging to different rows were scattered along the curve in the $Pcdh15^{av3J/av3J}$ mutants, reflecting the loss of row organization in these mutants. These results suggested that EPS8 could influence stereocilia length, but only up to a certain length; this maximal elongation may be regulated by mechanisms in addition to EPS8 levels.

Total GNAI3, GPSM2, and WHRN fluorescence at stereocilia tips was not affected in tip-link mutants (Figs 10A, 10B, 10E, 10F, 10I, 10J, 12A–12D and S4A–S4D). In $Tmie^{KO/KO}$ mutants, as previously reported [6], GNAI3, GPSM2, and WHRN fluorescence at stereocilia tips was considerably reduced (Fig 10C, 10D, 10G, 10H, 10K and 10L), with statistically significant reductions for GNAI3 and WHRN at P7.5 (Fig 12A and 12B) and GPSM2 at P15.5 (Fig 12C). These results showed that TMIE (and TMC1/2) affected distribution of GNAI3, GPSM2, and WHRN directly, not via mechanotransduction-mediated ion flux. TMC2 still targeted to stereocilia in $Pcdh15^{av3J/av3J}$ mutants, but TMC1 disappeared (S3 Fig), suggesting that TMC2 may be sufficient for substantial accumulation of GPSM2, GNAI3, and WHRN to row 1 tips.

In control hair bundles at P7.5, GNAI3 frequency distributions were very different from those of EPS8; most stereocilia had little or no GNAI3, but a minority had a considerable amount (blue in Figs 12E and S4Q). This distribution matched the immunocytochemistry images, where GNAI3 was found nearly exclusively at row 1 tips (Figs 10 and S4). In $Cdh23^{v2J/v2J}$ and $Pcdh15^{av3J/av3J}$ mutants, the frequency distribution was moderately altered; most stereocilia still had little or no GNAI3, but relatively larger numbers had intermediate levels of signal (orange in Figs 12E and S4Q). Even though GNAI3 levels dropped significantly in $Tmie^{KO/KO}$ bundles (Fig 12A), the frequency distribution resembled that of tip-link mutants (green in Fig 12I). The distribution of WHRN at P7.5 was similar to but broader than that of GNAI3, and was altered similarly in tip-link and transduction mutants (Figs 12B, 12F, 12J, S4N and S4R). At P21.5, distribution of GNAI3 and WHRN among stereocilia tips was not altered substantially in tip-link mutants (S4O, S4P, S4S and S4T Fig).

Total GPSM2 intensity was decreased significantly in stereocilia tips of P15.5 $Tmie^{KO/KO}$ mutants, although EPS8 was not (Fig 12C and 12D). The distribution of GPSM2 at P15.5 in heterozygote stereocilia (Fig 12G and 12K) was similar to that of GNAI3 at P7.5 (Fig 12E and 12I), and the altered distribution in $Pcdh15^{av3J/av3J}$ and $Tmie^{KO/KO}$ mutants also resembled that of GNAI3 (Fig 12G and 12K).

Plots of stereocilia length against GPSM2 intensity (Fig 12M–12P) were fit well with an exponential relationship for both controls and mutants, similar to that seen with EPS8 (Fig 11O–11R). The exponential constant $k$ was considerably smaller for the GPSM2 fits ($k \approx 3$) than for the EPS8 fits ($k \approx 8$), consistent with GPSM2 tip accumulation only occurring in the longest stereocilia (i.e., row 1).

### *Tmie* does not interact genetically with *Pcdh15*

We created double-knockout $Pcdh15^{av3J/av3J};Tmie^{KO/KO}$ (*Pcdh15;Tmie* DKO) mice to clarify interaction between *Pcdh15* and *Tmie* (S5 Fig). KO;KO hair bundles demonstrated unaltered EPS8 levels and a single-Gaussian distribution, similar to that in tip-link mutants (S5I and S5J Fig). KO;KO bundles also showed reduced GNAI3 levels and altered distribution as seen in $Tmie^{KO/KO}$ bundles (S5K and S5L Fig). KO;KO bundles shared features of both Het;KO and

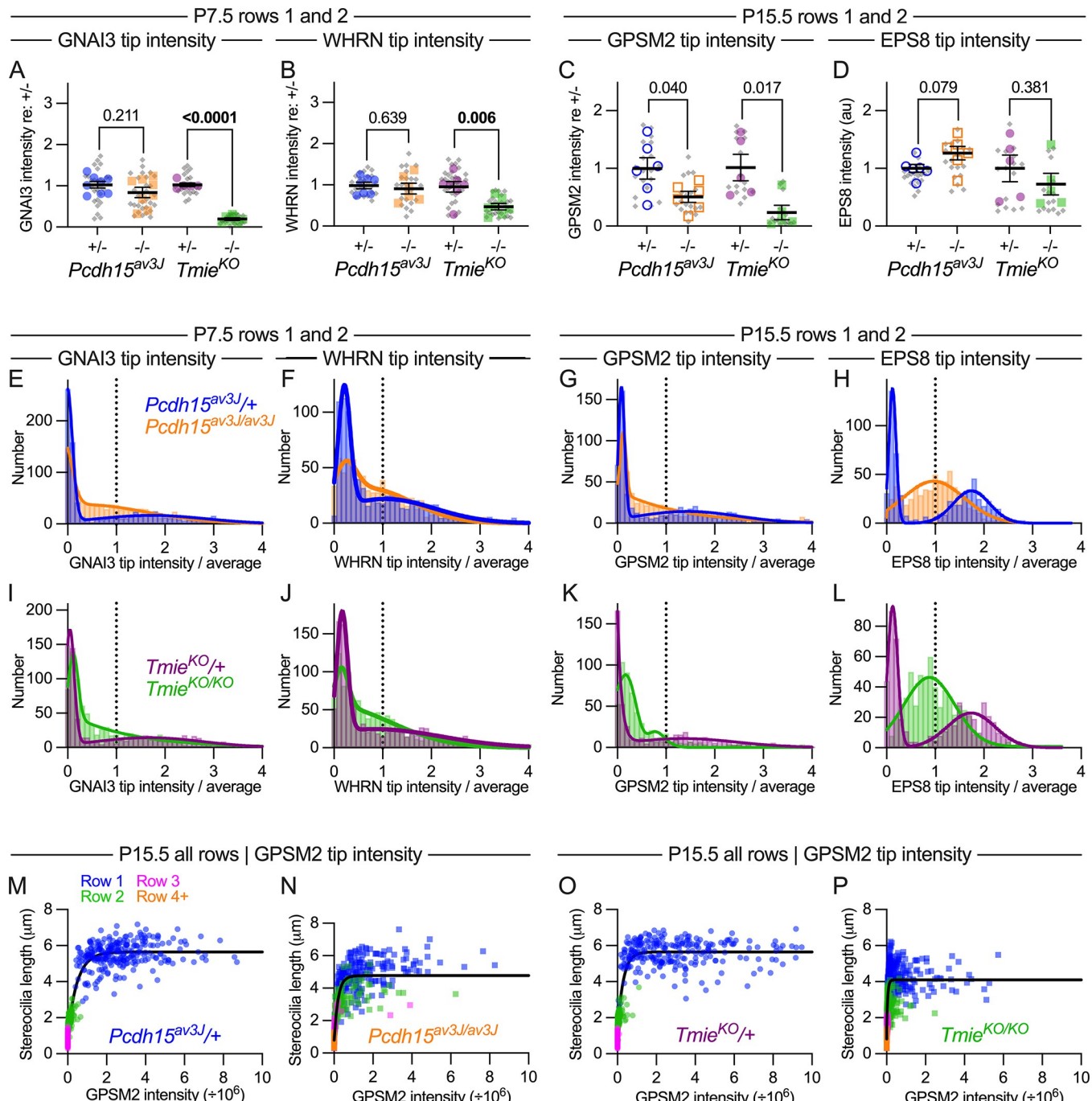

**Fig 12. Distribution of GNAI3, GPSM2, and WHRN at tips of stereocilia in tip-link and transduction mutant IHCs.** (**A, B**) GNAI3 (A) and WHRN (B) fluorescence average intensity per hair bundle for all measured stereocilia (rows 1 and 2) for $Pcdh15^{av3J}$ and $Tmie^{KO}$ hair cells at P7.5. Plotting and statistical testing were as in Fig 3. (**C, D**) GPSM2 (C) and EPS8 (D) fluorescence average intensity per hair bundle for all measured stereocilia (rows 1 and 2) for $Pcdh15^{av3J}$ and $Tmie^{KO}$ hair cells at P15.5. (**E, F**) Frequency distribution of GNAI3 and WHRN tip intensity in $Pcdh15^{av3J}$ hair cells at P7.5. (**G, H**) Frequency distribution of GPSM2 and EPS8 tip intensity in $Pcdh15^{av3J}$ hair cells at P15.5. (**I, J**) Frequency distribution of GNAI3 and WHRN tip intensity in $Tmie^{KO}$ hair cells at P7.5. (**K, L**) Frequency distribution of GPSM2 and EPS8 tip intensity in $Tmie^{KO}$ hair cells at P15.5. Tip intensities were from all rows 1 and 2 stereocilia; distributions were fit with double Gaussian functions. (**M–P**) Correlation of normalized GPSM2 intensity with stereocilia length. Data from individual rows were displayed with separate symbol colors but all data were used for fits; data were fit with $y = a + b \cdot \exp(-k \cdot x)$, where $y$ is stereocilia length, $x$ is GPSM2 intensity, and $a$, $b$, and $k$ are constants. The data underlying all the graphs shown in the figure can be found in figshare ([https://doi.org/10.6084/m9.figshare.21632636.v2](https://doi.org/10.6084/m9.figshare.21632636.v2)). IHC, inner hair cell.

KO;Het bundles (S5A–S5D Fig). Mutations in *Pcdh15* thus did not suppress transduction-mutant phenotypes, nor did mutations in *Tmie* suppress phenotypes seen in tip-link mutants, indicating a lack of genetic interaction for these phenotypes.

## Labeling stereocilia-free barbed ends with rhodamine-actin

Row 2 stereocilia shrink during stage III, then stabilize in length during stage IV (Fig 1G). Actin-filament free ends, measured in permeabilized hair bundles using rhodamine-actin as a probe, were particularly high at row 2 tips in control bundles during early postnatal development but decreased in cochleas of transduction mutants or cochleas treated with transduction-channel blockers [43]. Results with tip-link mutants were very similar to those obtained with $Tmie^{KO/KO}$ mutants [43]. The high level of row 2 labeling by rhodamine-actin seen in control bundles at P7.5 (Fig 13A, 13C, 13M and 13N) had disappeared by P21.5 (Fig 13G, 13I, 13P and 13Q); row 1 labeling was reduced as well at P21.5 (Fig 13P and 13Q).

In $Pcdh15^{av3J/av3J}$ or $Tmie^{KO/KO}$ mutants at P7.5, rhodamine-actin labeling was relatively low, with rows 1 and 2 tips having similar intensities (Fig 13B, 13D, 13M and 13N). That pattern was relatively unchanged at P21.5 (Fig 13H, 13J, 13P and 13Q). The data underlying all the graphs shown in the figure can be found in figshare (https://doi.org/10.6084/m9.figshare.21632636.v2).

By contrast, in $Myo15a^{sh2/sh2}$ mice, row 1 and row 2 rhodamine-actin labeling was elevated to the intensity level seen in row 2 from control mice at P7.5 (Fig 13E, 13F and 13O); rhodamine-actin labeling then declined precipitously in both rows by P21.5 (Fig 13K, 13L and 13R). Thus, during stage III, actin filament free ends appear to be suppressed by the row 1 complex but enhanced by transduction; in turn, increased free ends correlate with row 2 shortening.

## CAPZB no longer concentrates at P21.5 stereocilia tips in tip-link mutants

After P10, CAPZB—and to a lesser extent, EPS8L2—shifts from the shafts of all stereocilia to concentrate at row 2 tips in control hair bundles [6]; this shift does not occur in homozygous transduction mutants [6]. We confirmed results in $Tmie^{KO}$ mice (Figs 14C and 14D and 14G and 14H) and showed further that CAPZB no longer concentrated at row 1 or 2 tips in $Pcdh15^{av3J/av3J}$ IHCs (Fig 14A, 14B and 14G), instead remaining along stereocilia shafts (asterisks in Fig 14B). By contrast, CAPZB concentrated at tips of both rows in relatively equal amounts in in $Myo15a^{sh2/sh2}$ bundles (arrows in Fig 14F and 14G).

EPS8L2 was present at both row 1 and row 2 tips in control hair bundles, with a modest bias towards row 2; EPS8L2 was found at nearly equal levels in both rows of $Pcdh15^{av3J/av3J}$ and $Tmie^{KO/KO}$ IHC bundles (Fig 14A–14D and 14H; arrowheads in Fig 14B). In contrast to CAPZB, EPS8L2 was nearly completely absent from $Myo15a^{sh2/sh2}$ bundles (Fig 14E and 14F). These results suggested that the presence of transduction and the row 1 complex are required for selective targeting of CAPZB to row 2 tips during late development. The appearance of CAPZB at tips by P21.5 was correlated with the decline in free barbed end labeling at this age for both control and mutant bundles, suggesting that capping proteins, row 1 proteins, and transduction all stabilize F-actin at stereocilia tips during stage IV.

## Discussion

The results presented here support a comprehensive model that explains key steps of developmental assembly of the apical IHC hair bundle (Fig 15). By increasing resolution for our measurements of stereocilia dimensions, we defined more accurately Tilney's stages III and IV of stereocilia growth in these hair cells and the point of transition between them [1]. These improvements were made possible by boosting sampling frequency and increasing

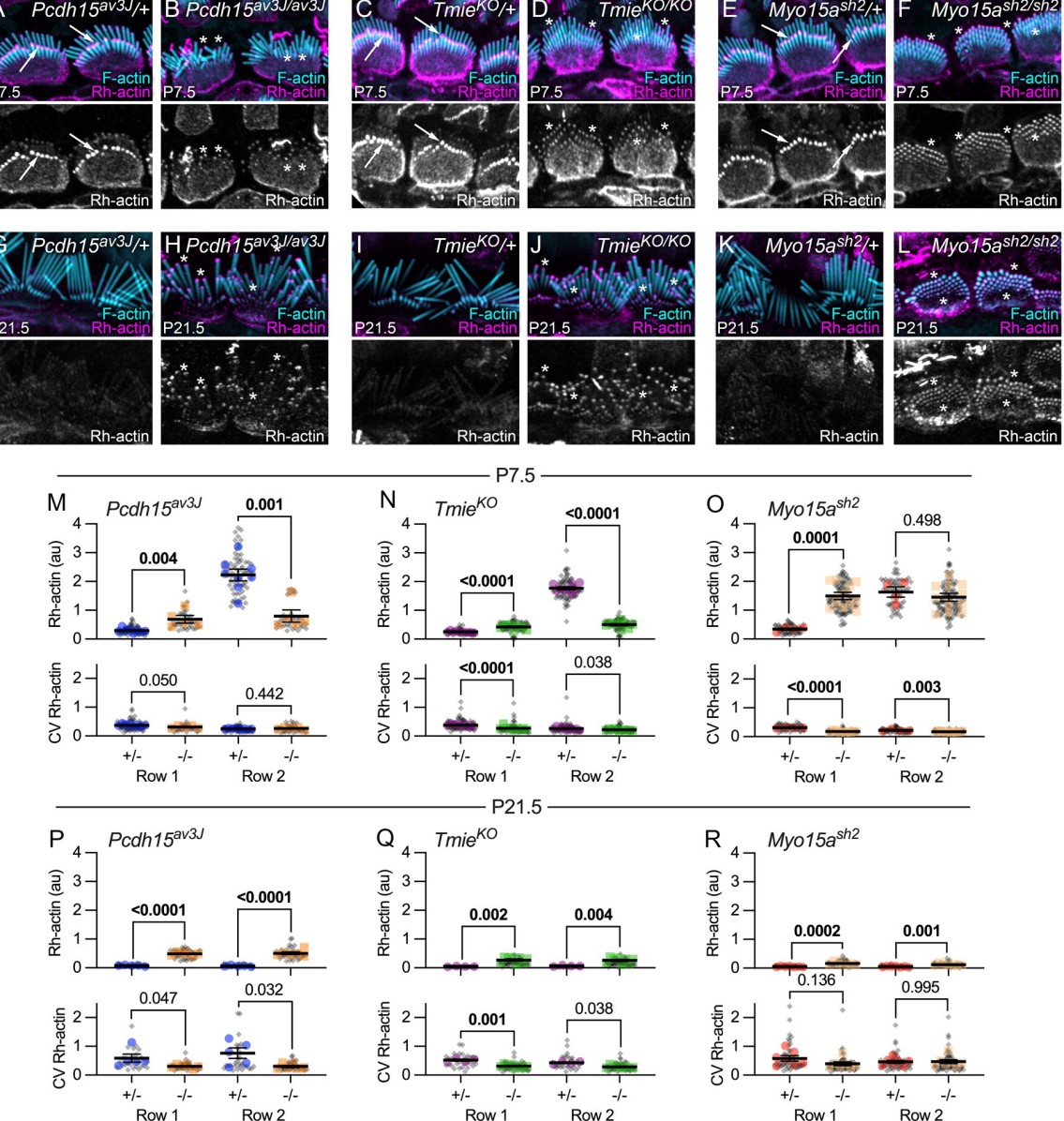

**Fig 13. Free barbed end labeling of IHC stereocilia with rhodamine-actin.** (**A–L**) Rhodamine-actin (Rh-actin) and phalloidin labeling of mutant IHC hair bundles at P7.5 and P21.5. $Pcdh15^{av3J}$ (A–B, G–H), $Tmie^{KO}$ (C–D, I–J), and $Myo15a^{sh2}$ (E–F, K–L). Arrows in A, C, and E indicate high levels of row 2 rhodamine-actin labeling in P7.5 control IHC bundles; asterisks in B, D, F, H, and J indicate similar rhodamine-actin levels at all stereocilia tips in P7.5 and P21.5 mutant IHC bundles. Panel widths: 24 μm. (**M–R**) Rh-actin tip intensity and intensity CV in $Pcdh15^{av3}$, $Tmie^{KO}$, and $Myo15a^{sh2}$ at P7.5 (M–O) and P21.5 (P–R). Plotting and statistical testing were as in Fig 3. The data underlying all the graphs shown in the figure can be found in figshare (https://doi.org/10.6084/m9.figshare.21632636.v2). CV, coefficient of variation; IHC, inner hair cell.

measurement precision using automated analysis. In addition, use of 4 mouse mutant lines allowed us to determine what role the tip-link proteins, transduction proteins, and the row 1 complex play in bundle development. Together, the results provide a picture of several simultaneous processes proceeding in parallel, with significant interconnections that reinforce key transitions.

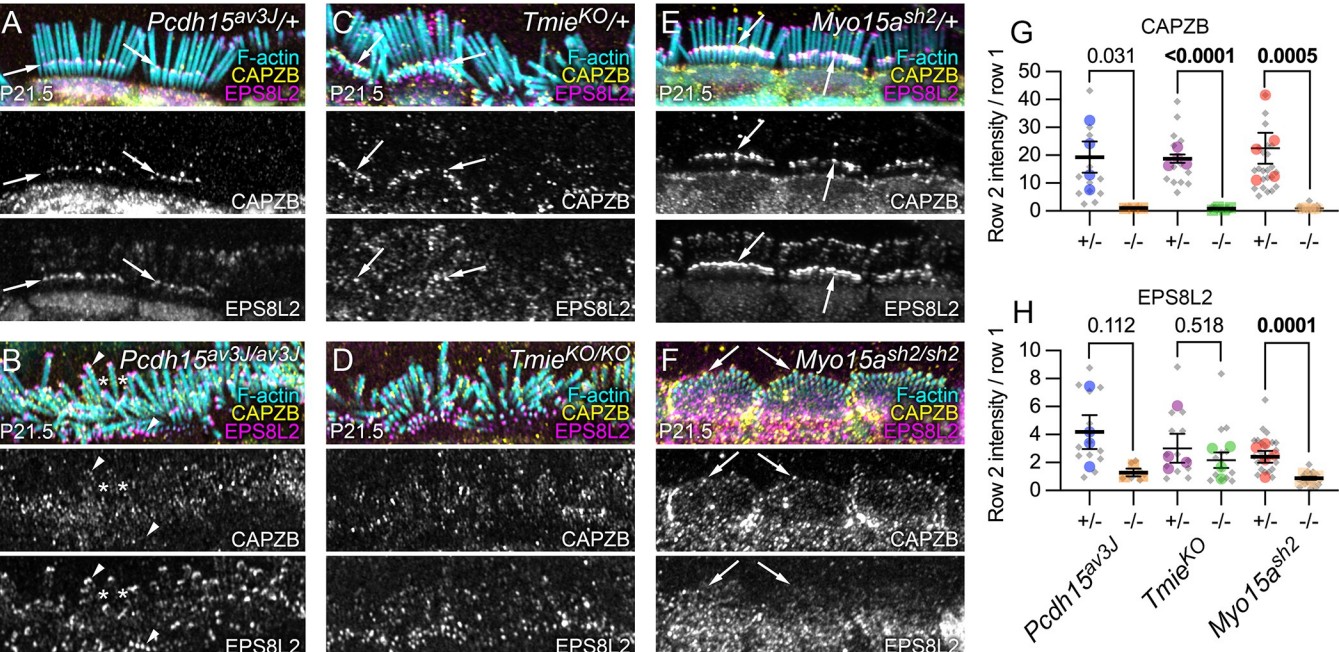

**Fig 14. Row 2 protein localization in mutant IHCs.** (**A–F**) Localization of CAPZB and EPS8L2 in *Pcdh15^{av3J}* (A, B), *Tmie^{KO}* (C, D), and *Myo15a^{sh2}* (E, F) heterozygous and homozygous IHCs at P21.5. Arrows in A, C, and E show high levels of row 2 CAPZB in heterozygotes. In B, asterisks show CAPZB labeling along stereocilia shafts and arrowheads show strong EPS8L2 labeling at rows 1 and 2 tips. Arrows in F show CAPZB at tips of all stereocilia in *Myo15a^{sh2/sh2}*. Panel widths: A-F, 30 μm. (**G, H**) Quantitation of CAPZB (G) and EPS8L2 (H) average intensity in row 2 tips divided by the average intensity in row 1 tips in heterozygotes and homozygotes at P21.5. Plotting and statistical testing were as in Fig 3. The data underlying all the graphs shown in the figure can be found in figshare (https://doi.org/10.6084/m9.figshare.21632636.v2). IHC, inner hair cell.

## Transduction mutants and tip-link mutant have overlapping phenotypes

Although transduction mutants and tip-link mutants share the loss of transduction as a phenotype, hair bundles in these 2 classes of mutants are quite distinct. These results show that not only is $Ca^{2+}$ entry through transduction channels important for regulating stereocilia dimensions [6,17,62], but both the transduction-channel proteins themselves and CDH23-PCDH15 links have significant roles as well. While we noted previously that GPSM2 and GNAI3 were reduced at row 1 tips during stages III and IV in transduction mutants [6], the retention of GPSM2 and GNAI3 at row 1 tips in *Cdh23^{v2J/v2J}* and *Pcdh15^{av3J/av3J}* hair bundles shows that the mislocalization resulted from the loss of transduction proteins, rather than transduction current. Moreover, misregulation of stereocilia length in tip-link mutants highlighted the importance of CDH23-PCDH15 links like transient lateral links and kinocilial links for ensuring that the stereocilia are coherently organized.

Table 1 summarizes the distinct phenotypes for transduction mutants and tip-link mutants, as well as shared phenotypes. The latter phenotypes are likely a direct result of the loss of transduction conductance, while the former phenotypes are specific to the transduction complex or CDH23-PCDH15 links.

## Steady growth of stereocilia building blocks and local regulation

In chick cochlea, hair bundles with widely varying numbers and dimensions of stereocilia still have the same total amount of F-actin in their bundles [60]. Similarly, we found that P7.5 bundles from mouse mutants with differing stereocilia number, length, and width still had the same total stereocilia volume. Moreover, in wild-type hair cells, the total stereocilia volume per

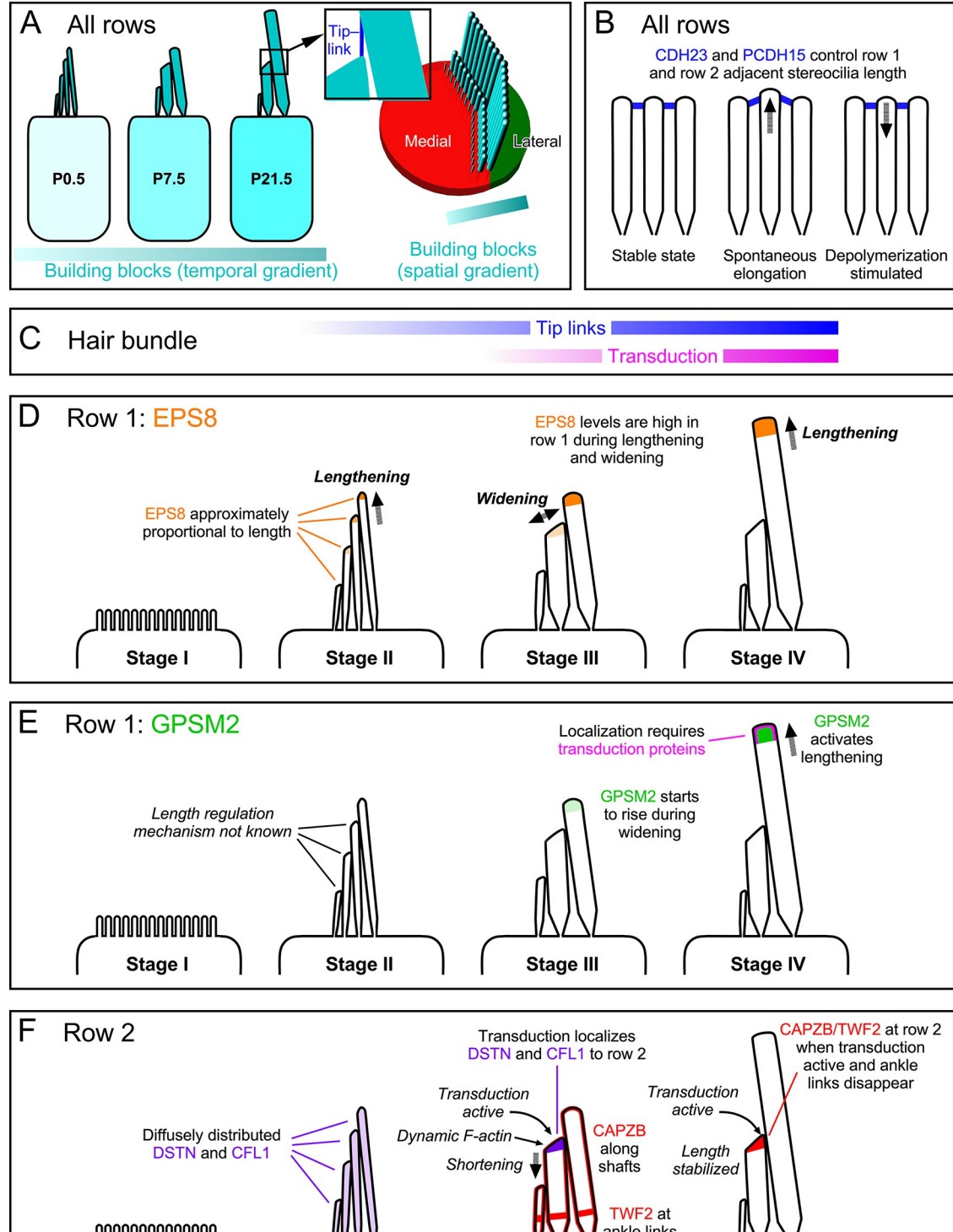

**Fig 15. Model for assembly of IHC hair bundle.** (**A**) Left, stereocilia building blocks (combined actin and actin crosslinkers) increase continuously in IHCs throughout early postnatal development. Right, a spatial gradient of building blocks may also exist, with building blocks having greater access to row 1 stereocilia than to shorter rows. (**B**) PCDH15 and CDH23 maintain coordination of adjacent stereocilia lengths, presumably by triggering actin depolymerization when spontaneous stereocilia elongation interrupts the stable state. (**C**) Timing of tip-link and transduction appearance relative to stages depicted in panels D–F. (**D–F**) Stage-dependent

processes. (**D**) EPS8 is present at all stages of stereocilia growth. In stage II, EPS8 is present at all tips in approximate proportion to stereocilia length. In stages III and IV, EPS8 levels drop at row 2 tips while they remain constant at row 1 tips. (**E**) If transduction proteins are present in the hair bundle, GPSM2 levels rise during stage IV, presumably leading to activation of EPS8 and actin-filament elongation. (**F**) At row 2, the actin-depolymerizing proteins DSTN and CFL1 (and their partner WDR1) are found diffusely in stereocilia. During stage III, transduction leads to localization of DSTN and CFL1 at row 2 tips, triggering actin-filament turnover and shrinkage of row 2. CAPZB and TWF2 are localized separately during this stage, but in stage IV, both are found together at row 2 tips if transduction is active. IHC, inner hair cell.

bundle increased monotonically despite the very different growth behavior of individual stereocilia rows during stages III and IV. While cellular actin transcript and protein levels do not rise during early postnatal bundle development [63–65] (see also umgear.org), actin cross-linkers do increase in expression over this period [64,66] (umgear.org). Moreover, loss of crosslinkers affects stereocilia dimensions [66–68], and FSCN2 overexpression lengthens stereocilia [69]. A continually increasing supply of actin crosslinkers, including ESPN, FSCN2, and PLS1, may therefore steadily increase the supply of "building blocks" (i.e., actin plus actin crosslinkers) for stereocilia growth during early postnatal development.

These results suggest that local regulatory mechanisms must be switched on and off rapidly, then control how these building blocks are exploited. During stage III, lengthening of stereocilia was completely inhibited; cross-sectional area grew linearly during this time, suggesting a constant rate of addition of actin filaments to the periphery of the stereocilia core. At P8, however, a sharp transition occurred; lengthening resumed and then proceeded exponentially for the next 2 weeks.

Stereocilia building blocks appear to be made available to the hair bundle in a lateral to medial gradient within the hair cell; even when tip links are not present and the row 1 complex is distributed to most stereocilia, a row 1 through row 3 gradient in length was still apparent. This gradient was even apparent in $Myo15a^{sh2/sh2}$ mutants, which lack the row 1 complex at any tips. A plausible explanation is that building blocks are more accessible at the fonticulus, the aperture in the cuticular plate that the kinocilium extends through. This region is not only filled with microtubules and vesicles [70] but also could be a region where soluble molecules like actin monomers and crosslinkers more freely diffuse. The alternative explanation that stereocilia growth starts later in shorter stereocilia is not supported by the comparison of rows 1 and 2 kinetics in C57BL/6 hair cells (Fig 1).

**Table 1. Comparison of transduction mutants and tip-link mutants.**

| Mutant type | Primary phenotypes | Secondary phenotypes | Functional consequences |
|---|---|---|---|
| Transduction mutants | Loss of individual transduction proteins | Transduction complex does not assemble or travel to stereocilia tips | Reduced GPSM2 and GNAI3 at row 1 ↓ Reduced row 1 stereocilia length ↓ Freed building blocks ↓ Supernumerary rows |
| | Loss of transduction pore | Loss of transduction (e.g., Ca²⁺ entry) | Reduced row 2 width Reduced free barbed ends in row 2 Defective pruning Increased EPS8 in row 2 tips No row 2/3 shortening during stages III–IV |
| Tip-link mutants | Loss of tip links and channel gating | | |
| | Loss of transient lateral links | Disarrayed stereocilia placement Uneven stereocilia lengths in a row | Reduced mechanosensitivity due to broadened deflection-current relationship |
| | Loss of kinocilial links | Central row 1 stereocilia displaced laterally | Notch in hair bundle |

Blue color indicates phenotypes of transduction mutants, green color indicates phenotypes of tip-link mutants, and orange color indicates shared phenotypes.

## The row 1 tip complex controls lengthening in stages II and IV

The biochemical properties of the row 1 tip complex, composed of EPS8, MYO15A, WHRN, GNAI3, and GPSM2, suggest how the complex controls stereocilia length. EPS8 and MYO15A each can control actin polymerization [71,72], and early in development, the 2 proteins target to all stereocilia in amounts approximately proportionally to stereocilia length [7]; moreover, together both proteins form a phase-separated complex that may promote actin polymerization [10]. In the absence of transduction [6] or tip links (this work), the EPS8-MYO15A complex is no longer exclusively targeted to row 1 during postnatal development and instead appears at all tips, also in proportion to length. These results suggest that the EPS8-MYO15A complex is targeted to all tips during normal development, but that other events cause their accumulation with the rest of the complex members only at the tips of row 1.

As long as functional RGS12 is present [73], GPSM2 and GNAI3 are exclusively found at row 1 tips [8,9]; they assemble with the EPS8-MYO15A complex [7–9] using WHRN as an adapter [7]. GPSM2 forms large protein condensates in vitro, both by itself and with the entire row 1 complex [11], and it is reasonable to suggest that GPSM2 drives the formation of similar condensates at the tips of row 1 stereocilia. When the row 1 complex is formed, GPSM2 activates EPS8's F-actin bundling activity, which likely promotes stereocilia lengthening [11]. GNAI3 can target to stereocilia tips independently from other row 1 proteins [7] and may connect the membrane at row 1 tips to the rest of the row 1 complex [11].

Previous characterization of the row 1 tip complex has not distinguished its activity during the 2 lengthening stages of hair-bundle development [1]. In apical IHCs, stage II—initial lengthening of stereocilia—occurred prior to our first measurements at P0.5. Nevertheless, Tadenev and colleagues demonstrated that GNAI3 and GPSM2 are required for this initial lengthening stage. First, both proteins appear at row I tips during stage II [7]. In addition, hair cells that express pertussis toxin, which inactivates all GNAI isoforms, have very short stereocilia, as do *Gpsm2*-null hair cells [7]. These stereocilia are much shorter than those of bundles at P0.5, which is at the end of stage II. Bundles lacking EPS8, MYO15A, or WHRN each show the same short-stereocilia phenotype [74–76].

Although these simple knockouts do not probe later steps in development, our experiments suggest that GNAI3 and GPSM2 are also required for the final stereocilia lengthening that occurs during stage IV. GNAI3 and GPSM2 each build up at tips of row 1 stereocilia during stages III and IV, with GPSM2 peaking later than GNAI3. $Cdh23^{v2J/v2J}$ and $Pcdh15^{av3J/av3J}$ mutants, which retain GNAI3 and GPSM2 in row 1, have row 1 lengths nearly equal to those of controls. By contrast, $Tmie^{KO/KO}$ mutants, which do not accumulate GNAI3 or GPSM2 in row 1 stereocilia during stage IV, have shorter row 1 stereocilia.

## Loss of CDH23 or PCDH15 leads to uncoordinated stereocilia lengthening

Our experiments also revealed several specific functions for CDH23 and PCDH15, presumably as heterotetrametric links, during development of apical IHC hair bundles. While CDH23-PCDH15 tip links gate transduction channels and transduction impacts bundle development [6], we show here that CDH23 and PCDH15 are also essential for coordinating the lengths of adjacent stereocilia in the same row. While we cannot rule out effects specific to CDH23 or PCDH15, the similarity in phenotype between $Cdh23^{v2J}$ and $Pcdh15^{av3J}$ mouse lines and demonstrated interaction between them [26,77,78] makes it most likely that roles for these 2 proteins, like length coordination, require their interaction.

CDH23 and PCDH15 make up the kinocilial links [33,34], and loss of either molecule leads to detachment of the kinocilium and an altered hair-bundle structure [37,79]. Loss of kinocilial links in the tip-link mutants likely explains the notch that develops in the bundle, where the

stereocilia that should have been connected to the kinocilium are translocated medially on the apical surface. Because they are only found on a few stereocilia, however, the kinocilial links cannot be responsible for the bulk of the length coordination carried out by CDH23 and PCDH15.

CDH23 and PCDH15 have each been suggested to contribute to the transient lateral links [24,26,31,32]; these links are likely essential for coordinating adjacent stereocilia lengths. Transient lateral links bridge the gap between adjacent stereocilia along their shafts and are prominent in mouse IHCs and OHCs between E17.5 and P0; they begin to disappear after P0, and are no longer seen after P5 except at stereocilia tips [80]. Although some links between adjacent stereocilia remain in both $Pcdh15^{av3J/av3J}$ and $Cdh23^{v2J/v2J}$ hair bundles (Fig 2), these are likely ankle links [80,81].

Transient lateral links are similar to the thread-like links that interconnect the distal microvilli tips in the intestinal brush border [82]; these links are composed of CDHR2 (PCDH24) and CDHR5, members of the adhesion complex, and are essential for microvillar clustering and length uniformity [82]. Length uniformity in microvilli of the brush border has been proposed to arise from stimulation of actin dynamics through the adhesion complex or by physical constraints, as a link could prevent an adjacent microvillus from elongating [83]. In IHCs, we propose that dimeric complexes of CDH23 and PCDH15 carry out the same function throughout development.

Nevertheless, the morphology of $Pcdh15^{av3J};Myo15a^{sh2}$ DKO hair bundles was dissimilar to that of CACO-2$_{BBE}$ intestinal cells with shRNA knockdown of PCDH24; these latter cells had microvilli of widely differing lengths that lacked hexagonal arrangement [82]. DKO bundles retained a modest gradient in stereocilia length, despite the lack of tip links, and adjacent stereocilia were similar in length; while packed less tightly than either $Pcdh15^{av3J}/+;Myo15a^{sh2}/+$ or $Pcdh15^{av3J}/+;Myo15a^{sh2/sh2}$ stereocilia, packing was still hexagonal. The retention of length coordination, albeit to very short lengths, suggests that additional mechanisms contribute to regulation of stereocilia length in addition to the row 1 complex and PCDH15. Physical constraints of the overlying tectorial membrane probably do not contribute to length regulation; when the tectorial membrane is eliminated genetically, IHC and OHC bundles retain their normal morphology and stereocilia lengths are aligned [84,85].

A system for aligning lengths of adjacent stereocilia may have evolved to produce optimal activation of transduction channels. Alignment of stereocilia lengths in each row should coordinate transduction channel gating and increase sensitivity. For small stimuli, the shear between 2 stereocilia can be approximated by $x = \gamma \cdot X$ [86], where $X$ is the displacement applied to the hair bundle's tip and $\gamma$ is a geometrical gain factor; $x$ is the distance that the gating spring is stretched by the stimulus and hence is the relevant activation delivered to the transduction channels [87]. In turn, $\gamma = S / L$, where $S$ is the spacing between stereocilia at their bases and $L$ is the length of the stereocilia at the point of shear. In wild-type bundles, spacing between adjacent stereocilia in a given row is constant within that row and from bundle to bundle [86]. By contrast, in tip-link mutants, variability in $L$ of row 2 will produce variability in $x$; if a cell has variations in $x$ with a constant $X$, channel open probabilities will vary, perhaps considerably. In turn, variability in open probabilities will broaden the displacement-transduction current relationship of the bundle and reduce sensitivity. Tight control of stereocilia length within a row is thus required to maintain the hair cell's high sensitivity.

## Transduction proteins stimulate accumulation of the row 1 complex

Our results here indicate that transduction proteins themselves regulate the amount of GNAI3 and GPSM2 that accumulate at row 1 tips during stage IV. While little GNAI3 (or GPSM2) is

found in row 1 of $Tmie^{KO/KO}$ mutant hair bundles, GNAI3 and GPSM2 concentrate relatively normally in row 1 stereocilia in $Cdh23^{v2J/v2J}$ and $Pcdh15^{av3J/av3J}$ mutants. Stages III and IV localization of these proteins to row 1 tips occurs long after the initial lengthening of stereocilia adjacent to the kinocilium, so transduction mediated through kinocilial links [34] is likely not responsible. TMIE and the TMCs do localize to row 1 near tips during early postnatal development [23], and at least TMC2 is present in row 1 in $Pcdh15^{av3J/av3J}$ mutants (S3 Fig).

Accordingly, if TMC channels transiently localize to tips of row 1 stereocilia during development, GNAI3 or GPSM2 may bind directly or indirectly to them (Fig 15E). This coupling could be mediated by the established CIB2-WHRN interaction [88]; CIB2 is part of the transduction-channel complex [89–91] and WHRN interacts with GPSM2 [7,11]. TMC channels are no longer at row 1 tips in older stereocilia [92], so localization of GNAI3 and GPSM2 to row 1 by the transduction complex may only be necessary at the beginning of stage IV to initiate formation of a robust row 1 complex.

## Transduction controls actin dynamics at row 2 tips

Transduction controls actin-regulatory mechanisms in row 2 differently than those in row 1. For example, during stage III, row 2 stereocilia shorten while those of row 1 are static. In addition, the complement of proteins at row 2 tips differs substantially from that at row 1 tips. Our results confirm the observations of McGrath and colleagues, who demonstrated that row 2 tips transiently have abundant free actin filament ends, but not in $Tmie^{KO/KO}$ mutants nor in the presence of channel blockers [43]. $Pcdh15^{av3J/av3J}$ mutants also lose strong row 2 rhodamine-actin labeling, confirming that transduction itself—rather than transduction proteins—is responsible for this effect.

Effects of transduction on protein localization likely occur because transduction channels are partially open at rest during development [93], and because substantial $Ca^{2+}$ entry occurs when channels are open [16]. The steep local gradient of $Ca^{2+}$ may in turn localize specific proteins to row 2 tips but not elsewhere in the row 2 stereocilia.

Transduction recruits DSTN (ADF) and CFL1 to stereocilia tips [43], which may depend on local $Ca^{2+}$ entry (Fig 15F). Moreover, in many contexts, $Ca^{2+}$ influx increases DSTN and CFL1 activity by activating $Ca^{2+}$-dependent phosphatases [44]. In tip-link or transduction mutants, not only is the high level of rhodamine-actin labeling at row 2 tips reduced, but rhodamine-actin labeling at row 1 tips also increases. These results suggest that actin dynamics are regulated by transduction at both row 1 and row 2 tips, but in opposite directions. Transduction contributes to the polarization of EPS8 into row 1, as both $Tmie^{KO/KO}$ and $Pcdh15^{av3J/av3J}$ mutants have relatively similar amounts of EPS8 in all tips; in this case, the similarity in phenotype between the 2 mutants suggests that transduction currents themselves mediate EPS8 exclusion from the shorter rows. By preventing accumulation of EPS8 at row 2 tips, transduction could thereby promote its accumulation at row 1, thus contributing to the polarization in actin dynamics between the 2 rows.

These results suggest that transduction stimulates actin depolymerization at the tips of the shorter rows [42,43]. Moreover, by increasing actin dynamics, transduction thus also controls stereocilia remodeling (row 2) or resorption (rows 3 and beyond) during development. A key question is why this model, which suggests that transduction promotes row 2 shortening, is contradicted by experiments showing that transduction-channel block stimulates row 2 shortening [17]. A plausible explanation for this discrepancy is that these latter experiments were conducted on stage IV hair bundles [6]. Thus, while transduction promotes row 2 shortening during stage III, changes that occur during stage IV reverse the sign of transduction's effect. The F-actin capping complex of CAPZB and TWF2 appears during stage IV only if transduction is present [6], so the effect of blocking transduction during stage IV may be to dissociate the capping complex and trigger stereocilia shortening.

## A model for hair-bundle length regulation during stages III and IV

Our model for development of mouse apical IHC hair bundles (Fig 15) references Tilney's chick bundle development model [1]; it also exploits key observations made in developing mouse cochlear hair cells [7–9,43] and biochemical studies of the row 1 complex [7,11]. In this model, mouse bundles undergo temporally segregated initial lengthening (stage II), widening (stage III), and final lengthening (stage IV) steps. A key feature of bundle development is the creation of 3 stereocilia rows, each with unique dimensions for their stereocilia. Establishment of these rows occurs when symmetry within the bundle is broken, allowing distinct actin-regulatory mechanisms to control F-actin growth within individual stereocilia rows.

At least 2 global mechanisms, not specific to any row, also regulate growth of hair bundles. The first is that the concentration of building blocks for stereocilia, including G-actin, actin crosslinkers, and other essential actin cytoskeleton proteins, slowly increases in each hair cell over development. In all rows, EPS8 may mediate incorporation of these building blocks into growing stereocilia (Fig 15A). A second global mechanism is coordination of stereocilia length, which occurs within each row because of the presence of CDH23 and PCDH15, presumably through the transient lateral links (Fig 15B).

A key event during development of the bundle is the breaking of symmetry between the rows. Asymmetry is created not only by increasing the length of row 1, but also by limiting the growth of the shorter rows. The 3 main symmetry-breaking mechanisms that affect actin regulation are (1) establishment of the row 1 complex by GNAI3 and GPSM2; (2) the appearance of tip links, which pull on the tips of the shorter stereocilia but not on row 1 tips; and (3) appearance of transduction in the shorter rows. These mechanisms in turn control 3 actin-regulatory processes: first, F-actin bundling and polymerization, which lengthen or widen stereocilia; second, F-actin depolymerization, which shortens or thins stereocilia; and finally, F-actin capping, which stabilizes filaments and hence stereocilia length. By locally altering the balance of these 3 regulatory processes, stereocilia of varying dimensions can be created.

GPSM2 and GNAI3, which are only recruited to row 1, may boost EPS8 F-actin-bundling activity to lengthen row 1 during stage II (Fig 15E); regardless of the mechanism of control, stage II lengthening likely is mediated by EPS8-MYO15A complexes, which are present at this early stage at rows 2 and 3 tips [7]. GPSM2 and GNAI3 also control lengthening during stage IV (Fig 15E). While EPS8 levels are high across stages III and IV, GPSM2 and GNAI3 levels are low at the beginning of stage III widening. As stage III progresses, however, GNAI3 begins to accumulate at row 1 tips, reaching a peak during early stage IV. This accumulation depends on the presence of the transduction complex. GPSM2 appears at row 1 tips later during stage IV (Fig 15E), presumably requiring GNAI3 to establish its presence first, and recruitment of GPSM2 may be directly responsible for the burst in lengthening that occurs then. During stage IV, only row 1 of apical IHCs lengthens.

Tip links appear early in development of the hair bundle (Fig 15C), before the appearance of transduction [52,56,94]. Tip-link tension pulls the membrane at the tip of the short stereocilium of the interconnected pair away from the F-actin core beneath (tip-link tenting), which eliminates the membrane's suppression of actin polymerization [95,96]. In one model, actin polymerization proceeds as the membrane is tented, allowing stereocilia lengthening [40]. During stage II, the shorter rows may lengthen simultaneously with row 1, but stereocilia in the shorter rows may be limited in their maximal length because the molecular motor pulling on the tip link would presumably run to the end of its track on the adjacent (taller) stereocilium. Thus, bundles formed by the end of stage II have multiple rows of thin stereocilia with a relatively short stereocilia staircase spacing (Fig 1A). Bundles around P0 are also often tilted strongly in the negative direction [52,56], which may occur when tip links draw all of the

stereocilia together. Negatively tilted bundles are especially apparent in OHCs. Because of the different structures at each of their ends, tip links thus contribute to the distinction between row 1 and the shorter rows.

The appearance of transduction during stage III (Fig 15C) allows for further divergence of row identity (Fig 15F). Steady $Ca^{2+}$ influx, due to the transduction channels being partially open at rest, is present at all rows except row 1 [22]. DSTN and CFL1 are recruited to the tips of shorter stereocilia if transduction is active, and these proteins may also be activated indirectly by entering $Ca^{2+}$. Activated DSTN and CFL1, in turn, increase the turnover of actin filaments at tips of these rows, which may directly lead to both row 2 shortening and pruning (disappearance) of other rows. In addition, EPS8 declines at row 2 tips during stage III (Fig 15D), further distinguishing these tips from those of row 1 stereocilia. The presence of dynamic actin at row 2 tips may explain this decline. EPS8 may still target to the tips of shorter rows, but could be released after filament severing by DSTN and CFL1. The F-actin capping protein EPS8L2 instead concentrates at row 2 tips [6,15], and its mechanism of localization and function likely differs from that of EPS8.

Changes in length of one row may affect the availability of building blocks for remaining rows, especially later in development. For example, without robust lengthening of row 1 (e.g., in $Myo15a^{sh2/sh2}$ hair bundles), more building blocks are available to thicken shorter stereocilia rows, which also stabilizes them. If depolymerization activity is low due to a lack of transduction currents (e.g., in transduction or tip-link mutants), supernumerary stereocilia can thicken. In addition, allocation of stereocilia building blocks appears to extend along a lateral-medial gradient, such that the stereocilia closest to the lateral edge (the bare zone) receive building blocks at a higher rate than those farthest away. Finally, the taller a stereocilium, the farther it is from cytoplasmic pools of DSTN and CFL1 and cytoplasmic changes in $Ca^{2+}$. Short stereocilia whose filaments are not capped by EPS8, EPS8L2, CAPZB, or TWF2 will be more susceptible to resorption. As shown here and previously [6], localization of CAPZB at stereocilia tips requires transduction. In older mutant hair cells that have lost transduction, row 3 and even row 2 stereocilia degenerate [62], which could be due to the loss of CAPZB and a subsequent loss of stereocilia actin [14].

## How the mutant phenotypes arise

The mutant phenotypes described here each result from failure in one or more of the 3 main symmetry-breaking events that normally occur during development. In $Myo15a^{sh2/sh2}$ hair bundles, without trafficking of the row 1 complex, stereocilia length asymmetry cannot be created; lengthening requires EPS8, which is transported by MYO15A. The lack of lengthening of the taller rows frees stereocilia building blocks; these building blocks are subsequently used to thicken rows 3 and 4+, as well as eventually creating cytocauds. In addition, without mechanisms to uniquely distinguish row 1, all stereocilia become similar to row 2. For example, all rows show high levels of rhodamine-actin incorporation at tips and equal levels of accumulation of DSTN and CFL1. Capping proteins that usually enrich at row 2 (EPS8L2, CAPZB, and TWF2) are present at all rows, presumably preventing pruning of the shorter rows.

While $Tmie^{KO/KO}$ and tip-link mutants both lack transduction, they differ in the establishment of the other 2 main symmetry-breaking events. In $Tmie^{KO/KO}$ mutants, members of the row 1 complex are still present, but neither GPSM2 nor GNAI3 traffic efficiently to row 1 tips in the absence of transduction proteins, at least during stage IV. Stage IV lengthening of row 1 stereocilia is therefore significantly reduced as there is no additional stimulation of EPS8 activity in that row by GPSM2. Since row 1 still grows taller than the other rows, both tip links and the lateral-medial gradient in building blocks contribute to forming shallow gradients in length and width. In

*Cdh23$^{v2J/v2J}$* and *Pcdh15$^{av3J/av3J}$* tip-link mutants, members of the row 1 complex still traffic and accumulate at normal levels at tips, so stage IV lengthening continues to occur and taller stereocilia are created. Because kinocilial links are lost and hair bundles change in shape, allocation of GNAI3, GPSM2, and WHRN becomes more variable; in some cases, they are delivered to stereocilia beyond just those bordering the bare zone. More randomized allocation of row 1 complex members and the varying positions of stereocilia across the lateral-medial resource gradient may also create length variability in *Cdh23$^{v2J/v2J}$* and *Pcdh15$^{av3J/av3J}$* mutants.

Due to the loss of tip links, all stereocilia tips in *Cdh23$^{v2J/v2J}$* and *Pcdh15$^{av3J/av3J}$* hair bundles develop a rounded shape; beveled tips of the shorter rows of both control and *Tmie$^{KO/KO}$* stereocilia fail to develop. EPS8 accumulates at more equal levels across all stereocilia tips in tip-link mutants as compared to the *Tmie$^{KO/KO}$* mutants, which may occur because all stereocilia develop the rounded tip shape of row 1. In tip-link and transduction mutants, loss of transduction promotes equal but reduced levels of DSTN/CFL1 activity at all tips. Although *Dstn;Cfl1* double-knockout mice retain a polarized EPS8 distribution [43], seemingly contradicting aspects of our model, the induction of the double knockout occurred postnatally, after accumulation of EPS8 already occurred at row 1 tips. Regardless of the mechanism, EPS8 accumulation and more equalized distribution of DSTN/CFL1 apparently allows some supernumerary rows to thicken and persist.

## Conclusions

Development of the hair bundle is guided by 3 symmetry-breaking events—the formation of the row 1 complex, the asymmetric distortion of the lower-stereocilium membrane caused by tip-link tension, and the establishment of mechanotransduction only at stereocilia tips of shorter rows. During stages II, III, and IV of bundle development, CDH23 and PCDH15 are essential for coordination of stereocilia length within a row. During stage II, before transduction, lengthening of row 1 is accompanied by lengthening of shorter rows because tip links under tension stimulate lengthening; in wild-type IHCs, this produces a bundle with 3 to 4 rows and an even staircase. Lengthening of row 1—and hence the short rows too—stops at the beginning of stage III, when transduction first appears. Stereocilia widen during stage III since building blocks continue to be supplied. During stage III, row 2 stereocilia (and row 3) shorten modestly due to the action of DSTN and CFL1 but also due to the exclusion of EPS8. The row 1 complex assembles during stages III and IV in a stepwise manner, with the level of accumulation dependent on the expression of transduction proteins. Rapid growth of row 1 stereocilia does not occur until sufficient GPSM2 is present; these stereocilia then lengthen while those of the shorter rows continue to shrink. Thus, the interplay of the row 1 complex and transduction creates the final apical IHC bundle, with its dramatic length differences between the rows.

While results here are from apical inner hair cells, development of hair bundles from apical OHCs, as well as those of mid and basal IHCs and OHCs, may also follow the specific stages described here. This scheme was initially developed to describe bundle development in the chicken cochlea [1], and it has proved applicable to mammalian hair cells. Nevertheless, we expect that other cochlear hair cells in the mouse will regulate these stages at different times and to different degrees; differential regulation in turn will produce bundles with distinct stereocilia lengths and widths, as is seen across the cochlea. In the future, we will more broadly apply our quantitative approach to bundle development to determine the developmental principles that apply to all bundles.

## Materials and methods

### Ethics statement

All animal studies were performed in accordance with the guidelines established by the Oregon Health & Science University Institutional Animal Care and Use Committee (protocol

#IP00000714) and the Johns Hopkins University Institutional Animal Care and Use Committee (protocol #M016M271).

## Mice

Mice were housed in individually ventilated cages (IVCs; Thoren, Hazelton, Pennsylvania, United States of America) with a maximum of 5 animals per cage with food and water ad libitum. Breeder pairs were separated 16 to 18 days after crossing and single-housed animals were provided with nesting material; pregnant dams were given breeder chow (5058 PicoLab; LabDiet) in place of standard chow (5LOD Irr Rodent; LabDiet). Mice were housed under barrier-specific pathogen-free conditions in accordance with OHSU IACUC. Mouse pups were assumed to be born at midnight, so animals used on the first day are referred to as P0.5. Data labeled as "P21.5" here refers to mice at weaning age; while most animals were P21.5 (including all animals used in Fig 1), we used a range of ages from P20.5 to P25.5. Both male and female pups were used; because newborn mice cannot be sexed without genetic analysis, we used a mixture of sexes in all experiments. We analyzed IHCs from the basal half of the apical approximately 1/3 of the cochlea (17% to 33% of the cochlear length, measured from the apex; roughly the 125th IHC from the apex through the 250th), and we refer to them as apical IHCs.

For developmental analysis, we used C57BL/6J mice (RRID:IMSR_JAX:000664, Jackson Laboratories, Bar Harbor, Maine, USA). While C57BL/6J mice exhibit progressive hearing loss relative to other strains, at 1.5 months of age (i.e., older than the mice we used), auditory brainstem responses and cochlear hair cell preservation were identical to strains with excellent hearing, such as CBA/Ca [97]. All mutant mice were backcrossed for more than 10 generations to C57BL/6J. The $Cdh23^{v2J}$ line has been described previously [37,54]; the $v2J$ mutation changes a donor splice site and produces stop codon containing mutant splice forms with that are predicted to produce truncated proteins lacking the transmembrane domain. The $av3J$ mutation in $Pcdh15$ also leads to a premature stop codon prior to the transmembrane domain [37,53]. Accordingly, both $Cdh23^{v2J}$ and $Pcdh15^{av3J}$ are expected to be full null mutations [37]. The $Tmie^{KO}$ line has been described previously [6,18], as has $Myo15a^{sh2}$ [59,74].

## Primary antibodies

Anti-GPSM2 Cat# HPA007327 (RRID: AB_1849941) was from Sigma-Aldrich (St. Louis, Missouri, USA). Rabbit polyclonal anti-GNAI3, directed against a C-terminal mouse GNAI3 peptide (KNNLKECGLY), was Cat# G4040 (RRID: AB_259896) from Sigma-Aldrich. Mouse monoclonal anti-EPS8 (clone 15), against mouse EPS8 amino acids 628–821, was Cat# 610143; RRID: AB_397544 from BD Bioscience (San Jose, California, USA). Rabbit polyclonal anti-WHRN (#8127), against a C-terminal mouse WHRN peptide with an added cysteine (KERDYIDFLVTEFNVML-C), was from the Ulrich Müller lab. Mouse monoclonal anti-CAPZB2, against a C-terminal peptide of human CAPZB2, was Cat# AB6017 (RRID: AB_10806205) from EMD Millipore (Burlington, Massachusetts, USA). Mouse monoclonal anti-EPS8L2, against human EPS8L2 amino acids 615–715, was Cat# ab57571 (RRID: AB_941458) from Abcam (Cambridge, United Kingdom). Anti-PCDH15 (HL5614) was generated in rabbit and obtained from Zubair Ahmed and anti-CDH23 was generated in goat by GenScript (Piscataway, New Jersey, USA) against mouse CDH23 EC15/16 (CATRPAPPDRERQ) [98].

## Immunofluorescence sample preparation

Inner ears from C57BL/6J mice or mutant mice littermates dissected at the indicated ages in cold Hank's balanced salt solution (Thermo Fisher Scientific Cat# 14025076) supplemented

with 5 mM HEPES (pH 7.4) (dissection buffer). To allow perfusion of the fixative, small openings were made within the periotic bones.

For measurements of stereocilia dimensions from C57BL/6J or mutant mice using only phalloidin (Figs 1 and 3–8), ears were fixed in 4% formaldehyde (Electron Microscopy Sciences, Hatfield, Pennsylvania, USA; Cat# 1570) in 1× PBS for 0.5 to 1 h at room temperature. Ears were washed in PBS, then cochleas were dissected out from the periotic bone and the lateral wall was removed. After permeabilization in 0.2% Triton X-100 in 1× PBS for 10 min at room temperature, cochleas were incubated with phalloidin (0.4 U/ml Alexa Fluor 488 phalloidin; Thermo Fisher Scientific Cat# A12379) in 1× PBS for 12 to 16 h at 4˚C. Organs were washed 3 times in PBS for 5 min per wash and mounted in Vectashield (Vector Laboratories, Newark, California, USA; Cat# H-1000).

For immunofluorescence using antibodies against row 1 or 2 proteins (Figs 8–11 and S4), ears were fixed in 4% formaldehyde (Electron Microscopy Sciences Cat# 1570) in dissection buffer for 20 to 30 min at room temperature. As noted previously, row 2 protein antibodies (EPS8L2, CAPZB) required fixation of at most 20 min, whereas row 1 protein antibodies were not sensitive to fixative duration. Ears were washed in PBS, then cochleas were dissected out from the periotic bone and the lateral wall was removed. Cochleas were permeabilized in 0.2% Triton X-100 in 1× PBS for 10 min and blocked in 5% normal donkey serum (Jackson ImmunoResearch, West Grove, Pennsylvania, USA; Cat# 017-000-121) diluted in 1× PBS (blocking buffer) for 1 h at room temperature. Organs were incubated overnight at 4˚C with primary antibodies in blocking buffer and then washed 3 times in 1× PBS. Dilutions were 1:500 for anti-GNAI3; 1:250 for anti-GPSM2, anti-EPS8, anti-EPS8L2, anti-WHRN, and anti-CAPZB. Tissue was then incubated with secondary antibodies, which were 2 µg/ml donkey anti-rabbit Alexa Fluor 488 (Thermo Fisher Scientific Cat# A21206) and 2 µg/ml donkey anti-mouse Alexa Fluor 568 (Thermo Fisher Scientific Cat# A10037); 1 U/ml CF405 phalloidin (Biotium, Fremont, California, USA; Cat# 00034) was also included for the 3 to 4 h room temperature treatment. Tissue was washed 3 times in PBS and mounted on a glass slide in approximately 50 µl of Vectashield and covered with a #1.5 thickness 22 × 22 mm cover glass (Corning Life Sciences, Durham, North Carolina, USA; Cat# 2850–22,) or a #1.5 thickness 22 × 40 mm cover glass (Thermo Fisher Scientific Cat# CLS-1764-2240).

Immunofluorescence using anti-CDH23 and anti-PCDH15 antibodies was performed as outlined previously [98]. Briefly, ears were fixed in 4% formaldehyde (Electron Microscopy Sciences Cat# 1570) in dissection buffer for 20 to 30 min at room temperature then washed twice in calcium/magnesium-free HBSS (Thermo Fisher Scientific Cat# 14-175-095m). Tissue was blocked for 2 h in 10% normal donkey serum (Jackson ImmunoResearch Cat# 017-000-121) diluted in calcium/magnesium-free HBSS with 2 mM EDTA, and this blocking buffer was used for all antibody steps. Tissue was incubated overnight at 4˚C with 1:100 anti-CDH23 and 1:200 anti-PCDH15, washed 3 times with calcium/magnesium-free HBSS, then incubated for 2 to 3 h at room temperature with 2 µg/ml donkey anti-goat Alexa Fluor 488 (Thermo Fisher Scientific Cat# A21206), 2 µg/ml donkey anti-rabbit Alexa Fluor 568 (Thermo Fisher Scientific Cat# A10037), and 1 U/ml CF405 phalloidin (Biotium Cat# 00034). Tissue was washed 3 times in calcium/magnesium-free HBSS and mounted on a glass slide in approximately 50 µl of Vectashield and covered with a #1.5 thickness 22 × 22 mm cover glass (Corning Cat# 2850–22).

### Rhodamine-actin labeling

Labeling of free barbed ends of actin filaments was performed as outlined [43]. Inner ears from C57BL/6J mice or mutant mice littermates dissected at the indicated ages in cold Hank's balanced salt solution (Thermo Fisher Scientific Cat# 14025076) supplemented with 5 mM

HEPES (pH 7.4) (dissection buffer). Cochleas were dissected and lateral wall, Reissner's membrane, and tectorial membrane were removed. Live tissue was then briefly washed in cytoskeletal buffer (20 mM HEPES (pH 7.4), 138 mM KCl, 4 mM $MgCl_2$, 3 mM EGTA, 1% bovine serum albumin (Sigma-Aldrich Cat# A3803), 2 mM ATP (Sigma-Aldrich Cat# 10127531001), and 0.05% saponin (Sigma-Aldrich Cat# S7900)) before incubating for 5 min at room temperature in 40 ng/µl rhodamine-labeled actin (Cytoskeleton, Denver, Colorado, USA; Cat# AR05) freshly prepared in cytoskeletal buffer. Samples were then washed in cytoskeletal buffer and fixed for 1 h at room temperature in 4% formaldehyde (Electron Microscopy Sciences Cat# 1570) in PBS. Tissues were rinsed twice in PBS, then incubated for 1 h at room temperature in 0.4 U/ml Alexa Fluor 488 phalloidin (Thermo Fisher Scientific Cat# A12379) diluted in PBS with 0.05% saponin. Tissue was washed 3 times in PBS and mounted on a glass slide in approximately 50 µl of Vectashield and covered with a #1.5 thickness 22 × 40 mm cover glass (Thermo Fisher Scientific Cat# CLS-1764-2240).

## Fluorescence microscopy

Structured illumination (SIM) images were acquired at 26 to 30˚C with a 63× 1.4 NA oil immersion lens on a Zeiss (Oberkochen, Germany) lattice-based Elyra 7 microscope with dual PCO.edge 4.2 sCMOS cameras for detection. Illumination grid selection and z-spacing was guided by the software and kept consistent across images. Post-acquisition processing was performed with software-recommended standard filtering for the 488-nm channel, without baseline subtraction and with "scale to raw" checked. Contrast was manually adjusted to retain both dim and bright structures due to the high dynamic range of the phalloidin signal. Verification of channel alignment and measurement of the microscope point-spread function was carried out as previously described [50].

Airyscan images were acquired at room temperature using a 63×, 1.4 NA Plan-Apochromat objective on a Zeiss 3-channel LSM980 laser-scanning confocal microscope equipped with an Airyscan.2 detector and run under ZEISS ZEN (v3.1, 64-bit software; Zeiss) acquisition software. Settings for x-y pixel resolution, z-spacing, as well as pinhole diameter and grid selection, were set according to software-suggested settings for optimal Nyquist-based resolution. Raw data processing for Airyscan-acquired images was performed using manufacturer-implemented automated settings. Display adjustments in brightness and contrast and reslices and/or maximum intensity Z-projections were made in Fiji software (https://imagej.net/software/fiji/).

For cochlea imaging, for each stain, 2 to 4 images were acquired from 1 to 2 cochlea per genotype per age for each experiment, and experiments were conducted 3 to 4 times. Ears from control and mutant littermates or from different ages of C57BL/6J mice (of both sexes) were stained and imaged on the same days for each experiment to limit variability. Genotyping was performed either prior to dissection or performed on tails collected during dissection for younger animals (<P8). Genotypes were known by the experimenter during staining and image acquisition. During image acquisition, the gain and laser settings for the antibody and phalloidin signals were adjusted to reveal the staining pattern in control samples, and the corresponding KO samples used the same settings. Image acquisition parameters and display adjustments were kept constant across ages and genotypes for every antibody/fluorophore combination.

## Image rendering and measurements of stereocilia dimensions

Surface rendering was carried out with Imaris software (imaris.oxinst.com; versions 9.7.0, 9.8.0, 9.9.0; Oxford Instruments, Abingdon, UK); the rendered surfaces were stored as sparse

voxel octrees [99], which allowed them to be manipulated and measured using Imaris. A batch processing file was created in Imaris and applied to each dataset. Within each batch, background subtraction was first performed in the image processing tab using a filter width of 1 μm for the phalloidin channel for all images. Volumetric surface areas of the stereocilia in each image were then created from the phalloidin channel using creation parameters guided by the program and edited as needed so that the phalloidin signal for each stereocilium was accurately surrounded by a surface. Due to the high dynamic range of the signal, and the low intensity of the signal in control row 3 or 4 stereocilia, creation parameters were guided to create the most accurate surfaces for rows 1 and 2, which often left some low intensity rows 3 or 4 stereocilia undetected. Manual cutting of any connected stereocilia surfaces was performed so that each stereocilium was modeled by a unique surface. For any visible stereocilia lacking a surface (most often in rows 3 or 4), a surface was created using the magic wand tool in the Create tab of Imaris. If the intensity of the stereocilium was too similar to the phalloidin intensity of the surrounding cuticular plate, a surface could often not be generated. Thus, rows 3 and 4 stereocilia are only sparsely represented in older control hair bundles. Stereocilia surfaces were labeled as corresponding to each stereocilia row (1, 2, 3, or 4+) and as corresponding to each cell in the image. Measurements of Volume, BoundingBoxOO (object-oriented) Lengths, Number, and Position (x,y) for all labeled surfaces were exported from Imaris to Microsoft Excel. BoundingBoxOO lengths are generated within Imaris by considering the minimal rectangular box that fully encloses each object with no constraints to the orientation; the dimensions of that box are reported as Lengths A, B, and C, with A being the shortest principal axis and C the longest principal axis. To measure length and width of each stereocilium, object-oriented BoundingBoxOO Length C and the average of lengths A and B were used, respectively. For volume, we used the surface volume measurement provided by Imaris, which quantifies the amount of space each object occupies. The number of stereocilia and the sum of stereocilia volume per cell were calculated in Excel from the Imaris output. To measure the center 10 stereocilia, 10 neighboring row 1 and row 2 stereocilia at the center of each bundle were selected within Imaris, and measurements for this subset were exported to Excel. Position (x,y) data could be used to determine neighboring stereocilia and calculate differences in dimensions for adjacent stereocilia within each row or adjacent rows 1 and 2 stereocilia (Fig 5).

Three-dimensional rendering of stereocilia combined with EPS8 and GPSM2 tip staining from Airyscan z-stack images (Figs 11O–11R and 12M–12P) was also performed in Imaris using similar methods. Background subtraction was first performed in the image processing tab using a filter width of 1 μm for all channels for all images. Volumetric surface areas of the stereocilia in each image were then created from the phalloidin channel using creation parameters guided by the program and edited as needed so that the phalloidin signal for each stereocilium was accurately surrounded by a surface. Volumetric surface areas of the EPS8 or GPSM2 signals in each image were also created in their respective channels using creation parameters guided by the program and filtered to only include those with at least 0.01 overlap volume ratio with the phalloidin stereocilia surfaces, so that only EPS8/GPSM2 surfaces at stereocilia tips were included. All surfaces were manually edited to split any joining surfaces and create any surfaces that were missed for low-intensity phalloidin, EPS8, or GPSM2 signals. Once all surfaces were edited, the 2 sets of surfaces were merged and the phalloidin and EPS8 or GPSM2 surfaces corresponding to each individual stereocilium were joined in the edit tab, creating a surface that encompassed both the phalloidin signal and tip complex signals for each stereocilium. Each stereocilium surface was then labeled according to its row position. Thus, for each stereocilium, we measured both the length (BoundingBoxOO Length C) and the corresponding intensity sum for each channel in the image, with the channel 1 (568 nm) intensity sum measuring EPS8 amount and the channel 2 (488 nm) measuring GPSM2

amount. These measurements were exported to Excel (Microsoft, Redmond, Washington, USA), pooled, and then plotted within Prism (GraphPad Software, Boston, Massachusetts, USA).

## Quantitation of proteins at stereocilia tips

For Figs 9I–9K, 11A–11N, 12A–12L, 13M–13R and 14G–14H, quantitation of protein expression at rows 1 and 2 tips was carried out after importing Airyscan z-stacks into Fiji. For analysis of C57BL/6J mice throughout development, CAPZB and EPS8L2 in P21.5 mutants, and rhodamine-actin signal, summed z-projections of Airyscan stacks were made that included row 1 and row 2 tips in the same projection. For samples in which rows 1 and 2 were in separate Z planes, separate Z-projections were made for row 1 and row 2 tips. Regions of interest (ROIs) were selected at 10 (or more) row 1, and 10 (or more) row 2 tips per hair bundle. ROIs were circles that encompassed most of each tip. Using Fiji's Measure function, the area and mean gray value were measured for each ROI; measurements were also made outside the stereocilia and above the epithelium to determine background. The total fluorescence signal in a tip (tip signal) was thus area • mean gray value • minus total signal for the background. Tip signals below the background were assigned the value of 0, which were used in calculating averages.

For analysis of mutant mice in Figs 11, 12, S4 and S5, due to increased length variability, ROIs were selected at row 1 or row 2 tips within individual x-y slices from the z-stacks. Images were kept as multichannel stacks and phalloidin was used to guide selection at the tip of each stereocilium. For these images, all rows 1 and 2 stereocilia were measured within each cell. Measurements and calculation of tip signal were otherwise carried out as above. Mean intensities were calculated as the average tip intensity for all row 1 and row 2 stereocilia within each cell. Mean intensities were normalized to the average tip intensity for all rows 1 and 2 stereocilia from all het cochlea imaged on the same day. Histograms were created in Prism using frequency distributions of the number of values, with a bin width of 0.2 μm. The distributions were fit with a least squares regression for a sum of 2 Gaussians.

## Scanning electron microscopy

For scanning electron microscopy, periotic bones with cochleas were dissected in Leibovitz's L-15 medium (Thermo Fisher Scientific Cat# 21083–027) from P8.5 control and mutant littermates from mutant crosses. To provide access for fixative solutions, several small holes were made in periotic bones; the still-encapsulated cochleas were fixed for an hour in 2.5% glutaraldehyde in 0.1 M cacodylate buffer (Electron Microscopy Sciences Cat# 15960) supplemented with 2 mM $CaCl_2$. Cochleas were washed with distilled water, the cochlear sensory epithelium was dissected out, and then, the tectorial membrane was removed manually. Cochlear tissues were dehydrated in an ethanol series and critical-point dried using liquid $CO_2$ with a Leica (Wetzlar, Germany) EM CPD300. After immobilization on aluminum specimen holders using carbon tape, specimens were sputter coated with 3 to 4 nm of platinum (Leica EM ACE600). Samples were imaged using a Helios Nanolab 660 DualBeam Microscope (FEI).

## Data presentation and statistical analysis

For quantitation of length, width, and adjacent stereocilia length differences in Fig 1, we derived the data from the center 10 stereocilia from each row. Volumes were instead calculated from Imaris volume measurements from all stereocilia in each hair bundle. The data were derived from 2 litters per time point, 2 cochleas per litter, 1 image per cochlea, and 3 to 6 cells per image; thus *n* (the number of cells) was 12 to 24, and *N* (the number of cochleas examined)

was 4. Each point represents the mean ± standard deviation for all cell averages from all cochleas (i.e., *n* samples). Blinding of sample age was not possible as the age was obvious from the bundle and tissue morphology.

For quantitation in Figs 3–8, we compared measurements of dimensions between heterozygote and knockout hair cells. We carried out volume, length, and width measurements for each stereocilium, and then derived adjacent stereocilia length and volume sum measurements. Blinding of samples was not possible as the mutant phenotypes are easily discerned. To avoid problems with pseudoreplication [57,58,100], we considered the 5 hierarchical levels in our datasets—litter, animal, cochlea, cell, and stereocilium. The 2 cochleas from each animal were analyzed independently; we did not track individual animals. Cochlea mounting, intensity of labeling reagents, distance from the coverslip (potentially leading to spherical aberration and decrease in signal), and exact position along the apical-basal gradient all substantially affected the measurements and had a greater influence than animal or litter. For length, width, and volume measurements, measurements from all stereocilia of each row were averaged, providing a single measurement per cell. For these figures, all data were derived from 2 to 3 litters per genotype, 1 to 2 cochleas per litter, 2 images per cochlea, 4 total cochlea images, 5 to 6 cells per image, all rows 1 and 2 stereocilia for each hair cell, and some row 3 stereocilia for each hair cell. For each condition, *n* was 23 to 24 and *N* was 4. For length and width (and their corresponding CVs), we used the nested (hierarchical) *t* test function in Prism to compare the results from heterozygote and knockout cochleas; while comparing the results from the 4 cochleas per condition, the nested *t* test approach takes into account the structure of the data, i.e., the variance in individual cell measurements for each condition [58]. For comparisons of dimensions of *Pcdh15*$^{av3J}$;*Myo15a*$^{sh2}$ genotypes (Fig 6), as well as volume measurements for mutants and C57BL/6 (Fig 8), we used the nested one-way ANOVA function in Prism, adjusting for multiple comparisons using the Šidák correction [101].

For comparisons of intensity of row 1 tip proteins (Figs 11, 12 and S4), as well as rhodamine-actin labeling at rows 1 and 2 tips (Fig 13) and row 2 protein intensities (Fig 14), we used nested *t* tests as above. For comparisons of intensity of row 1 tip proteins in *Pcdh15*$^{av3J}$;*Tmie*$^{KO}$ genotypes (S5 Fig), we used ordinary one-way ANOVA tests with the Tukey correction for multiple comparisons.

## Supporting information

**S1 Fig. Variability in stereocilia length and width measurements.** (**A**) CVs for stereocilia length from all stereocilia at indicated ages. Blue lines in panels A–D show developmental trends for row 1 length or width CV. (**B**) CVs for stereocilia length from center 10 stereocilia at indicated ages. *P* values from unpaired Student's *t* tests comparing all row 1 stereocilia versus center 10 row 1 stereocilia: P0.5, 0.1861; P1.5–P21.5, <0.0001. *P* values from unpaired Student's *t* tests comparing length CVs for all row 2 stereocilia versus center 10 row 2 stereocilia: P0.5, 0.4933; P1.5–P21.5, <0.0001. (**C**) CVs for stereocilia width from all stereocilia at indicated ages. (**D**) CVs for stereocilia width from center 10 stereocilia at indicated ages. *P* values from unpaired Student's *t* tests comparing length CVs for all row 1 stereocilia versus center 10 row 1 stereocilia: P0.5, 0.0473; P1.5, <0.0001; P2.5, 0.0893; P3.5–P4.5, <0.0001; P5.5, 0.0001; P7.5, <0.0001; P10.5, 0.0038; P13.5, 0.0823; P16.5, 0.0126; P21.5, 0.118. *P* values from unpaired Student's *t* tests comparing all row 2 stereocilia versus center 10 row 2 stereocilia: P0.5–P21.5, <0.0001. (**E–G**) Models of stereocilia arrangement. A peripheral-to-central gradient of stereocilia length for row 1 and row 2 was seen at P7.5 (F; red gradient symbol), rather than uniform lengths (E; red line). Stereocilia lengths largely equalized by P21.5 (G). The data underlying all the graphs shown in the figure can be found in figshare (https://doi.org/10.6084/m9.figshare.

21632636.v2).
(TIF)

**S2 Fig. CDH23 and PCDH15 immunoreactivity in mouse tip-link mutants.** (**A, B**) CDH23 and PCDH15 localization in $Cdh23^{v2J}$/+ (A) and $Cdh23^{v2J/v3J}$ (B) IHCs. (**C, D**) CDH23 and PCDH15 localization in $Pcdh15^{av3J}$/+ (C) and $Pcdh15^{av3J/av3J}$ (D) IHCs. Panel widths: 30 μm.
(TIF)

**S3 Fig. Localization of TMC1-HA and TMC2-Myc in $Pcdh15^{av3J/av3J}$ mutant IHCs.** (**A**) TMC1-HA in P7 $Tmc1^{HA/HA}$;$Pcdh15^{av3J}$/+ cochlea at the hair-bundle level. Inset shows labeling in rows 1 and 2. (**B**) TMC1-HA in same P7 $Tmc1^{HA/HA}$;$Pcdh15^{av3J}$/+ cochlea at the soma level. (**C**) TMC1-HA in P7 $Tmc1^{HA/HA}$;$Pcdh15^{av3J/av3J}$ cochlea (bundles). (**D**) TMC1-HA in same P7 $Tmc1^{HA/HA}$;$Pcdh15^{av3J/av3J}$ cochlea (somas). (**E**) TMC2-Myc in P4 $Tmc1^{Myc/Myc}$; $Pcdh15^{av3J}$/+ cochlea (bundles). Inset shows labeling in rows 1 and 2. (**F**) TMC2-Myc in P4 $Tmc1^{Myc/Myc}$;$Pcdh15^{av3J/av3J}$ cochlea (bundles). Panel widths: 35 μm (insets, 7 μm).
(TIF)

**S4 Fig. Localization of GNAI3 and WHRN in $Cdh23^{v2J/v2J}$ and $Pcdh15^{av3J/av3J}$ mutant IHCs.** (**A–L**) Localization of row 1 complex proteins in $Cdh23^{v2J}$ and $Pcdh15^{av3J}$ heterozygotes and homozygotes IHCs. Panel widths: 30 μm. (**A, B**) GNAI3 and EPS8 in $Cdh23^{v2J}$/+ and $Cdh23^{v2J/v2J}$ at P7.5. (**C, D**) WHRN and EPS8 in $Cdh23^{v2J}$/+ and $Cdh23^{v2J/v2J}$ at P7.5. (**E, F**) GNAI3 and EPS8 in $Cdh23^{v2J}$/+ and $Cdh23^{v2J/v2J}$ at P21.5. (**G, H**) WHRN and EPS8 in $Cdh23^{v2J}$/+ and $Cdh23^{v2J/v2J}$ at P21.5. (**I, J**) GNAI3 and EPS8 in $Pcdh15^{av3J}$/+ and $Pcdh15^{av3J/av3J}$ at P21.5. (**K, L**) WHRN and EPS8 in $Pcdh15^{av3J}$/+ and $Pcdh15^{av3J/av3J}$ at P21.5. (**M–P**) GNAI3 and WHRN normalized fluorescence average intensity per hair bundle for all measured stereocilia (rows 1 and 2) for $Cdh23^{v2J}$ and $Pcdh15^{av3J}$ IHCs at P7.5 and P21.5, respectively. Intensities were normalized to the heterozygote average for each genotype pair. Plotting and statistical testing were as in Fig 3. (**Q–T**) Frequency distribution of GNAI3 and WHRN tip intensity in $Cdh23^{v2J}$ hair cells at P7.5 and P21.5. The data underlying all the graphs shown in the figure can be found in figshare (https://doi.org/10.6084/m9.figshare.21632636.v2).
(TIF)

**S5 Fig. $Pcdh15^{av3J}$;$Tmie^{KO}$ double-knockout mice.** (**A–D**) Scanning electron micrographs showing P8.5 IHC hair bundles from $Pcdh15^{av3J}$/+;$Tmie^{KO}$/+ (Het;Het) $Pcdh15^{av3J}$/+;$Tmie^{KO/KO}$ (Het;KO), $Pcdh15^{av3J/av3J}$;$Tmie^{KO}$/+ (KO;Het), and $Pcdh15^{av3J/av3J}$;$Tmie^{KO/KO}$ (KO;KO) cochleas. (**E–H**) Localization of GNAI3 and EPS8 in IHCs of $Pcdh15^{av3J}$;$Tmie^{KO}$ genotypes at P7.5. (**I**) EPS8 fluorescence average intensity per bundle for all measured stereocilia in bundles of each genotype. (J) Frequency distribution of EPS8 tip intensity in bundles of each genotype. Het;Het and Het;KO distributions were fit with double Gaussians; KO;Het and KO;KO distributions were fit with single Gaussians. (**K**) GNAI3 fluorescence average intensity per bundle for all measured stereocilia in bundles of each genotype. (**L**) Frequency distribution of GNAI3 tip intensity in bundles of each genotype. Distributions from each genotype were fit with double Gaussians. For statistical comparisons in I and K, we used ordinary one-way ANOVA tests with the Tukey correction. The data underlying all the graphs shown in the figure can be found in figshare (https://doi.org/10.6084/m9.figshare.21632636.v2).
(TIF)

## Acknowledgments

We thank Jennifer Goldsmith for mouse husbandry and Michael Bateschell for laboratory support. John Brigande and Ulrich Müller provided helpful comments on the manuscript. We

utilized the following core facilities: lattice SIM, confocal microscopy, and Imaris software access from the OHSU Advanced Light Microscopy Core at The Jungers Center; electron microscopy from the OHSU Multiscale Microscopy Core.

## Author Contributions

**Conceptualization:** Jocelyn F. Krey, Benjamin J. Perrin, Peter G. Barr-Gillespie.

**Data curation:** Peter G. Barr-Gillespie.

**Formal analysis:** Jocelyn F. Krey, Paroma Chatterjee, Julia Halford, Christopher L. Cunningham, Benjamin J. Perrin, Peter G. Barr-Gillespie.

**Funding acquisition:** Peter G. Barr-Gillespie.

**Investigation:** Jocelyn F. Krey, Paroma Chatterjee, Julia Halford, Christopher L. Cunningham, Benjamin J. Perrin.

**Methodology:** Jocelyn F. Krey, Christopher L. Cunningham, Benjamin J. Perrin.

**Project administration:** Peter G. Barr-Gillespie.

**Supervision:** Peter G. Barr-Gillespie.

**Validation:** Jocelyn F. Krey, Peter G. Barr-Gillespie.

**Visualization:** Peter G. Barr-Gillespie.

**Writing – original draft:** Jocelyn F. Krey, Peter G. Barr-Gillespie.

**Writing – review & editing:** Jocelyn F. Krey, Paroma Chatterjee, Julia Halford, Christopher L. Cunningham, Peter G. Barr-Gillespie.

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
