## [Editor Report · Decision Letter 0]

14 Dec 2022

Dear Peter, 

Thank you for submitting your manuscript entitled "Control of stereocilia length during development of hair bundles" for consideration as a Research Article by PLOS Biology.

Your manuscript has now been evaluated by myself in discussion with the other PLOS Biology editorial staff, as well as by an academic editor with relevant expertise. I am writing to let you know that we would like to send your submission out for external peer review as a Resource article.

Before we can send your manuscript to reviewers, we need you to complete your submission by providing the metadata that is required for full assessment. To this end, please login to Editorial Manager where you will find the paper in the 'Submissions Needing Revisions' folder on your homepage. Please click 'Revise Submission' from the Action Links and complete all additional questions in the submission questionnaire. Please also make sure to update the file type to "Resource", and I welcome any reviewer suggestions you might have. We also allow up to three exclusions if you feel these are needed.

Once your full submission is complete, your paper will undergo a series of checks in preparation for peer review. After your manuscript has passed the checks it will be sent out for review. To provide the metadata for your submission, please Login to Editorial Manager (https://www.editorialmanager.com/pbiology) within two working days, i.e. by Dec 16 2022 11:59PM.

Feel free to email us at plosbiology@plos.org, or to email me directly, if you have any queries relating to your submission. And my apologies that it took this long to get you this first pass decision. Many people seem to already be heading off for holidays and that's been delaying our discussions with Academic Editors.

With best wishes,

Kris

Kris Dickson, Ph.D., (she/her)

Neurosciences Senior Editor/Section Manager

PLOS Biology

kdickson@plos.org

---

## [Decision Letter · Decision Letter 1]

12 Feb 2023

Dear Dr Barr-Gillespie,

Thank you for your patience while your manuscript "Control of stereocilia length during development of hair bundles" went through peer-review at PLOS Biology. Your manuscript has now been evaluated by the PLOS Biology editors, an Academic Editor with relevant expertise, and by several independent reviewers.

In light of the reviews, which you will find at the end of this email, we are pleased to offer you the opportunity to address the comments raised by the reviewers in a revision. While the reviewers provide quite a number of suggestions for improvements to the manuscript, much of this relates to improving the interpretations and explanations so we anticipate the revision should not take you very long. When the revision comes back in, we will assess your revised manuscript and your response to the reviewers' comments with our Academic Editor aiming to avoid further rounds of peer-review, although we cannot rule out a need to consult with the reviewers when the revision comes back in.

We expect to receive your revised manuscript within 1 month. Please email us (plosbiology@plos.org) if you have any questions or concerns, or would like to request an extension of this deadline. 

**IMPORTANT - SUBMITTING YOUR REVISION**

*Resubmission Checklist*

*Published Peer Review*

*PLOS Data Policy*

*Blot and Gel Data Policy*

Sincerely,

Kris

Kris Dickson, Ph.D., (she/her)

Neurosciences Senior Editor/Section Manager

PLOS Biology

kdickson@plos.org

REVIEWS:

Do you want your identity to be public for this peer review?

Reviewer #1: No

Reviewer #2: No

Reviewer #3: No

Reviewer #1: The development of the precisely organized mechanosensory stereocilia bundles in the inner ear hair cells is still a puzzling problem of cell biology. This manuscript provides comprehensive and meticulous quantification of the stereocilia growth in postnatal development in wildtype mice and mouse mutants lacking either only mechanotransduction or also tip links gating the mechanotransduction channels. These developmental data are correlated with the expression of the proteins that are known to be essential for the development of the proper shape of the hair cell bundle. Even though a similar study has been recently published by the authors (Krey et al., Curr. Biol., 2020), the extended set of data provided in the current manuscript reveal new roles of tip link proteins and allow building the molecular model of hair bundle development.

Specific comments:

Line 110: Could the exact location of the "apical IHCs" be better specified here (not only in Methods)?

Legend to Fig. 1H: Fit line for the row 2 data peaked at ~P7.5 and then decline. This cannot be a "fit with an exponential climb". 

Fig. S1E is confusing - the hair bundle is never "flat headed" at P7.5 as it is depicted in this cartoon. Maybe, the authors meant to show the shape of the bundle at P0.5? Then, still, the panel A data demonstrate significant variability in the lengths of stereocilia within the row 1. Please, explain or correct.

Line 195: Data in Fig. 1N seem to contradict the data in Fig. S1B. Fig. S1B shows that the variability of stereocilia lengths within row 2 is larger than that in row 1 from ~P4.5 onward. In contrast, panel N in the main Fig. 1 shows no difference in the length variability between the rows. The row 1 vs 2 difference might be an important observation for the mechanisms of row 2 regulation and, therefore, it is worth clarifying. 

Lines 205-206: I think the statement "…while phenotypes specific for tip-link mutants arise from the absence of the links themselves" is valid only partially. The correct statement would be "…while phenotypes specific for tip-link mutants arise from the absence of the links themselves and/or the lack of signaling from intracellular domains of PCDH15 and/or CDH23".

Legend to Fig. 2: "Some links were still present in both Cdh23v2J/v2J and Pcdh15av3j/av3J bundles, but were not located at the apices of a stereocilia tips". This is misleading since there are plenty of tip link-like structures in the panel Eii (right in the middle of the bundle). They seem to be absent in Pcdh15av3j/av3J mutants, which may be consistent with the different roles of PCDH15 and CDH23 in formation of the transient links. In any case, the quoted statement above requires either clarification or the exact definition of "the apices of a stereocilia tips" and proper quantification.

Lines 232-233: "…consistent with tip links causing membrane tenting and increased actin polymerization". Well, beveled row 2 tips are also observed after blocking transduction channels and retraction of row 2 stereocilia (Velez-Ortega et al., 2017). Thus, it is not clear whether the beveled row 2 tips in TmieKO/KO IHCs are caused by the increased actin polymerization around the lower end of the tip link or by increased disassembly of the peripheral actin filaments (which would be more consistent with the thinning of row 2 stereocilia reported in Fig. 4E). In the latter case, mechanical tension or some signaling associated with PCDH15 may protect the filaments surrounding the lower end of the tip link. 

Fig. 3: Presumably, all mutant hair cells (Cdh23v2J/v2J, Pcdh15av3J/av3J, and TmieKO/KO) develop without functional mechanotransduction. Yet, only TmieKO/KO hair cells (as well as other transduction mutants, see Krey at all., 2020) show the mis-localization of the row 1 complex and retarded growth of row 1 stereocilia at P21.5. How is it possible, if TMC1 is also missing in the tip link mutants (at least in Pcdh15av3J/av3J mutants, Fig. S3C)?

Fig. 5, panels D & F: I cannot understand why the coefficients of variation for delta lengths (bottom graphs) are similar between wild type and Cdh23v2J/v2J and Pcdh15av3J/av3J mutants, while the spread of the same raw data (upper panels) is visually different. In general, it is very hard to understand this convoluted way of presenting relatively simple differences in the shapes of different bundles. Why the same data cannot be presented as a collection of graphs, which would show the length of individual stereocilium (Y-axis) relative to its position within a row (X-axis)? Two-way ANOVA would be useful for statistical analysis, while the source of within-the-row variations would become visually obvious.

Lines 340-341: "…perhaps because the minimal staircase prevented tip links from sufficiently tenting membranes off the tips of the shorter rows". Alternatively, the tenting may be harder to generate when the link pulls the membrane at the wrong place, on the side of a stereocilium instead of the tip.

Fig. 6: Do the double mutant IHCs loose all their links between stereocilia? SEM images and/or link quantification would be helpful here.

Fig. 8: This is great! Very nice piece of data.

Line 628: You meant "…during stage III, when lengthening is arrested"?

Line 836: Could the authors provide SEM evidence for the tip links running "to the end of the track"? The links connecting the tips of the adjacent stereocilia in a column should be quite prominent.

Lines 843-862: I would refrain from speculating on the mechanisms of resorption of supernumerary stereocilia. The manuscript data are focused on row 1 and 2 and formation of the staircase. There is only very limited amount of data regarding rows 3 and 4+ and, therefore, these speculations are not yet supported by the data.

Lines 884-886: "More randomized allocation of row 1 complex members and the varying positions of stereocilia across the lateral-medial resource gradient may also create length variability". This statement seems contradicting the idea of the control of length variability through lateral PCDH15 and CDH23 links.

Lines 904-906: Then, it should be possible to prevent stage III and force stereocilia to grow without widening by blocking the transduction channels. Is it true? 

Line 1078: Remove extra "E".

Reviewer #2: * we uploaded a formatted version of the comments below (.doc) and a summary diagram *

The author's objective is to describe P7-P21 stereocilia development in IHCs at the cochlear apex and to investigate structural and molecular defects in 3 categories of mutants that were shown to affect elongation. This work was started in Krey 2020 [6], where focus was on transduction mutants and the role of Ca++ in shaping stereocilia dimensions.

In this new work, the main difference is the inclusion of "tip-link" mutants (that in fact also disrupt transient lateral and kinocilial links), the use of an improved microscopy method (lattice SIM; allows for example to address elusive row 3 stereocilia), surface rendering and an (even more) exacting approach with more hair cells and stereocilia data points measured in a finer time course. As the work endeavors to sum up a lot of results and to serve as current reference in the field, it is fitting for a Methods and Resources article. These methods are further applied to an interesting budding question: how do mechanotransduction elements influence stereocilia morphogenesis? The work is very well executed and sets a high standard for rigorous and thorough methods, quantification, and data reporting.

The next challenge, however, is to present so many results, often disparate, as a coherent whole, and chiefly to direct the reader towards more notable new results that are solid while acknowledging and nuancing those that remain more uncertain. This aspect could be improved in a revised version that could perhaps be shorter as well. We feel that the unusually large body of data is challenging to navigate, hiding some gems while alternatively devoting much space to a few conclusions that are not necessarily fully supported by evidence here or in Krey 2020.

In short (further detailed below), we find that several claims about the behavior of the actual row 1 complex (GPSM2-GNAI3 and WHRN; see point B2 below) are not fully convincing, with caveats that call for language to be revised to better reflect evidence reported here, in Krey 2020, and in other referenced papers. Interestingly, this comment holds particularly true for the Introduction and Results, whereas by contrast the Discussion presents a fairer and more balanced summary of results in general, although with some degree of repetition. Discussion positions should be adopted in the rest of the manuscript, as readership is likely to focus chiefly on Intro and Results first.

The detailed comments below are made in hopes that they can help the authors improve the manuscript, notably A) better highlight some remarkable results and conclusions while B) avoid possibly hasty conclusions about the behavior of the row 1 complex. We are not requesting additional or revised experiments, except if the authors want to maintain currently less-well supported claims (see below)

A) RESULTS THAT ARE UNDER-STATED

A1) From the onset, the main expectation for this paper seems to be a close comparison of two categories of mutants that both abolish transduction, allowing the authors to tease apart the specific role of MET channel proteins or fibrous links from the process of transduction itself.

This is made explicit in the text early in the manuscript:

Abstract Line 30:"Use of the tip-link mutants also allowed us to distinguish the role of transduction from effects of transduction proteins themselves."

Line 204: "Phenotypes shared by tip-link and transduction mutants most likely result from the loss of transduction, while phenotypes specific for tip-link mutants arise from the absence of the links themselves.". 

Surprisingly, however, this angle is somewhat abandoned en route, even though remarkable and complete results are obtained and detailed on this front.

In short, this work shows that the two phenotypes are interestingly distinct from one another. "Tip-link" mutants have overall normal row 1 lengths and tip complex but variable lengths in each row and disarrayed stereocilia placement (packing issues), likely through lack of transient and kinocilial links. In contrast, transduction protein mutants have shorter row 1 lengths and reduced row 1 complex, but more even stereocilia lengths within rows; they also have a unique extra rows phenotype classic of "mature row 1" complex mutants (Myo15a etc.). 

What appears to be shared are the following defects:

- row 2 width reduction (Fig. 4) 

- increase in row 3-4 volume (Fig. 7), or a shared defect in pruning

- reduced free barbed ends in row 2 at P7 (Fig. 13)

This somewhat limited overlap is surprising and novel, and is thus expected to be at the core of the paper's Discussion and summary diagram in Fig. 15 - but is not. To try to illustrate this point, we attached a rough diagram of what we would have expected to (also) see in a graphical summary.

For one, these results suggest that MET function itself (Ca++) plays a lesser role in stereocilia morphogenesis compared to the model proposed previously (Krey 2020, Velez-Ortega 2017). Instead, links (transient, kinocilial) are paramount yet distinct in function from the presence of transduction protein at row 2-3 tips. 

In Krey 2020, blocking MET at early stages with tubocurarine yielded extra stereocilia rows, yet largely normal row 1 complex enrichment. Consequently, extra rows appear to result from ablating MET function when transduction proteins are present, so why do "tip-link" mutants lack extraneous stereocilia rows despite lacking transduction? This conundrum deserves acknowledgement and discussion. It would also be interesting to speculate on the "other" role of transduction proteins besides enabling transduction. 

A2) Other solid and novel results are discussed but may benefit from being further highlighted. The long Discussion seems to show some redundancy with some themes "echoed" back a few times - consolidating might be more effective (see also Minor points list).

These interesting and novel results are for example:

- First formalized report that peripheral stereocilia are initially shorter compared to central ones.

- Dynamic fit allowing to distinguish different cross-sectional area increase regimes (linear before P8, exponential afterwards)

- Total stereocilia volume constant in all mutants. Interesting support for the idea that hair cell "building blocks" remain constant and thus stereocilia dimensions are defined by how building blocks are put together, not their availability.

- Proposed mechanism to coordinate lengths in the same row (links; but kinocilial link contribution to packing issues for example is not highlighted)

- Absence of TMC1 (but not TMC2) in Pcdh15 KO

- Free barbed ends largely lost at P21 but equally retained in the 3 mutant categories- is it bundle immaturity?

- Missing discussion of compound mutants in Figure 6 and S5. Interesting but significance remains unclear.

B) OVERSTATED CONCLUSIONS 

B1) Conclusions about "stereocilia development" are strictly based on select stages of IHCs at the cochlear apex.

P0.5-P21.5 IHC behavior at the cochlear apex, in both normal and mutant contexts, may not seamlessly transpose to earlier stages, other cochlear positions, or to outer hair cells. For example, could the sharp increase of row 1 length at P8.5, or the transition between linear and exponential cross-sectional areas also at P8.5 be specific to the apex? The Tilney model of phase I-IV would greatly benefit from having mouse stages associated not only for the apex, but also the cochlear base to help translate these results to other contexts.

For example, is row 1 length at the cochlear base stable during stage III as reported here at the apex? It may instead be reduced during phase III to reach the short statures characteristic of this region?

The row 1 complex is prominent in OHCs too starting at earlier embryonic stages, and it may behave differently than IHCs at the apex in this context.

B2) Assuming that EPS8 (and MYO15A) are part of a "row 1 complex" during development

Before P7.5 at least, if not later, EPS8 and MYO15A are clearly distributed in all rows, unlike GPSM2, GNAI3 and WHRN. This was reported by others, in Krey 2020, and is in fact even shown here [Fig 9B,F Fig 10C, G etc., Fig11D, H, L]. However this distinction is understated in the text of the Introduction and Results. Grouping EPS8-MYO15A and GPSM2-GNAI-WHRN together is too simplistic, especially for this rigorous work focusing on developmental stages where these proteins have distinct row distribution patterns.

Of note, the fact that GPSM2-GNAI3 but not MYO15A-EPS8 is reduced at row 1 tips in transduction mutants in this work is an independent clue that these proteins form in fact distinct modules.

Locations where this issue is problematic:

Line 50: "A complex of GPSM2 and GNAI3, coupled to MYO15A and EPS8 by WHRN, catalyzes lengthening of stereocilia beyond about a micrometer [7-9]; this complex concentrates exclusively in row 1.". 

In fact EPS8 and MYO15A are prominent in all rows notably row 2 until at least P7. This sentence makes a definitive statement that only applies to adults and should be edited.

Line 443: "As in C57BL/6 stereocilia, in heterozygotes from mutant IHCs, GNAI3, GPSM2, and EPS8 were concentrated at tips of row 1 stereocilia (Fig 10A-D, I-L)."

This is inexact, or at least misleading, since EPS8 is well visible at row 1 unlike the other proteins.

Line 446 "..but EPS8 was seen at all stereocilia tips in similar amounts". 

This formulation indirectly recognizes that EPS8 is normally at row 2 tips but in lower amounts, emphasizing the need to present it as such in controls before underscoring uniformity in mutants.

Figure 9. should point out row number to show that EPS8 is in all rows. Fig. 9B shows EPS8 at row 2 and so does Fig. 9F, P7.5. This could be made more clear in the images. It is notable that in Fig. 9, the next stage investigated after P7.5 is P15.5 where indeed EPS8 is very low at row 2. This minimizes the fact that EPS8 is enriched at row 2 (and 3+) from embryogenesis until probably ~P10 (Fig. 9I).

All through Fig 10, EPS8 at row 2 is quite obvious at P7.5 as well, and could be pointed as such. Best example in C, G but visible in other panels too.

On the quantitative side, frequency distribution representations in Fig. 11 clearly show substantial EPS8 enrichment at row 2 at P7.5 and possibly later (Fig. 11D, H, L; two bumps of tip intensity/average values).

Lines 486 and 499, the text refer to "two prominent peaks" and " two well separated peaks", respectively, without ever relating this observation to significant EPS8 signal at row 2 in controls.

Line 465: "Quantitation of row 1 tip complex proteins allowed us to determine the distribution of these proteins between row 1 and 2 stereocilia more thoroughly (Fig 11-12; S4 Fig)."

The cited figures include P7.5 where in fact EPS8 is notably enriched at row 2 in controls.

Line 484: "Frequency distribution plots, using normalized EPS8 intensities (signal at each tip divided by the total signal for all row 1 and 2 stereocilia measured in the hair bundle), revealed how EPS8 was uniformly targeted to stereocilia tips in tip-link mutants". 

Same comment as above line 446. This sentence makes conspicuous that the normal distribution of EPS8 in controls was actually not accurately described in the first place.

B3) Assuming GPSM2-GNAI3 is active at stage II and reduced at stage III (before an obvious increase at stage IV)

This manuscript does not show stage II results, if we assume from Figure 1 that P0 apex is already stage III. The model proposed in Figure 15E states that GPSM2 is present and required for stage II elongation. Indeed, P0 IHCs halfway through the basal turn were reported to have shorter row 1 stereocilia already (Tarchini 2016), and if previous GPSM2 activity at this stage and position qualifies as stage II (and not already as stage III ?) this conclusion is justified.

However, a reduction of GPSM2-GNAI3 or WHRN at row 1 tips after P0 and before P7 was not reported before, including in Krey 2020. We would argue it is not demonstrated here either. Krey 2020 showed a steady increase during stage III (Krey 2020; Fig. 3A-B), with clear GNAI3 enrichment at P2.5 and P4.5 at the cochlear apex. The authors would need to analyze earlier stages at the stage II-III transition in more details to make this claim (possibly including the cochlear base to assess how broadly applicable conclusions are). Is GPSM2-GNAI3 actually temporarily reduced at tips between stage II and IV? GPSM2-GNAI3 can be detected at P0 at the base in IHCs already, and would need to be reduced there by ~P2-P4 to claim a reduction during phase III. 

Another issue with the model proposed here is that previous publications show GPSM2 as closely enriched with GNAI3 at ~P2-P4 already (stage III), so it is surprising that GPSM2 would be downregulated at tips (more so than GNAI3) to then peak again during stage IV, but in a delayed manner compared to GNAI3 . 

An alternative explanation to the results in Figure 9 could be that the GNAI3 antibody used in this study is more sensitive than the GPSM2 antibody, picking up signal at tips more efficiently at P7.5. Or, the normalization process could possibly explain different peak times? See line 441-442, normalization method was different for GPSM2 and GNAI3.

Locations where this issue is problematic:

Abstract Line 23 "Tip proteins that determine row 1 lengthening did not accumulate simultaneously during stages III and IV; while the actin-bundling protein EPS8 peaked at the end of stage III, GNAI3 peaked several days later—in early stage IV—and GPSM2 peaked near the end of stage IV."

Line 98: "Here we defined with higher precision the changes in stereocilia dimensions during postnatal development of apical IHCs from postnatal day 0.5 (P0.5) to P21.5."

As mentioned above, protein labeling at P0-P7 is not tested with the same precision as P7 and older in mutant contexts, with fewer images and measurements (Fig. 9-11).

Line 427: "GNAI3 was relatively low during stage III, but increased to a peak at P10.5, near the beginning of stage IV"

This text does not capture well the graph results for GNAI3 in Fig. 9J. In fact, GNAI3 increases mostly during stage III. Krey 2020 reported similar results, with a large increase before P7.5.

Line 430: "Our results showing sequential accumulation of EPS8, then GNAI3, and finally GPSM2 at stereocilia tips are not consistent with the earlier suggestion that the row 1 tip complex assembles simultaneously [9]."

This sentence needs to be removed or heavily edited. First, see comment B2 above about the invalid "row 1 complex" to describe 5-protein complex during development. Second, the previous work cited (Tadenev 2019) actually already showed that EPS8 (and MYO15A) are detected at tips in all HCs before GPSM2-GNAI3, so this is not an accurate citation of the publication. Third, if Tadenev 2019 indeed proposed simultaneous enrichment of GPSM2-GNAI3, so did in fact the author's previous paper Krey 2020: "GPSM2 showed a nearly identical labeling pattern to that of GNAI3 (Figures S4A and S4B)."

This sentence also disregards that conclusions in Tadenev 2019 were made also including OHCs where GPSM2-GNAI3 are heavily enriched at tips during embryogenesis already; earlier and at higher levels than in IHCs. This is bringing back point B1 about how universal or not the dynamic changes observed in IHCs at the cochlear apex are.

Line 823: "GPSM2 and GNAI3 levels are low during stage III widening". 

Same comment as above. This work and Krey 2020 in fact show increased amounts from P0.5, which does not seem compatible with transient reduction at stage III.

Line 824" Towards the end of stage III, however, GNAI3 begins to accumulate at row 1 tips, reaching a peak during early stage IV". 

Same comment as for Line 427; does not seem correct in light of the graph, where GNAI3 is steadily raising and high in stage III already.

Line 907:"The row 1 complex reassembles during stage IV in a stepwise manner"

Same comment, there is no clear evidence for disassembly in stage III.

B4) Assuming GPSM2-GNAI3 presence at tips depends on TMIE (and TMC1/2)

Available evidence suggest that GPSM2 and GNAI3 require TMIE (and TMC1/2) to be durably maintained, and not enriched, at row 1 tips. This manuscript actually shows a progressive decline of GPSM2 and GNAI3: for example, reduced, yet obvious enrichment for GNAI3 at P7, and further reduced enrichment at P21 in Tmie KO. The same is shown in Krey 2020, including in Tmc1; Tmc2 double KO. GPSM2 and GNAI3 are not labeled before P7 in transduction mutants in either work as far as we can tell. However, language in the manuscript suggests in several locations (below) that GPSM2-GNAI3 presence at row 1 tips requires transduction proteins, proposing that TMIE and TMCs could be transiently present at row 1 and involved.

To support strict requirement, the authors would need to label GPSM2-GNAI3 at earlier postnatal stages in Tmie KO (i.e. when transduction is active and GPSM2-GNAI3 are clearly at tips already in controls, P0 base until P7). As far as we know, it cannot be ruled out that GPSM2-GNAI3 still have normal row 1 enrichment in transduction mutants at these earlier stages. The evidence of detectable GNAI3 at P7 in fact rules out strict requirement already.

 The reference cited to support the puzzling presence of transduction proteins at row 1 is Cunningham 2020. We note however that this publication does not make any claim about row distribution of the epitope-tagged TMIE and TMC1/2 fusion proteins. While images show broad (yet specific) foci across P3-P6 stereocilia shafts, these tagged proteins do not appear to be enriched at row 1 tips. In the work under review, Fig. S3 does not clarify row identity of TMC1-2 signals either. 

In addition, we note that at P7.5, row 1 length is only significantly reduced in sh2 but not in Tmie mutants (Fig. 3K,L). This is independent evidence that proteins are row 1 are not fully dependent on transduction for function. Similarly, at P21, length reduction is greater in sh2 compared to Tmie KO.

Finally, as GPSM2 and GNAI3 are highly enriched at OHC row 1 tips during embryogenesis, it is unlikely that transduction proteins are required in this cell type even if they were required in IHCs. This further underscores the difficulty of outlining a universal stereocilia differentiation program (point B1).

The following excellent sentence in the discussion should be the standard for modifications in the rest of the manuscript:

"In TmieKO/KO mutants, members of the row 1 complex are still present, but neither GPSM2 nor GNAI3 traffic efficiently to row 1 tips in the absence of transduction proteins, at least during stage IV."

Locations where these comments apply:

Line 71: "Although row 1 lacks transduction [22], in transduction mutants, GNAI3 and GPSM2 do not accumulate substantially at row 1 tips in transduction mutants [6]."

As GPSM2-GNAI have been reported at OHC tips during embryogenesis already, and were not investigated before P7.5 in this work or in Krey 2020, it would be more accurate to state that they are not maintained normally at tips in these mutants.

Line 446: " These results contrasted with those from TmieKO/KO IHCs, where GPSM2 and GNAI3 largely disappeared from hair bundles (Fig 10D, L);"

This fails to capture the fact that loss of enrichment is progressive with GNAI3 still well visible at P7. 5 for example. So maintenance is affected, specifically at least from the end of stage III.

Line 448: "; WHRN intensity was reduced, but remained most abundant in row 1 (Fig 10H)." 

In fact, WHRN at P7.5 has the same profile as GNAI3 which is instead described as largely missing in several places in the text: WHRN is reduced but still present at row 1 tips. The accurate description for WHRN should apply to GNAI3 as well. Summarizing GNAI3 as instead largely missing appears to be a bias.

Line 520: (referring to reduced row 1 proteins in Tmie but not tip-link mutants): "These results confirm that TMIE (and TMC1/2) affect distribution of GNAI3, GPSM2 and WHRN directly, not via transduction-mediated ion flux [6].

Also Abstract Line 33:"These results reinforced the suggestion that the transduction proteins themselves facilitate localization of proteins in the row 1 complex."

Also Discussion Line 740:"Our results here confirm our previous suggestion that transduction proteins themselves regulate the amount of GNAI3 and GPSM2 that accumulate at row 1 tips during stage IV [6]."

We are confused as to why this is not a new conclusion in this current study. Krey 2020 does not reach the conclusion that GPSM2-GNAI3 and WHRN do not depend on Ca++ or transduction. In Krey 2020, the possibility that transduction proteins themselves are needed is acknowledged, but results are largely interpreted and discussed as linked to Ca++. It would in fact be beneficial, and as far as we can tell, fair, to claim novelty for this important result here.

Line 521: "TMC2 still targeted to stereocilia in Pcdh15av3J/av3J 522 mutants, but TMC1 disappeared (S3 Fig), suggesting that TMC2 may be sufficient for localization of GPSM2, GNAI3, and WHRN to row 1 tips.."

The conclusion here does not seem valid. In Krey 2020, the authors showed reduced but still clearly enriched GNAI3 at tips in Tmc1, 2 DKO at P7.5. Consequently, as with TMIE, GPSM2-GNAI3 are likely to rely on TMCs for maintenance rather than initial enrichment.

Line 668:"In the absence of transduction [6] or tip links (this work), the EPS8-MYO15A complex is no longer exclusively targeted to row 1 and instead appears at all tips, also in proportion to length.

See section B2, "exclusively" is incorrect without mention of stage.

Line 739 title "Transduction proteins localize the row 1 complex"

As mentioned above, this is an overstatement. Maintenance appears more likely. 

Line 745: "TMIE and the TMCs do localize to row 1 near tips during early postnatal development [23]"

Line 748: "Accordingly, GNAI3 or GPSM2 may bind directly or indirectly to TMC channels localized to tips of row 1 stereocilia during development (Fig 15E).:

Line 825: "may be due to binding of GNAI3 to the complex at the tip of row 1, where it is transiently during development." 

These are all overstatements. See comments above about Cunningham 2020 not discussing row identity. In particular, images do not appear to show convincing enrichment of TMIE or TMC1/2 at row 1 tips.

MINOR POINTS

In abstract and throughout text, language changes between "tip-link" and "tip link". Ex line 27 vs line 30

Abstract Line 38:"Reduced rhodamine actin labeling at row 2 stereocilia tips of tip-link and transduction mutants suggests that..."

This is only true at P7.5 (during development) while at P21 the opposite is reported, an increase (a delay?). Maybe clarify the stage? In adults, transduction appears to instead stabilizes barbed-ends.

Line 71 & 73- repeated 'in transduction mutants'

Line 79- along with [31-33] consider citing also Caberlotto PNAS 2011 where Ush1g but also Cdh23 KO is reported to reduce row 2 and 3 stereocilia length. I think this publication is not cited. It describes variably lost stereocilia, an extreme outcome possibly mirrored in this study by the reduced number of row 3 stereocilia (Fig. 2I-J).

Line 80 and 82 - best to define 'membrane tenting' first, line 82 is defined after first use line 80

Line 84: "Tension in tip links thus promotes actin polymerization in the shorter stereocilia rows, independent of channel gating [37]."

Is this in fact experimentally verified? When tenting/tip-links are removed, MET channels are no longer gated, so how can the specific role of tension be assessed? Caberlotto et al. Bioarchitecture 2011 [37] is a speculative paper when it comes to how absence of Usher1 proteins leads to reduced row 2 and 3.

Comment on the Introduction: given the interesting results obtained in this study, the Introduction should probably remind readers that "tip-link" proteins form ubiquitous transient shaft links, and kinocilial links as well. As concluded by the authors, these links probably explain the bulk of stereocilia defects reported in Cdh23 and Pcdh15 mutants.

Line 156 - It might be useful in this text section to explicitly define what phase III and IV correspond to exactly.

Fig S1. Both panels E and F are labelled P7.5, is this correct?

Fig. 6 - It would be useful to have the genes indicated as caption in (A), as in previous figures. Het; Het or Het; KO is not clear when browsing figures. The full genotype was actually indicated in similar circumstances in Fig. S5.

Fig. S4 C, D: WHRN caption in merge but GNAI3 in single channel. Same issue in all the right column panels.

Line 283: figure citation appears incorrect. (Fig 4Gi-Fi) should be Fig 4Ai - Fi ? and (Fig 4Aii-Dii) should be Fig 4Ai-Di talking about width not CV.

Lines 292-294 wrong panels referenced in figure legend, there is no I-P in Figure 4

Line 296: what would be a systematic variation in length within a row? how would it be distinguished from stochastic? Or, how do results in Fig. 5 allow to decide for stochastic? If CV is high for both controls and mutants it is not the base of this distinction.

Line 355 Fig 6D-E should be Fig6D, F, etc. In general, correct panel labels for Figure 6 descriptions in Results main text.

Line 443 "heterozygotes from mutant IHCs" meaning is unclear

Line 454 Fig 10 I-L refers to localization of GPSM2, not GNAI3

Line 457 panel labels should be "…I and J, K and L"

Line 462 in Figure legend 10 seems redundant

Lines 477 & 478 error in panel labels in Figure 11 legend.

Line 567: "In Pcdh15av3J567 /+ controls, the high level of row 2 labeling by rhodamine-actin seen in control bundles at P7.5 (Fig 13A, M) had disappeared by P21.5 (Fig 13G, P);"

This sentence is a little confusing. It seems it could apply to all controls and not just Pcdh15 controls (i.e. free barbed ends are drastically lost at row 2 at P21.5 compared to P7.5).

Line 580: "By contrast, in Myo15ash2/sh2 mice, row 1 and row 2 rhodamine-actin labeling was elevated to the intensity level seen in row 1 from control mice at P7.5 (Fig 13E-F, O)".

This is confusing; should the sentence say "..elevated to the intensity level seen in row 2 from control mice at .."? It seems clear from images and graphs that sh2 mutants at P7 have a reduced accumulation at row 2 but increased at row 1 (uniform by row). 

In this figure, it is surprising to see how modest the raise in Rhod-actin signal is at P21 in the mutants compared to the control on the graphs. Visually, signal appears to be absent in controls and massively increased in mutants. This is probably linked to the Y axis range adopted for much higher signals in earlier controls at row 2?

Line 528-529 Figure 12 legend: Cdh23 mutants not shown in Figure 12. Error in panel labels and lack of GPSM2 mention in line 531. Legend missing for M-P

Line 624: Error in panel labelling. Fig 15D is EPS8, E is GPSM2, F capping proteins

Line 667, 679: the reference for MYO15A-EPS8 at all tips and independent GNAI3 there is Tadenev 2019 and not Tarchini 2016 [7].

Line 703: Absent kinocilial links could well explain part of the defects as well, especially the lack of packed distribution. Besides adding links to the Introduction (see above), this part of the Discussion could be broadened to discuss kinocilial links as well.

Line 755:"during stage III, row 2 stereocilia shorten when row 1 lengthens,.."

This is confusing, as here stage III row 1 stereocilia were shown to be very stable and not to lengthen (Fig. 1G). Or should this sentence be about stage IV, not III ?

Line 787: The Discussion at this point appears to bring back earlier discussion points and show some degree of redundancy, as if two drafts were fused at this position without fully consolidating by topic. Possibly a similar comment starting at Line 863, "How the mutant phenotypes arise".

Related: two ideas are presented in two different locations for a potential binding between GPSM2-GNAI3 and the transduction apparatus:

Line 749: "This coupling could be mediated by the established CIB2-WHRN interaction [85];"

Line 825: "may be due to binding of GNAI3 to the complex at the tip of row 1, where it is transiently during development." 

Reviewer #3: General comments

Krey et al. provide a comprehensive description of how cochlear inner hair cell (IHC) bundles acquire their stereotypical shape: 3 rows in an uneven stair-step arrangement of tightly-regulated heights. They probe developmental regulatory mechanism with several types of mutants: transduction mutants affecting molecules that belong to the mechanoelectrical transduction apparatus at the tips of rows 2 and 3; tip link mutants in cadherins that connect the transducing tips of rows 2 and 3 to taller stereocilia in rows 1 and 2, respectively; mutations in G-protein-related proteins that congregate at stereociliary tips; and mutants affecting actin-capping and -severing proteins. 

The data are quite beautiful and the writing is a model of clarity, especially given the quantity of data being presented. Setting up the staircase of stereocilia depends in complex ways on both the groups of proteins just mentioned and on developmental time, with the hair bundle passing through a series of stages to reach the mature configuration. Krey et al. lay out, in considerable but not excessive detail, the morphological changes in control IHCs and the various mutant IHCs as functions of time and hair bundle stage, and pull the data together in a summary model, graphically illustrated in Fig. 15. At several stages - especially Introduction, Discussion and Conclusions - they summarize the significance of the new observations, better than I will do here. 

The work leverages the mammalian hair bundle for its exceptional value as a model for cellular structure-function studies. Building on previous work (well-summarized), the authors offer a number of new insights - notably, separating the accumulation of stereocilia 'building blocks" from their regulation and isolating structural development roles for molecules already known to be necessary for gating and expressing mechanotransduction. Along the way, new information addresses previous arguments in the literature, specifically indicating that Tmie, part of the met channel complex, does not interact with protocadherin 15, and that members of the row 1 tip complex do not co-assemble at once. Given the importance of hair bundle structure for filtering different kinds of mechanical stimuli, these carefully curated and detailed data invite comparative studies on the regulation of other hair bundles. 

Suggestions/ questions:

1. Graphical illustrations - Figure 15 was so welcome that I wonder whether individual, smaller scale, summary diagrams at the end of each major set of experiments would help readers keep the important observations in mind (or at least this reader).

2. Tip links and row 3 vs row 2 asymmetries - 

a. A major discussion point, tip links are only depicted in the SEMs of Fig. 2 - for the sake of non-aficionados, I suggest finding a way to include them in Fig 15. 

b. Given the importance of tip-link tension and tip tenting to the model of hair bundle growth, what is known about how tip links compare across rows - are they of similar length and angle for rows 2 and 3? 

c. In Fig 15 legend, transduction is said to lead to "accumulation of DSTN and CFL1 at row 2 tips, triggering…shrinkage of row 2". What is going on at row 3 - which also has transduction, but apparently no accumulation of DSTN and CFL1 ? and ends up remarkably thinner than row 2.

Detailed comments

Line 377 - I did not find row 3 stereocilia obvious in the control bundles in Fig. 7A-D - label?

Line 873 - doesn't scan - "…slowly pruning the shorter rows"?

Fig. 15 - Legend does not match figure - looks as though panel C was added late. Apparently it is meant to show the time course of tip link and transduction appearance, but best to make that clear by showing the stages underneath, as

---

## [Editor Report · Decision Letter 2]

11 Mar 2023

Dear Dr Barr-Gillespie,

Thank you for your patience while we considered your revised manuscript entitled "Control of stereocilia length during development of hair bundles" for publication as a Methods and Resources at PLOS Biology. This revised version of your manuscript has been evaluated by the PLOS Biology editors and by the Academic Editor.

Based on our Academic Editor's assessment of your revision, we are likely to accept this manuscript for publication, provided you satisfactorily address the data and other policy-related requests stated below.

In addition, we would like you to consider a suggestion to improve the title:

"The actin-bundling protein EPS8 controls stereocilia length during the development of hair bundles"

We expect to receive your revised manuscript within two weeks. 

*Published Peer Review History*

*Press*

Sincerely,

Ines

--

Ines Alvarez-Garcia, PhD

Senior Editor

PLOS Biology

Thank you for providing the data underlying all the graphs shown in the figures in Figshare. However, I am missing the data for the following figure, so please provide this or let us know where is the file:

Fig. 2I-L

Also the following files seem to be mislabelled. Please add the correct label to the files deposited in Figshare:

Fig. 6D-K data should be Fig. 6G-N

Fig. 9E-G data should be Fig. 9I-K

Fig. S4A-H data should be Fig. S4M-T

---

## [Editor Report · Decision Letter 3]

14 Mar 2023

Dear Dr Barr-Gillespie,

Thank you for the submission of your revised Methods and Resources manuscript entitled "Control of stereocilia length during development of hair bundles" for publication in PLOS Biology. On behalf of my colleagues and the Academic Editor, Walter Marcotti, I am delighted to say that we can in principle accept your manuscript for publication, provided you address any remaining formatting and reporting issues. These will be detailed in an email you should receive within 2-3 business days from our colleagues in the journal operations team; no action is required from you until then. Please note that we will not be able to formally accept your manuscript and schedule it for publication until you have completed any requested changes.

PRESS

Sincerely, 

Ines

--

Ines Alvarez-Garcia, PhD

Senior Editor

PLOS Biology
